# Fibrillarin homologs regulate translation in divergent cell lineages during planarian homeostasis and regeneration

Jiajia Chen [1,2,3,4], Yucong Li [2,3,4,5], Yan Wang[6], Hui Wang[7,8,9], Jiaqi Yang[6], Xue Pan[1,2,3,4], Yun Zhao[2,3,4,5], Hao Xu[1,2,3,4], Penglei Jiang[7,8,9], Pengxu Qian [7,8,9], Hongwei Wang [6], Zhi Xie[6] & Kai Lei [2,3,4✉]

## Abstract

Tissue homeostasis and regeneration involve complex cellular changes. The role of rRNA modification-dependent translational regulation in these processes remains largely unknown. Planarians, renowned for their ability to undergo remarkable tissue regeneration, provide an ideal model for the analysis of differential rRNA regulation in diverse cell types during tissue homeostasis and regeneration. We investigated the role of RNA 2′-O-methyltransferase, Fibrillarin (FBL), in the planarian *Schmidtea mediterranea* and identified two FBL homologs: *Smed-fbl-1* (*fbl-1*) and *Smed-fbl-2* (*fbl-2*). Both are essential for planarian regeneration, but play distinct roles: *fbl-1* is crucial for progenitor cell differentiation, while *fbl-2* is important for late-stage epidermal lineage specification. Different 2′-O-methylation patterns were observed upon *fbl-1* and *fbl-2* knockdown, suggesting their roles in translation of specific mRNA pools during regeneration. Ribo-seq analysis further revealed differing impacts of *fbl-1* and *fbl-2* knockdown on gene translation. These findings indicate divergent roles of the duplicate *fbl* genes in specific cell lineage development in planarians and suggest a role of rRNA modifications in translational regulation during tissue maintenance and regeneration.

**Keywords** Cell Differentiation; Duplicated Fibrillarin; Epidermal Lineage; Planarian; rRNA Modification
**Subject Categories** Development; Post-translational Modifications & Proteolysis; Translation & Protein Quality

## Introduction

Translational regulation is crucial to a myriad of cellular processes. Numerous studies have suggested an integral role for enhancing ribosome biogenesis and protein synthesis in stem cells and cell fate determination across various species (Gay et al, 2022; Lv et al, 2021; Sanchez et al, 2016; Zhang et al, 2014). Many studies have investigated mRNA modifications, mRNA regulation, and protein synthesis in tissue renewal and regeneration (Bansal et al, 2017; Cui et al, 2023; Dagan et al, 2022; David, 2022; Schaeffer et al, 2023; Zhulyn et al, 2023). However, the implication of translational regulation through rRNA modification on stem cell proliferation and differentiation during adult tissue turnover and regeneration remains largely unexplored.

Fibrillarin (FBL) is a nucleolar rRNA 2′-*O*-methyltransferase required for rRNA processing. The main structure of FBL contains two functional domains: a glycine/arginine-rich (GAR) region with nuclear localization signals and an RNA binding domain with methyltransferase (MTase) activity (Pereira-Santana et al, 2020; Rodriguez-Corona et al, 2015). Depending on its functional domain, FBL catalyzes the 2′-*O*-methylation of rRNA to process pre-rRNA into 18S rRNA and 28S rRNA. Previous studies have illustrated that FBL functions in the nucleolus to process rRNA through liquid-liquid phase separation via its GAR domain (Yao et al, 2019). Guided by small nucleolar RNA (snoRNA), FBL participates in different cellular processes (Li et al, 2018; Ren et al, 2019; Yi et al, 2021). In vertebrate development, FBL is a crucial regulator in embryonic stem cells (ESCs), maintaining pluripotency and influencing cell viability and differentiation through the p53 signaling pathway (Morral et al, 2020; Watanabe-Susaki et al, 2014; Zhang et al, 2014). Its inhibition or mutation can disrupt early mouse development, cell cycle progression, and neuron differentiation (Bouffard et al, 2018; Newton et al, 2003; Wu et al, 2022). In the animal kingdom, the evolution of FBL into two clusters, FBL and FBL-like, underscores the importance of FBL duplication across various species (Pereira-Santana et al, 2020).

[1]School of Life Sciences, Zhejiang University, Hangzhou, Zhejiang, China. [2]Westlake Laboratory of Life Sciences and Biomedicine, Hangzhou, Zhejiang, China. [3]Key Laboratory of Growth Regulation and Translational Research of Zhejiang Province, School of Life Sciences, Westlake University, Hangzhou, Zhejiang, China. [4]Institute of Biology, Westlake Institute for Advanced Study, Hangzhou, Zhejiang, China. [5]Fudan University, Shanghai, China. [6]State Key Laboratory of Ophthalmology, Zhongshan Ophthalmic Center, Sun Yat-sen University, Guangdong Provincial Key Laboratory of Ophthalmology and Vision Science, Guangzhou, China. [7]Center for Stem Cell and Regenerative Medicine and Bone Marrow Transplantation Center of the First Affiliated Hospital, Zhejiang University School of Medicine, Hangzhou 310058, China. [8]Liangzhu Laboratory, Zhejiang University, 1369 West Wenyi Road, Hangzhou 311121, China. [9]Institute of Hematology, Zhejiang University & Zhejiang Engineering Laboratory for Stem Cell and Immunotherapy, Hangzhou 310058, China. ✉E-mail: leikai@westlake.edu.cn

Despite the low expression level of FBL-like in mammals, which has made its function less studied, the roles of FBL homologs in rRNA modification and subsequent events during tissue turnover and regeneration remain largely unknown. This highlights the need for a more in-depth exploration of the functions of FBL homologs in organisms with regenerative capabilities.

Planarians serve as a remarkable animal model for investigating regeneration mechanisms due to their unique whole-body regenerative capabilities. This allows for a comprehensive exploration of gene functions within the context of stem cell biology (Reddien, 2018). Tissue regeneration is a sophisticated process necessitating precise regulation at various levels, including transcription, post-transcription, translation, and post-translation. Precise regulations of stem cell proliferation and differentiation are critical for cell renewal and regeneration after amputation or injury (Scimone et al, 2014). Recent multi-omics studies have begun to dissect the regulatory mechanisms of planarian regeneration, unveiling the role of genomic networks in polarity remodeling via the Wnt signaling pathway, the expression of fate-specific transcription factors during the G2/M phase of the cell cycle, and even epigenetic regulation by m6A across various cell types (Dagan et al, 2022; David, 2022; Pascual-Carreras et al, 2023; Raz et al, 2021). However, our understanding of translational regulation, a crucial effector of cellular processes, remains limited during planarian regeneration.

Our study identified two fibrillarin homologs in planarian *Schmidtea mediterranea*, *Smed-fbl-1* (*fbl-1*) and *Smed-fbl-2* (*fbl-2*). We found that *fbl-1* and *fbl-2* are differentially expressed in neoblasts and epidermal lineage cells, respectively. Knockdown of *fbl-1* and *fbl-2* influenced site-specific 2′-O-methylation and specific mRNA translation. Our analysis of transcription and translation efficiency revealed that regulation of alternative splicing, neurotransmitter secretion, and Wnt signaling are differentially governed by *fbl-1* or *fbl-2*, respectively. This study unveiled that *fbl* homologs in planarians have evolved to regulate distinct cell types, providing insight into the conserved function and adaptive evolution of duplicate *fbl* genes in planarians. This marks a step forward in our understanding of translational regulation in adult tissue homeostasis and regeneration.

# Results

## Two homologs of *fbl* are required for planarian homeostasis and regeneration

Based on the glycine-arginine-rich (GAR) and methyltransferase (MTase) protein domains of the human FBL protein, we identified two homologs of FBL in planarian *Schmidtea mediterranea*, termed *smed-fbl-1* (*fbl-1*) and *smed-fbl-2* (*fbl-2*) (Fig. EV1A). Unlike the mammalian FBL and FBL-like-1 groups, the planarian FBL proteins are included in a closely related sister group, similar to those found in Xenopus and hydra (Fig. EV1A). By aligning the protein sequences of the GAR domain from fourteen different species, we observed a clustering pattern in which residues 73–74 were conserved between the homologs of non-mammalian species (Fig. EV1B). To investigate the functions of *fbl* homologs in planarians, we performed RNAi experiments to knockdown (KD) *fbl-1* and *fbl-2*, respectively. We used quantitative real-time PCR

(qPCR) and whole-mount in situ hybridization (WISH) to confirm the KD efficiency, which was over 80% at the mRNA level (Fig. EV1C,D). KD of either *fbl* did not influence the expression of the other (Fig. EV1C,D).

To understand the functions of *fbl-1* and *fbl-2* in tissue regeneration, we performed the RNAi feeding and amputated the KD animals at 7 days post-feeding (dpf) (Fig. 1A). At 7 dpf, *fbl-1* and *fbl-2* KD animals were morphologically similar to *egfp* KD controls (Fig. 1B). In a long-term observation, *fbl-1* KD and *fbl-2* KD animals began to die at 16 dpf and 20 dpf, respectively (Fig. 1C). Moreover, the *fbl-1* KD animals failed to regenerate the anterior and posterior tissues, while the *fbl-2* KD animals showed slower regeneration and smaller blastema at 7 days post-amputation (dpa) (Fig. 1D). Staining of tissues confirmed the regeneration defects after KD of *fbl-1* or *fbl-2*, with incomplete central nervous system and intestines, accounting for smaller ratio of brain length to body length (Fig. EV1E,F). The observed regeneration defects correlated with disrupted anterior (*notum*) and posterior (*wnt-1*) patterning reestablishment in both *fbl* KD animals at 72 hpa (Fig. 1E–I). These observations showed that the *fbl* homologs were both required for planarian homeostasis and regeneration.

## *fbl-1* and *fbl-2* are expressed in distinct cell types

To elucidate the functions of *fbl-1* and *fbl-2* in planarian biology, we first examined the expression patterns of these two genes. Colorimetric WISH and fluorescence in situ hybridization (FISH) analysis were performed and showed that the two *fbl* homologs exhibited distinct expression patterns (Figs. 2A,B and EV2A). Different from the expression of *fbl-2*, the expression of *fbl-1* exhibited irradiation sensitivity, which was similar to that of *piwi-1*+ cells (neoblasts) (Fig. EV2A,B). The dual FISH (dFISH) experiments showed that *fbl-1* is widely expressed in diverse cell types, including neoblasts (*piwi-1*+), epidermal (*prog-1*+), neural (*ston2*+), and intestinal progenitors (*hnf4*+) (Fig. 2C,D). The signals of *fbl-1* were evident in *piwi-1*+ neoblasts and *prog-1*+ epidermal early progenitors (Fig. 2D). Different from the expression pattern of *fbl-1*, the spatial expression pattern along the dorsoventral (DV) axis revealed that the *fbl-2*+ cells were distributed from the dorsal parenchyma to the dorsal epidermis (Fig. 2A,B).

To further examine the expression patterns of *fbl-1* and *fbl-2*, we conducted WISH and analyzed two published RNA-seq datasets on planarian regeneration (Zeng et al, 2018; Data ref: Zeng et al, 2018; Scimone et al, 2022; Data ref: Scimone et al, 2022) (Dataset EV1). The expression of *fbl-1* was upregulated at 48 hpa (Fig. EV2C,E,F). This upregulation coincided with the upregulation of several stem cell markers (*piwi-1*, *vasa-1*, *mcm2*) (Fig. EV2E). Additionally, the expression of *fbl-1* was induced in anterior-facing wounds, similar to the expression of stem cell markers (*piwi-1*, *vasa-1*, *mcm2*) observed in WISH and RNA-seq (Fig. EV2C,F). These results suggest that the expression of *fbl-1* is enriched in neoblasts.

The upregulation of *fbl-2* was correlated with the increased expression of *fos-1*, *follistatin*, *glypican-1*, and certain epidermal marker genes such as *egr-5* and *AGAT-1* during regeneration (24 hpa) (Fig. EV2D,E). *Follistatin*, *glypican-1*, and *AGAT-1* are expressed in post-mitotic cells of the muscular and epidermal lineages, respectively, and are essential for planarian regeneration (Benham-Pyle et al, 2021). WISH and RNA-seq data also showed that the expression of *fbl-2* is increased at the anterior- or posterior-

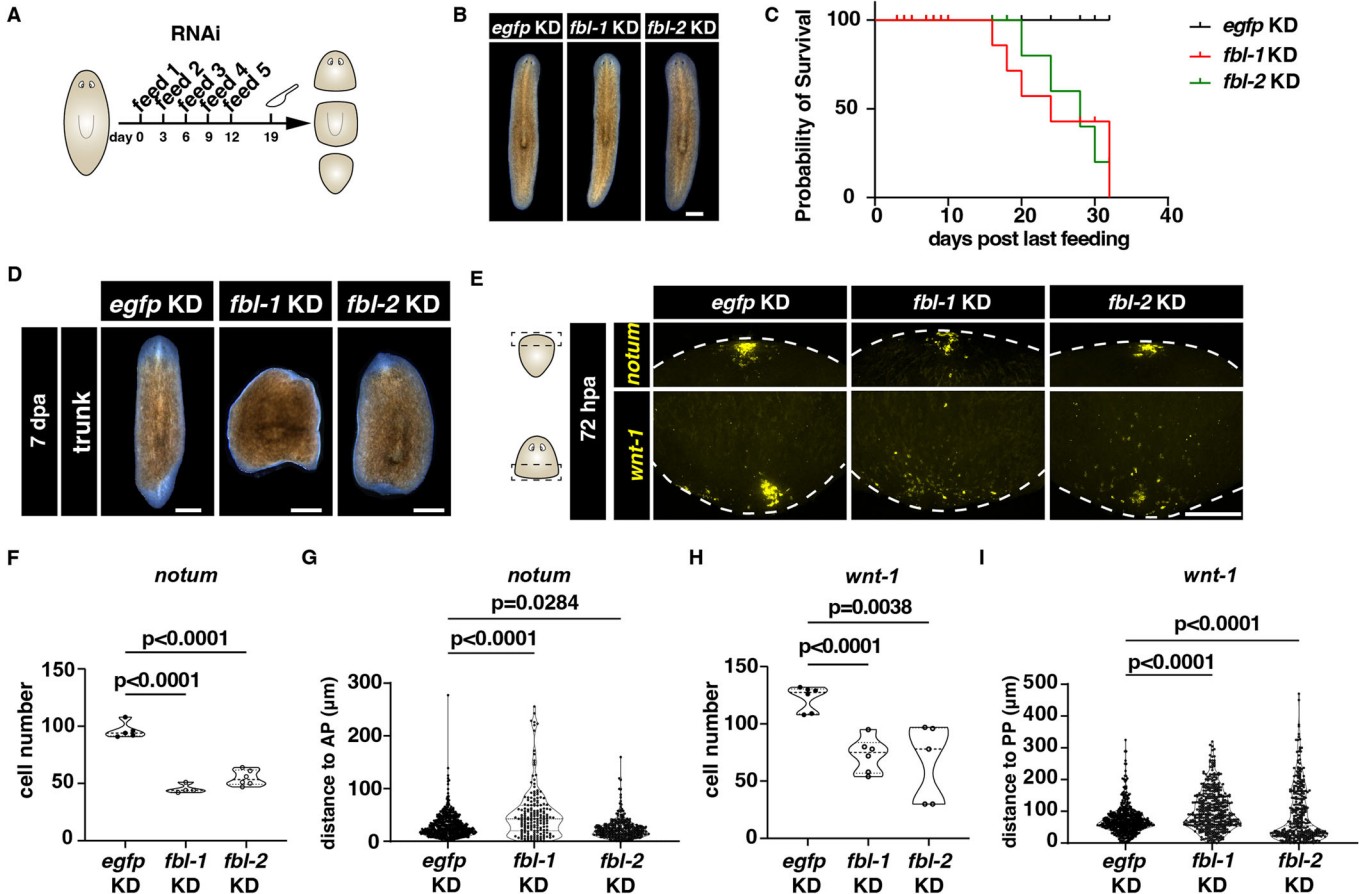

**Figure 1. Two *fbl* homologs are required for planarian homeostasis and regeneration.**

(A) Schedule of RNAi feeding every 3 days and amputation of *fbl-1* KD and *fbl-2* KD animals. (B) Live images show normal morphology upon *fbl-1* KD and *fbl-2* KD at 7 dpf. Scale bars = 500 μm. (C) Survival curve of *egfp* KD (n = 90), *fbl-1* KD (n = 98), and *fbl-2* KD (n = 92) animals. (D) Live images show defective regeneration of trunk fragments from *fbl-1* KD and *fbl-2* KD animals at 7 dpa compared to the *egfp* KD controls. The anterior side is upward for each animal. n = 30. Scale bar = 500 μm. (E) FISH images show the expression patterns of anterior (*notum*) and posterior (*wnt-1*) pole marker expression in tails and heads of *egfp* KD, *fbl-1* KD, and *fbl-2* KD animals at 72 hpa. Scale bar = 20 μm. Cartoon illustrations show the regions that were displayed for the expression of *notum* and *wnt-1*, respectively. (F) Violin plot of *notum*+ cell number quantification in tails of *egfp* KD, *fbl-1* KD, and *fbl-2* KD animals at 72 hpa. Each dot represents the cell number measured from an individual animal. n = 5 for each condition. Two-tailed unpaired student's *t*-test calculated the p values. Data were represented as mean ± SEM. (G) Violin plot of the quantification of the distance between *notum*+ cell and the anterior tip in tails of *egfp* KD, *fbl-1* KD, and *fbl-2* KD animals at 72 hpa. Each dot represents the cell distance measured from an individual *notum*+ cell from *egfp* KD (n = 5), *fbl-1* KD (n = 4), and *fbl-2* KD (n = 6) animals. Two-tailed unpaired student's *t*-test calculated the p values. Data were represented as mean ± SEM. (H) Violin plot of *wnt-1*+ cell number quantification in heads of *egfp* KD, *fbl-1* KD, and *fbl-2* KD animals at 72 hpa. Each dot represents the cell number measured from an individual animal. n = 6 for each condition. Two-tailed unpaired student's *t*-test calculated the p values. Data were represented as mean ± SEM. (I) Violin plot of the quantification of the distance between *wnt-1*+ cell and the posterior tip in heads of *egfp* KD, *fbl-1* KD, and *fbl-2* KD animals at 72 hpa. Each dot represents the cell distance measured from an individual *wnt-1*+ cell in *egfp* KD (n = 6), *fbl-1* KD (n = 6), and *fbl-2* KD (n = 5) animals. Two-tailed unpaired student's *t*-test calculated the p values. Data were represented as mean ± SEM. Source data are available online for this figure.

facing wound sites during the regeneration (24 hpa) (Fig. EV2D,F). These results suggest that the expression of *fbl-2* is enriched in post-mitotic cells and can be induced by injury or may correlate with the emergence of certain post-mitotic cells.

Given these expression patterns, we hypothesized that *fbl-2* is expressed in epidermal lineage cells. Planarian epidermal cell development involves multiple progenitor types (Cheng et al, 2018; Tu et al, 2015; Wurtzel et al, 2015; Zhu et al, 2015; Zhu and Pearson, 2018). The *prog-1*+ and *AGAT-1*+ cells have been identified as early and late progenitors of the epidermal cell lineage, respectively (Eisenhoffer et al, 2008). *egr-5* was post-mitotic epidermal cell markers that regulate epidermal differentiation and maturation (Tu et al, 2015). *AGAT-1* was further characterized as

specifiers of mature epidermis, while *vim-3*+ cells emerged following the *zpuf-6*+ transient state (Tu et al, 2015). To further examine in which cell type *fbl-2* was expressed, dFISH of *fbl-2*+ cells with epidermal lineage markers revealed that *fbl-2* is expressed predominantly in *egr-5*+ cells, with a high coexpression ratio (*egr-5*+*fbl-2*+: 80.4%). Smaller proportions of coexpression were observed in *AGAT-1*+ (4.4%), *AGAT-3*+ (6.89%), *zpuf-6*+ (11.3%), *vim-1*+ (4.17%), and *vim-3*+ (10.35%) cells (Fig. 2D). During regeneration, dFISH analyses of *fbl-2* and epidermal lineage markers revealed the enriched expression of *fbl-2* in *vim-3*+ cells (*vim-3*+*fbl-2*+: 85.1%) (Fig. 2E). The differentiation from *fbl-2*+*egr-5*+ cells to *fbl-2*+*vim-3*+ cells gradually increased back to a high ratio of *fbl-2*+*egr-5*+ cells in later stages of regeneration (Fig. 2E,F). This

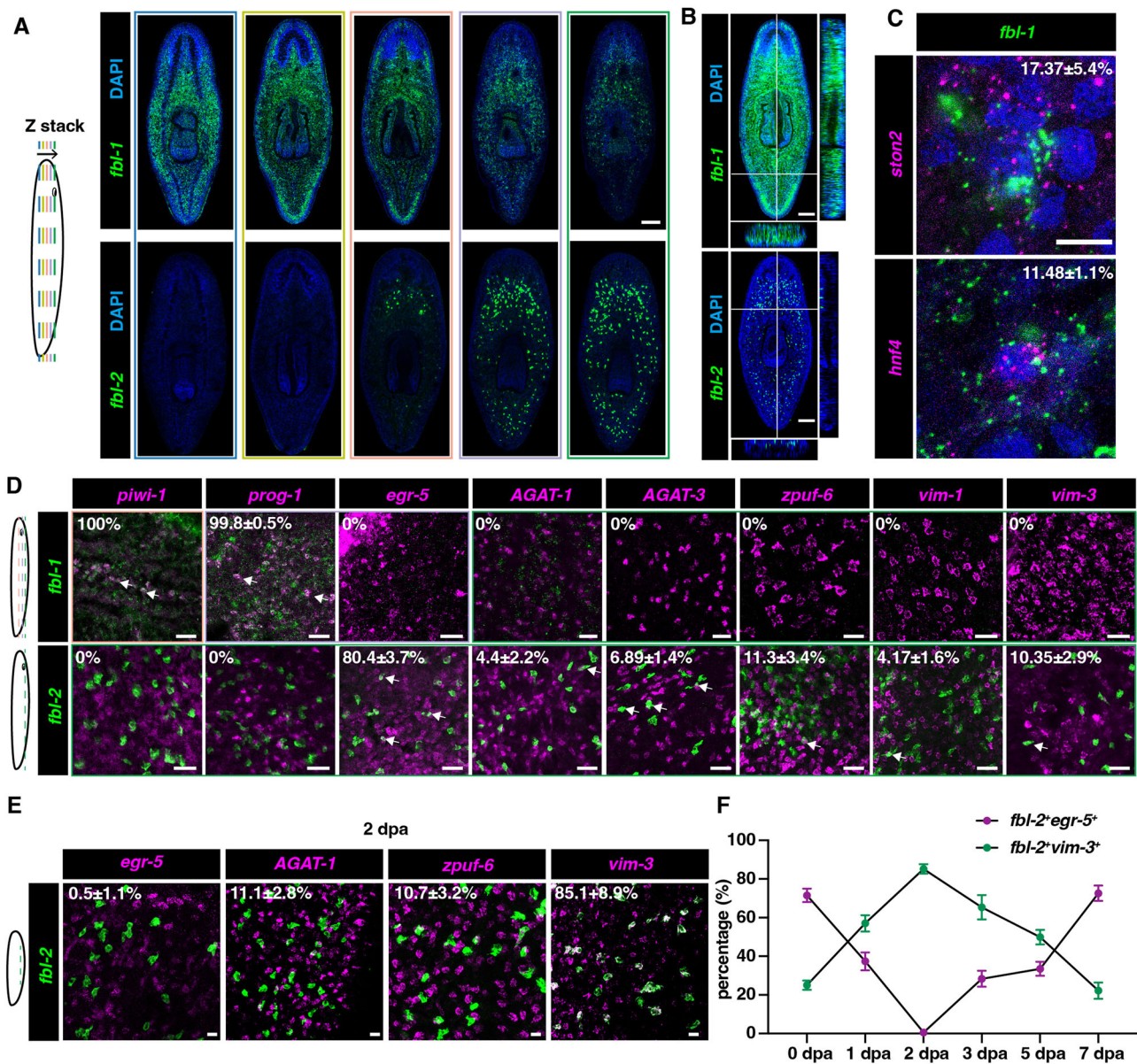

**Figure 2. Expression of *fbl-1* and *fbl-2* in distinct cell lineages.**

(A) The expressed patterns of *fbl-1* and *fbl-2* in planarians by FISH. The order of expression pattern is shown as the z stack in the cartoon illustration. The color lines in the cartoon illustration indicate the displayed focal panels. Scale bar = 200 μm. (B) Orthogonal views of the expression patterns of *fbl-1* and *fbl-2* in planarians by FISH. Scale bar = 200 μm. (C) Dual FISH of *fbl-1* transcripts with neural progenitor marker (*ston2*) and intestine progenitor marker (*hnf4*) in intact worms. The percentages indicate the ratio of *fbl-1*+ cells within each cell type. *n* = 6. Data were represented as mean ± SEM. Scale bar = 10 μm. (D) Dual FISH of *fbl-1* or *fbl-2* transcripts with pan-neoblast marker (*piwi-1*) and epidermal lineage markers (*prog-1*, *egr-5*, *AGAT-1*, *AGAT-3*, *zpuf-6*, *vim-1*, and *vim-3*) in intact worms. The percentages indicate the ratio of *fbl-1*+ or *fbl-2*+ cells within each cell type. White arrows highlight the double positive cells. *n* = 3. Scale bar = 50 μm. (E) Dual FISH of *fbl-2* transcripts with epidermal lineage markers (*egr-5*, *AGAT-1*, *zpuf-6*, and *vim-3*) at 48 hpa. The percentages indicate the ratio of *fbl-2*+ cells within each cell type. *n* = 3. Scale bar = 20 μm. (F) Quantification of *fbl-2*+*egr-5*+ and *fbl-2*+*vim-3*+ cells during regeneration at 0 dpa, 1, 2, 3, 5, and 7 dpa. *n* = 6 animals for each time point. Source data are available online for this figure.

differential proportion of *fbl-2*+*egr-5*+ cells between homeostasis and regeneration, coupled with an opposing proportion for *fbl-2*+*vim-3*+ cells, suggests that *fbl-2* is involved in the development of epidermal cells and possesses a distinct function during regeneration.

In summary, our findings suggested that *fbl-1* and *fbl-2* regulate planarian cell development via their spatiotemporal expression in different cell types.

### *fbl-1* is required for stem cell proliferation and differentiation into early progenitors, and *fbl-2* regulates epidermal integrity

Given the observed expression of *fbl-1* in neoblasts and progenitor cells, we suspected that *fbl-1* might be crucial for neoblast proliferation and cell differentiation to maintain tissue turnover. To test this, we observed a decrease in the number of neoblasts

($piwi$-$1^+$), and a significant reduction of H3P$^+$ cells following KD of $fbl$-$1$ (Figs. 3A,B and EV3A,B). We then analyzed $fbl$-$1$ KD planarians at 7 dpf for expression levels of progenitor markers. Notably, we found disrupted expression of an epidermal early progenitor marker $prog$-$1$ and intestinal progenitor and mature cell markers $hnf4$ and $gata4/5/6$ in $fbl$-$1$ KD animals (Fig. EV3C). Moreover, we observed a decrease in newly differentiated progenitors in $fbl$-$1$ KD animals, evidenced by the reduction of $ovo^+$PIWI-$1^+$, $prog$-$1^+$PIWI-$1^+$, and $hnf4^+$PIWI-$1^+$ cells compared with those in $egfp$ KD controls at 24, 48, and 72 hpa (Fig. 3C-H). Thus, our results indicate that $fbl$-$1$ is essential for stem cell proliferation and differentiation during tissue turnover and regeneration. This conclusion, however, does not exclude the possibility that a general neoblast defect leads to a reduction in progenitor production.

$fbl$-$2$ KD planarians showed no obvious change in the stem cell population but a noticeable decrease in proliferating cells at 14 dpf (Fig. 3I,J). Since the expression of $fbl$-$2$ is enriched in the epidermal lineage, this effect is likely to be an indirect feedback influence or a consequence of the animals that are approaching a declined mortality. We further investigated whether $fbl$-$2$ is involved in supporting epidermal cell differentiation. Distinct from $fbl$-$1$ KD planarians, $fbl$-$2$ KD resulted in a decreased $AGAT$-$1^+$ epidermal cell number compared with $egfp$ KD animals at 14 dpf (Fig. 3K). We subsequently analyzed the expression of genes expressed in $AGAT$-$1^+$ cells using qPCR and FISH (Fig. EV3D,E). It was worth noting that these genes were not completely expressed in the same cell populations (Tu et al, 2015; Zhu et al, 2015). While the expression of $egr$-$5$ and $zpuf$-$6$ increased, most of the other genes expressed in $AGAT$-$1^+$ cells were not changed according to the qPCR analysis (Fig. EV3D). However, using FISH, we detected a decrease in the cell number of $AGAT$-$1^+$, $vim$-$1^+$, and $vim$-$3^+$ cells, but an increase of the $zpuf$-$6^+$ cells following $fbl$-$2$ KD (Fig. EV3E). The difference in the results of qPCR and FISH could be due to the different sensitivity in the detection of small population cells out from the whole-body cells, and on the levels of transcripts versus cell numbers for each analysis. With long-time culture until 21 dpf, $fbl$-$2$ KD worms displayed damaged tails, in which exhibited reduced expression of $vim$-$3$ at both intact and damaged regions (Fig. EV3F–H). In contrast, $AGAT$-$1^+$ cells were consistently decreased in the intact region, but an accumulation of $AGAT$-$1^+$ cells was observed in the damaged region (Fig. EV3F,G,I). This observation suggested that the expression of $AGAT$-$1$ was not directly regulated by $fbl$-$2$, or the induced expression of $AGAT$-$1$ by injury was regulated by an independent mechanism. Additionally, consistent with the defects in homeostasis, $fbl$-$2$ KD animals exhibited loss of epidermal integrity and cell number following amputation with induced expression of $AGAT$-$1$ (Fig. 3L–N). Based on the expression pattern of $fbl$-$2$, our findings suggested that $fbl$-$2$ might promote the differentiation of a subset of epidermal cell lineage consisting of $egr$-$5^+fbl$-$2^+$ into $vim$-$3^+$ cells. When the process was impeded by $fbl$-$2$ KD, $zpuf$-$6$ expression would be accumulated.

In summary, we proposed that two FBL homologs, $fbl$-$1$ and $fbl$-$2$, play crucial but distinct roles in planarian tissue homeostasis and regeneration. $fbl$-$1$ appears to be critical for stem cell proliferation and differentiation of multiple cell lineages. In contrast, $fbl$-$2$ that is expressed in cells of the epidermal lineage, appears to regulate the specification of $egr$-$5^+$ cells. The defective

phenotypes in the reduction of late-stage differentiated cells in $fbl$-$1$ KD and $fbl$-$2$ KD animals may be the outcome of the defects in progenitor cells at earlier stages. Our findings emphasize the essential functions of $fbl$-$1$ and $fbl$-$2$ in planarian tissue homeostasis and regeneration. Based on these results, we carried out further experiments to assess the hypothesis that the specific rRNA modification by $fbl$ in different cell types augments ribosome heterogeneity, thereby preferentially facilitating the translation of proteins required for particular cell types.

## $fbl$-$1$ and $fbl$-$2$ knockdown reduce the methylation level of specific sites of 18S and 28S rRNA

Previous studies have shown that FBL depletion reduced 2′-O-methylation of rRNA in Hela cells and Xenopus (Delhermite et al, 2022; Erales et al, 2017). To validate the methylation activity of $fbl$-$1$ and $fbl$-$2$, we performed RiboMeth-seq to map and quantify rRNA 2′-O-methylation (Nm) in regeneration at 48 hpa after $fbl$-$1$ and $fbl$-$2$ KD using three biological repeats and $egfp$ KD as control with six biological repeats (Fig. 4A). The Methscore was calculated to assess the frequency of methylation at each nucleotide in 18S and 28S rRNA (Delhermite et al, 2022; Marchand et al, 2016; Sharma et al, 2017). The Methscore threshold >0.85 and <1 for fully modified sites and 0.65–0.85 for partially methylated sites was used in this study (Dataset EV2). To identify potential methylation sites in planarians, we examined the overlapping subsets of fully and partially modified sites from two ends in both 18S and 28S rRNAs depending on Methscore in the $egfp$ KD control group (Fig. EV4A–D). We discovered 10 fully methylated sites in 18S rRNA (Am28, Am755, Um1228, Gm1258, Um1266, Gm1268, Am1323, Um1377, Gm1425, and Cm1638), and 10 fully methylated sites in 28S rRNA (Am312, Gm313, Gm935, Am937, Am947, Am1049, Am1266, Cm1587, Am1599, and Gm1600) from 3′ and 5′ ends (Fig. EV4A,B; Dataset EV2). In addition, 14 and 9 partially methylated sites were identified in 18S rRNA (Gm152, Am161, Am338, Gm358, Gm642, Am647, Um665, Gm866, Gm936, Gm1017, Gm1325, Am1326, Gm1426, and Am1754) and 28S rRNA (Um181, Gm204, Am315, Gm397, Am504, Gm551, Am880, Gm987, and Gm1048), respectively (Fig. EV4C,D; Dataset EV2).

To identify the reported methylation sites, all known sites in human, mouse, yeast, zebrafish, and Xenopus were summarized in Dataset EV2 (Delhermite et al, 2022; Jansson et al, 2021; Ramachandran et al, 2020). In human cells, a total of 113 2′-O-methylation sites have been identified, with 42 sites located in 18S rRNA and 69 sites in 28S rRNA (Jansson et al, 2021). Those sites in human included most of the known sites in other species. We first aligned the sequences of 18S and 28S rRNA between planarians and humans. Thirty-four nucleotide sites in 18S rRNA and 16 nucleotide sites in 28S rRNA were the same between the two species (Fig. EV4E,F; Dataset EV2). Compared with methylation sites in other species, planarian possessed eight conserved fully methylated sites in 18S rRNA (Am28, Um1228, Um1266, Gm1268, Am1323, Um1377, Gm1425, and Cm1638), and seven conserved fully methylated sites in 28S rRNA (Gm935, Am937, Am947, Am1266, Cm1587, Am1599, and Gm1600) (Fig. EV4E,F; Dataset EV2). Thus, two fully methylated sites in 18S rRNA, three fully methylated 28S rRNA, and all the partially methylated sites we have identified were unique in planarians.

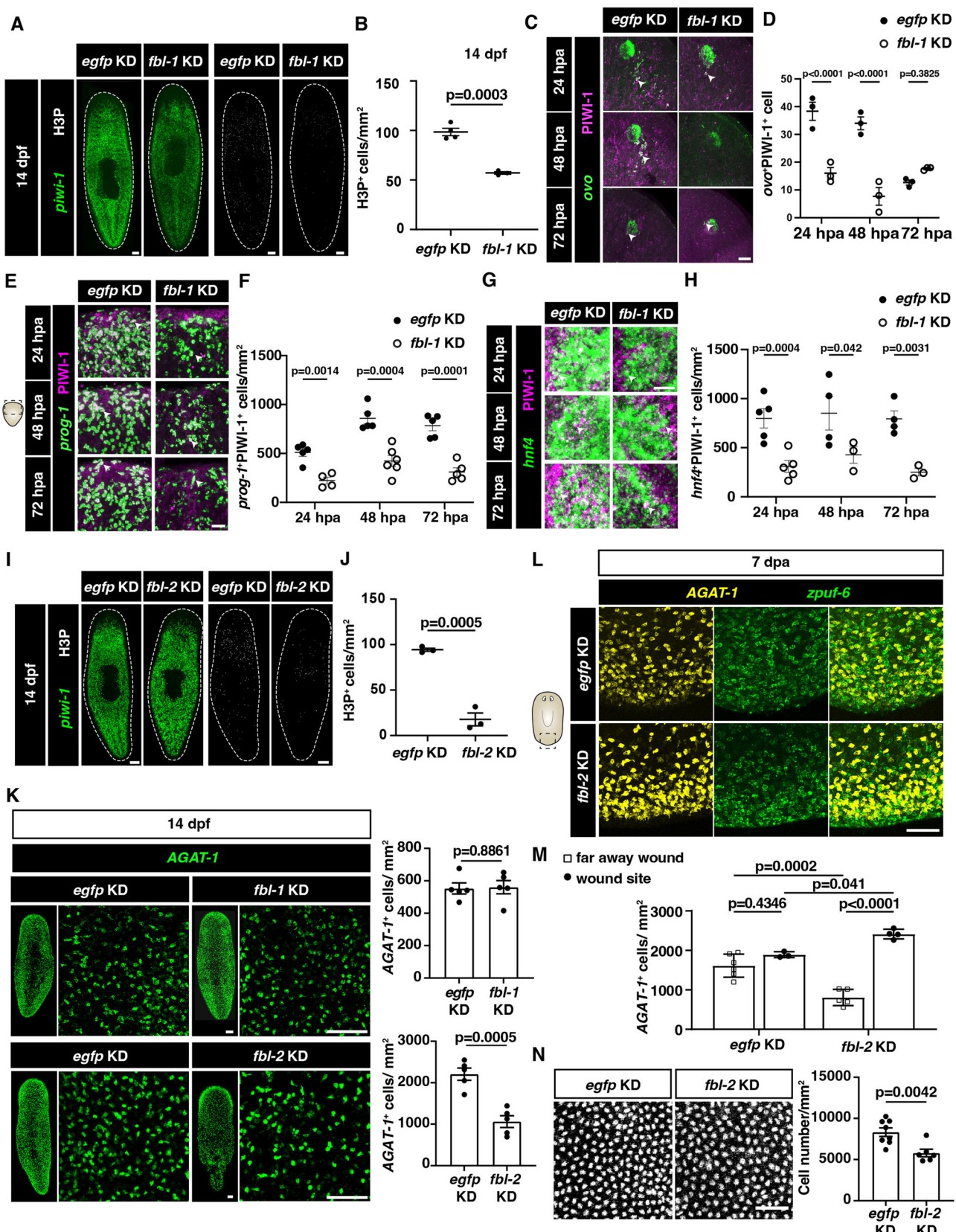

**Figure 3.** *fbl-1* regulates cell proliferation and multiple cell lineage differentiation, and *fbl-2* facilitates the epidermal specification.

(A) FISH images show the stem cells (*piwi-1*+) and proliferating cells (H3P+) at 14 dpf in *egfp* KD control and *fbl-1* KD animals. White dotted lines indicate boundary of animals. Scale bar = 100 μm. (B) Quantification of H3P+ cells in *egfp* KD control (*n* = 4) and *fbl-1* KD (*n* = 3) intact animals at 14 dpf. Each dot represents an individual replicate. Two-tailed unpaired student's *t*-test calculated the *p* value. Data were represented as mean ± SEM. (C) FISH images show KD of *fbl-1* blocks eye progeny (*ovo*+PIWI-1+) differentiation at 24, 48, and 72 hpa compared with that in *egfp* KD controls. White arrowheads indicate double positive cells. Scale bar = 50 μm. (D) Quantification of *ovo*+PIWI-1+ cells indicated as arrowheads in panel C, at 24, 48, and 72 hpa in *egfp* KD control and *fbl-1* KD animals. *n* = 3. Each dot represents an individual replicate. Two-tailed unpaired student's *t*-test calculated the *p* values. Data were represented as mean ± SEM. (E) FISH images show KD of *fbl-1* blocks epidermal progeny (*prog-1*+PIWI-1+) cell differentiation at 24 hpa (*n* = 4), 48 hpa (*n* = 6), and 72 hpa (*n* = 5) compared with that in *egfp* KD controls (*n* = 5). White arrowheads indicate double positive cells. Scale bar = 50 μm. (F) Quantification of *prog-1*+PIWI-1+ cells indicated as arrowheads in panel E, at 24, 48, and 72 hpa in *egfp* KD control and *fbl-1* KD animals. Each dot represents an individual replicate. *n* = 5–6. Two-tailed unpaired student's *t*-test calculated the *p* values. Data were represented as mean ± SEM. (G) FISH images show KD of *fbl-1* blocks intestinal progeny (*hnf4*+PIWI-1+) differentiation at 24, 48, and 72 hpa compared with that in *egfp* KD controls. White arrowheads indicate double positive cells. Scale bar = 50 μm. (H) Quantification of *hnf4*+PIWI-1+ cells indicated as arrowheads in panel G, at 24, 48, and 72 hpa in *egfp* KD control and *fbl-1* KD animals. *n* = 3–5, and each dot indicates an animal. Two-tailed unpaired student's *t*-test calculated the *p* values. Data were represented as mean ± SEM. (I) FISH images show no noticeable reduction of stem cells (*piwi-1*+) but fewer proliferating cells (H3P+) at 14 dpf in *fbl-2* KD animals compared with those in *egfp* KD controls. Scale bar = 100 μm. White dotted lines indicate the boundary of animals. (J) Quantification of H3P+ cells in *egfp* KD control and *fbl-2* KD intact animals at 14 dpf. *n* = 3. Each dot represents an individual replicate. Two-tailed unpaired student's *t*-test calculated the *p* value. Data were represented as mean ± SEM. (K) FISH images and quantification of *AGAT-1*+ cells in *fbl-1* KD and *fbl-2* KD animals at 14 dpf compared with those in *egfp* KD controls. Two-tailed unpaired student's *t*-test calculated the *p* values. *n* = 5. Each dot represents an individual replicate. Scale bar = 100 μm. (L) FISH images show stimulated *AGAT-1*+ and *zpuf-6*+ cells at the posterior poles of *egfp* KD and *fbl-2* KD animals at 7 dpa. Scale bar = 50 μm. (M) Quantification of *AGAT-1*+ cells at far away wound and the wound sites of *egfp* KD and *fbl-2* KD animals at 7 dpa. *n* = 3. Each dot represents an individual replicate. Two-way ANOVA with Sidak's multiple comparisons tests calculated the *p* values. Data were represented as mean ± SEM. (N) DAPI staining and quantification show decreased epidermal cell density after KD of *fbl-2* at 7 dpa. Scale bar = 50 μm. *n* = 5. Each dot represents an individual replicate. Two-tailed unpaired student's *t*-test calculated the *p* values. Data were represented as mean ± SEM. Source data are available online for this figure.

To validate the RiboMeth-seq data and assess the dynamic of Nm in planarians, we designed specific primers to detect the fully methylated sites of 18S rRNA and 28S rRNA identified in planarian by <u>R</u>everse <u>T</u>ranscription at <u>L</u>ow deoxy-ribonucleoside triphosphate concentrations followed by polymerase chain reaction (RTL-PCR). If the detected sites are methylated, the yield of cDNA products from reverse transcription by primers for examination (Pe: RT-U, or FU) should be less than that by primers for control (Pc: RT-A, or FD) at low dNTP condition. Our results showed that the intensity of amplified cDNA products for Am28 and Um1228 in 18S rRNA, and Am937 in 28S rRNA was indeed reduced at the low dNTP condition (Fig. EV4G–J), which are available to be further examined whether they are regulated during homeostasis (intact) and regeneration (24 hpa, 48 hpa, and 72 hpa). While the Nm of Um1228 was detected in both regeneration and homeostasis stages, we observed that the Nm of Am28 in 18S rRNA and Am937 in 28S rRNA were only detected at 48 hpa and 72 hpa but not at homeostasis and 24 hpa, which suggested the dynamics of 2′-O-methylation in planarian regeneration (Fig. EV4K–M). In summary, we confirmed three fully methylated sites in planarian by combining RiboMeth-Seq with RTL-PCR analyses.

We then examined the consequence of *fbl-1* KD and *fbl-2* KD on the methylation levels of the sites identified in planarians (Fig. EV4A–D). The fully methylated sites were displayed in light purple boxes, and partially methylated sites were indicated in light pink boxes (Fig. 4B,C). Compared with *egfp* KD controls, the *fbl-1* KD and *fbl-2* KD groups exhibited reductions in methylation levels at most of these fully methylated sites in the 18S and 28S rRNAs (Fig. 4B,C, light purple boxes and EVA, B), and had no effect on Gm1425 in 18S rRNA and Am312, Gm313 in 28S rRNA (Fig. 4B,C, light purple boxes). Moreover, the Nm of Am755, Gm1258 in 18S rRNA and Am1049 in 28S rRNA were distinctly identified in planarian and presented significantly differential methylation frequency after *fbl-1* KD and *fbl-2* KD (Fig. 4B,C), despite lack of success in validating these methylation sites using RTL-PCR (Fig. EV4G–J; Dataset EV2). All the partially methylated sites depicted in light pink boxes (Fig. 4B,C) exhibited no change after

inhibition of *fbl*. This might be due to subtle alterations that are difficult for bulk RiboMeth-Seq to detect, particularly if the baseline Methscore in the control group is low. Furthermore, future methylation analyses should ideally involve the isolation and examination of different cell types corresponding to the expression patterns of *fbl-1* and *fbl-2*.

Subsequently, we used RTL-PCR assay to validate the methylated rates of Am28, Um1228 in 18S rRNA, and Am937 in 28S rRNA after *fbl* KD (Fig. 4D–I). Compared with *egfp* KD group, the reduced methylation frequency of Um1228 in 18S rRNA caused by *fbl-1* KD and reduced methylation frequency of Am28 in 18S rRNA caused by *fbl-2* KD were also validated (Fig. 4E,G), whereas the Nm of Am937 displayed no significant change following either *fbl-1* KD or *fbl-2* KD (Fig. 4F,I). In summary, we found that the Nm of Um1228 and Am28 could be regulated by *fbl-1* and *fbl-2*, respectively. The Am28 displayed dynamic methylation frequency during planarian regeneration. These findings suggest a requirement of *fbl* on the 2′-O-methylation of rRNA during regeneration.

## *fbl* knockdown affects the distribution of the nucleolar protein but not global protein synthesis in regenerating planarians

Nucleolar formation necessitates the condensation of multiple nucleolar proteins, including FBL, localized at dense fibrillar components, and nucleostemin (NST), localized at granular compartments. To assess the nucleolus integration following *fbl-1* and *fbl-2* KD, we examined the nucleolar structure using the transmission electron microscope. Based on the subcellular structure and cellular distribution in planarian cells, we compared the nucleolar size in stem cells (characterized by heterochromatin and enriched free ribosomes) or epidermal cells (characterized by the granular endoplasmic reticulum and pigment granules) (Ballarin et al, 2021; McGee et al, 1997). However, no significant differences in the morphology of the nucleolus were observed after *fbl-1* and *fbl-2* KD (Fig. EV5A). It has been reported that FBL facilitates the assembly of granular compartments (Yao et al, 2019).

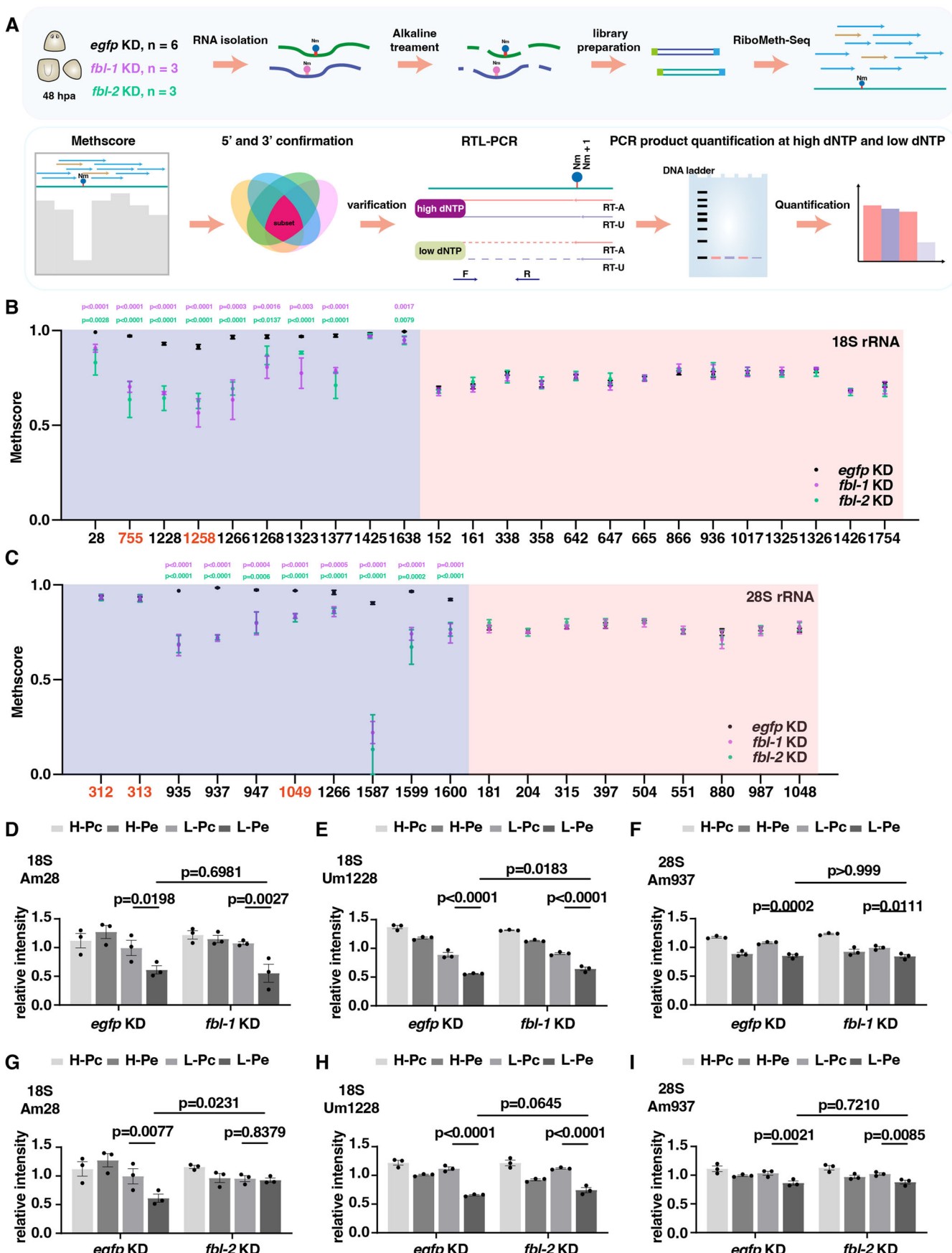

**Figure 4. Knockdown of *fbl-1* and *fbl-2* reduced the 2′-O-methylation level of rRNA.**

(A) Schematic diagram of experimental and analysis processes of RiboMeth-seq. (B, C) Methscore of 2′-O-methylation sites in 18S rRNA and 28S rRNA in *fbl-1* KD and *fbl-2* KD animals, respectively, compared to *egfp* KD controls. Black, purple, and green dots indicate the Methscores of *egfp* KD, *fbl-1* KD and *fbl-2* KD animals. light purple boxes indicate fully methylated sites and light pink boxes indicate partially methylated sites. Methylated sites in red text indicate specific in planarian. Two-tailed unpaired student's *t*-test calculated the *p* values. *egfp* KD, $n = 6$; *fbl-1* KD, $n = 3$; *fbl-2* KD, $n = 3$. Each dot represents an individual replicate. Data were represented as mean ± SEM. (D–F) Analysis of Am28, and Um1228 in 18S rRNA, Am937 in 28S rRNA after *fbl-1* KD. Quantification of PCR products generated with anchored reverse transcription and unanchored reverse transcription primers at different deoxynucleotide triphosphate conditions. Two-way ANOVA with Sidak's multiple comparisons tests calculated the *p* values. $n = 3$. Data were represented as mean ± SEM. Each dot represents an individual replicate. H high dNTP, L low dNTP, Pc primer for control (RT-A anchored reverse transcription primers, FD forward downstream primer); Pe primer for examination (RT-U unanchored reverse transcription primers, FU forward upstream primer). (G–I) Analysis of Am28, and Um1228 in 18S rRNA, Am937 in 28S rRNA after *fbl-2* KD. Quantification of PCR products generated with anchored reverse transcription and unanchored reverse transcription primers at different deoxynucleotide triphosphate conditions. Two-way ANOVA with Sidak's multiple comparisons tests calculated the *p* values. Data were represented as mean ± SEM. $n = 3$. Each dot represents an individual replicate. Source data are available online for this figure.

Therefore, to explore whether *fbl* KD affects the nucleolar components, we performed immunostaining for the NST protein in PIWI-1⁺ cells and *egr-5*⁺ cells in *fbl-1* and *fbl-2* KD animals, respectively. We observed a less aggregated pattern of NST in PIWI-1⁺ cells after *fbl-1* KD (42/91) and in *egr-5*⁺ cells after *fbl-2* KD (12/55) compared with *egfp* KD controls (59/77 and 31/46) (Fig. 5A,B). This is consistent with the phylogenetic analysis of two *fbl* homologs in planarians, suggesting that the evolutionary functions of *fbl-1* similar to those of other animals.

Besides the nuclear localization signal in the GAR domain, FBL is known to modulate pre-rRNA processing in the dense fibrillar compartment. To evaluate whether *fbl-1* and *fbl-2* KD impact pre-rRNA processing, we designed primers for reverse transcription polymerase chain reaction (RT-PCR) to quantify the transcriptional levels of 18S and 28S rRNAs. Following *fbl-1* KD, we observed a reduction in 28S rRNA (Fig. EV5B). The biogenesis of mature rRNA from the 47S pre-rRNA precursor involves two distinct maturation pathways that result in the generation of 18S, 28S, and 5.8S rRNAs. The major pathway proceeds through intermediates 45S pre-RNA to 30S and 32S species, while the minor pathway involves intermediates 45S, 41S, then 21S and 32S species (Fig. 5C). To investigate the impact of *fbl* KD on pre-rRNA processes, northern blot analysis was conducted using probes specific to ITS1 and ITS2 regions (Fig. 5C). Moreover, planarian 28S rRNA is processed into two fragments, α and β, through the removal of a short sequence in the hidden break. After heat denaturation, fragment α has a length similar to that of 18S rRNA (Kim et al, 2019; Natsidis et al, 2019; Sun et al, 2012). To distinguish between the mature forms of 18S rRNA and the 28S rRNA fragments, we have designed specific probes for both (Fig. 5C). The results showed that, compared to *egfp* KD controls, *fbl* KD led to a comparable level of 18S rRNA but a decreased level of mature 28S rRNA (Figs. 5D and EV5C). Furthermore, the maintenance of mature 28S rRNA was differently regulated by two *fbl*, that the fragment β of 28S rRNA was increased after *fbl-1* KD, while an unidentified band was detected after *fbl-2* KD, suggesting aberrant cleavage of the 28S rRNA (Figs. 5D and EV5C). Both processing pathways were affected by *fbl* KD, supported by increased 41S and 26S intermediates after *fbl-1* KD, whereas increased 41S and aberrant 28S intermediates after *fbl-2* KD (Figs. 5D and EV5C). Collectively, these results underscore the critical role of *fbl-1* and *fbl-2* in the coordination and fidelity of rRNA processing pathways in planarians.

We thus attempted to assess the global protein synthesis level using a puromycin labeling assay in intact and regenerating planarians. To evaluate the protein synthesis level, we collected protein samples of intact worms and at 48 hpa with three biological replicates after puromycin treatment with or without cycloheximide (CHX) for 24 h. The protein synthesis level was detected using an anti-puromycin antibody by western blot and normalized to total protein indicated by Ponceau S staining. The puromycin labeling of protein was found to be reduced in the presence of CHX in both the intact and 48 hpa groups. A significant increase in protein synthesis was found during planarian regeneration (48 hpa) compared with homeostasis (intact) (Fig. EV5D,E), underscoring the vital role of protein synthesis regulation during this process. Furthermore, we evaluated the global protein synthesis level following the inhibition of *fbl* during regeneration. Under puromycin and CHX treatment, global protein synthesis was reduced in the control group (*egfp* KD) (Fig. 5E,F). Compared with the control group, global protein synthesis was found to exhibit no significant change after *fbl-1* KD and was slightly increased following *fbl-2* KD at 48 hpa (Fig. 5E,F). Although it is challenging to exclude the possibility of a global translation change in specific cell types, our results suggested that the translation machinery remained functional in the regenerating planarians after the knockdown of *fbl-1* or *fbl-2*. Consequently, we hypothesized that the knockdown of *fbl-1* or *fbl-2* in planarians might regulate the translation of specific mRNA.

### *fbl-1* regulates the translation of genes involved in the RNA splicing process of neoblasts and progenitors

To gain a comprehensive understanding of the role of *fbl* in translational control, we performed Ribo-seq to identify the transcripts being translated, and we evaluated changes in transcription level using RNA-seq at 48 hpa. We obtained the worms at 48 hpa for both Ribo-seq and RNA-seq library construction to minimize the discrepancy. The worm samples were divided into a 25% portion for RNA-seq and a 75% portion for Ribo-seq, with three replicates. Quality control analysis confirmed that the ribosome-protected frame exhibited typical Ribo-seq features, including the expected mapping rates, the depletion of signals from 3′UTRs, and the presence of the characteristic 3-nucleotide (nt) periodicity as described in other Ribo-seq studies (Fig. EV6A,B). The quantile-quantile plot and Spearman correlation analysis indicated that the data between RNA-seq and Ribo-seq of the control group also exhibited a high correlation (Fig. EV6C,D; $R^2 = 0.8262$). Compared the transcriptome and translatome between *fbl-1* KD and *egfp* KD controls, 356 genes

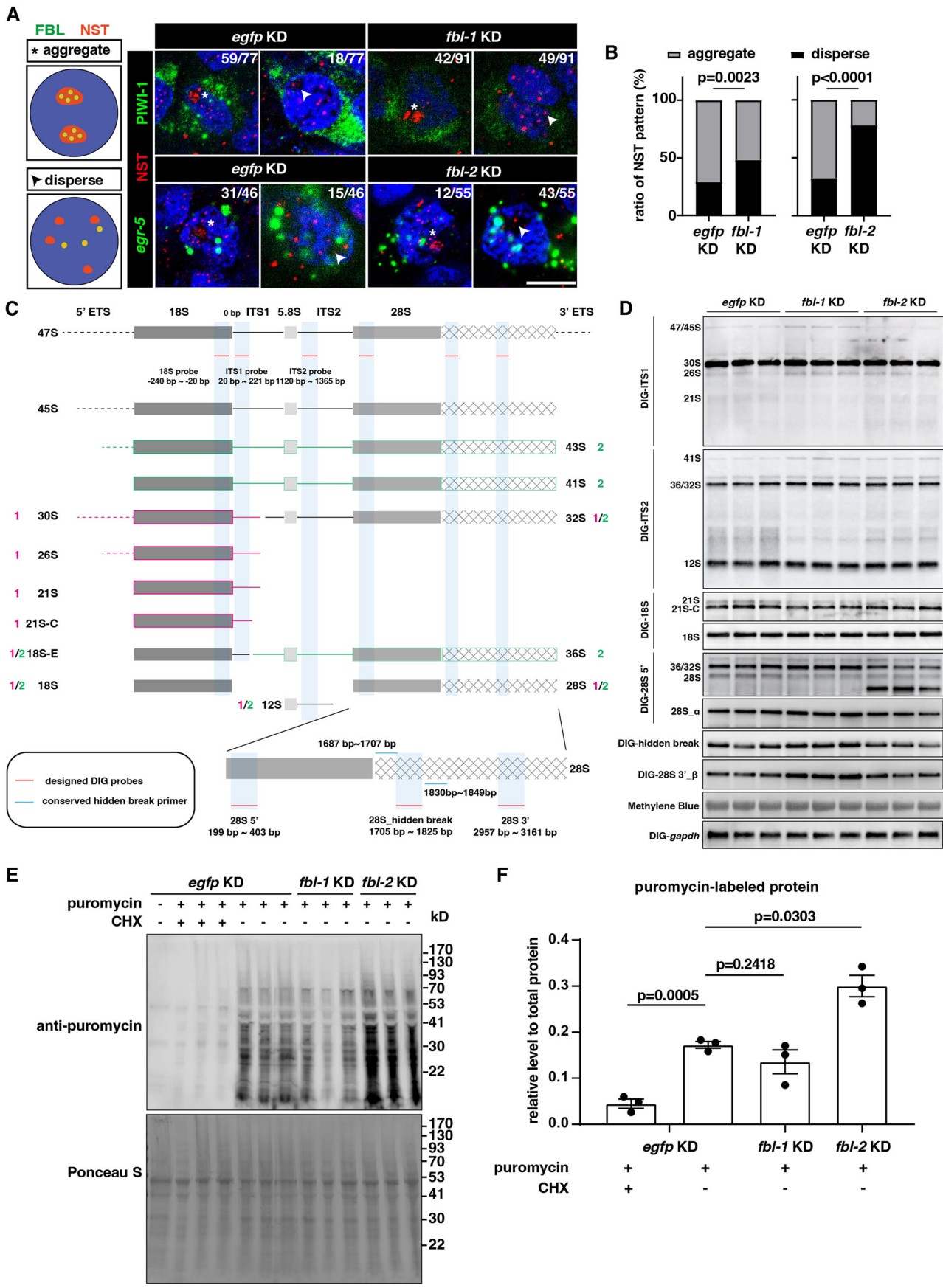

◀

**Figure 5.  Knockdown of *fbl-1* and *fbl-2* caused abnormal distribution of nucleolar protein without changing protein synthesis level.**

(A) Nucleolar protein NST (red) in PIWI-1[+] (green, upper row) and *egr-5*[+] (green, lower row) cells in *egfp* KD, *fbl-1* KD, and *fbl-2* KD animals, respectively. Left panel, diagrams of aggregate (star) and disperse (arrow) patterns of NST signals in cells; Right panel, the numbers indicate the ratio of displayed cases in all the examined cells. Scale bar = 10 μm. (B) Quantification of the ratio of aggregate and dispersed pattern of NST signals in all the examined cells upon *fbl* KD. Fisher's exact test calculated the *p* values. Data were represented as mean ± SEM. (C) Schematic of planarian precursor rRNA intermediates and mature 18S and 28S rRNA with their processing pathways. ETS external transcribed spacer, ITS internal transcribed spacer. The number 1 indicates the major processing pathway, and 2 indicates the minor processing pathway. Red lines indicate the sites of the designed probes. (D) Northern blot analysis of rRNA processing after *fbl* KD. Intermediate rRNAs were detected using DIG-labeled ITS1 and ITS2 probes. Mature 18S and two fragments (α and β) of 28S rRNAs were detected using DIG-labeled 18S and 28S probes, respectively. The loading control *gapdh* was detected using a DIG-labeled probe. (E) Overall protein synthesis rates after *fbl-1* KD and *fbl-2* KD under the indicated treatment. (F) Quantification of protein synthesis rates normalized to total protein after *fbl-1* KD and *fbl-2* KD. *n* = 3. Each dot represents an individual replicate. Two-tailed unpaired student's *t*-test calculated the *p* values. Data were represented as mean ± SEM. Source data are available online for this figure.

were increased and 566 genes were reduced at the transcriptional level, while 1240 transcripts were increased and 862 transcripts were reduced at the translational level (Dataset EV3). Additionally, the expression levels of transcripts enriched in neoblasts and epidermal early progenitors were decreased in *fbl-1* KD compared with *egfp* KD controls (Fig. EV6E), supporting our FISH results. These results suggested that the datasets of Ribo-seq and RNA-seq were reliable for a subsequently integrated analysis.

To assess the influence on translation after *fbl-1* KD, a scatter plot was utilized to show a low correlation between transcription and translation efficiency ($R^2 = 0.1041$), which suggested the specification of regulation in translation by *fbl-1* (Fig. 6A). To determine which transcripts are regulated by *fbl-1* in translation, we identified 652 transcripts that presented a decrease in translation efficiency (TE) without change in mRNA levels (TE down, log₂fold change (FC) $<-0.8$ in TE levels; $-0.8 < \log_2 FC < 0.8$ in mRNA levels) following *fbl-1* KD (Fig. EV6F).

Since the expression of *fbl-1* is enriched in neoblasts, to specifically study the function of *fbl-1* in neoblasts, we further integrated previous single-cell RNA-seq data to categorize genes based on their expression in different cell types (Fincher et al, 2018). Transcripts enriched in the TE down category primarily comprised genes specifically expressed in neoblasts (Fig. EV6G). Following *fbl-1* KD, transcripts in the TE down category that were expressed in neoblasts were strongly associated with the RNA splicing process, DNA replication, and translational termination (Fig. EV6H). To validate the biological processes enriched in GO term, we chose genes with translation at low FDR and high foldchange cutoff (FDR <0.05 and <−0.8 FC), and specific expression in neoblasts for RNAi experiment. KD of *small nuclear ribonucleoprotein polypeptide G* (*snrpG*) and *splicing factor 3b subunit 5* (*sf3b5*) resulted in head regression (Fig. 6B). KD of RNA polymerase II subunit I (*polr2i*) led to delayed planarian regeneration (Fig. 6D). KD of *snrpG*, *sf3b5*, and *polr2i* all caused the reduction of stem cell proliferation as shown by staining of H3P[+] cells (Fig. 6C,E). Based on flow cytometry analysis, planarian cells were broadly categorized as X1, X2, and Xins, in which X1 cells enrich the proliferating stem cells at the S/G2/M cell cycle phases (Hayashi et al, 2006; Reddien et al, 2005). Previous studies have discovered various splicing events in X1 cells, different from those in X2 and Xins cells (Solana et al, 2016). To further validate the dysfunction of the splicing process after *fbl-1* KD, we utilized bulk RNA-seq data and employed rMATS analysis to identify significantly differential splicing processes after *fbl-1* KD (FDR < 0.05) in 15 target genes expressed in neoblasts and progenitors (Solana et al, 2016). The resultant network plot

indicated pathway enrichment associated with DNA replication (Fig. 6F), consistent with the defects in cell proliferation. Notably, the genes *ring finger and WD repeat domain 3* (*rfwd3*) and *fizzy-related protein homolog* (*fzr1*) exhibited increased events in exon skipping (Fig. 6G,H). This resulted in an increase in the second isoform of *rfwd3* and the third isoform of *fzr1* (Fig. EV6I,J). In contrast, *DNA replication helicase/nuclease 2* (*dna2*) and *abnormal spindle-like microcephaly-associated protein* (*aspm*) exhibited decreased events in exon skipping (Figs. 6I,J and EV6K,L), despite *aspm* also being part of the RNA down category. While we cannot rule out that splicing dysfunction may be an indirect consequence of *fbl-1* suppression, our data support previous findings that emphasize the critical role of splicing processes in neoblasts and progenitor cells.

Moreover, we also observed that KD of genes associated with DNA replication, including *chromatin assembly factor 1 subunit A* (*chaf1A*), *cyclin-dependent kinase 1* (*cdk1*), *surfeit locus protein 2* (*surf2*), and *DNA primase subunit 2* (*prim2*), resulted in the failure of planarian regeneration (Fig. EV6M). Suppression of gene *growth arrest and DNA damage-inducible proteins interacting protein 1* (*gadd45gip1*), which was enriched from translational termination, caused regenerative defects at 7 dpa (Fig. EV6M). KD of these five genes enriched from GO terms displayed similar defects observed in *fbl-1* KD animals, which showed reduced proliferation of stem cells (Fig. EV6N). These results suggested that *fbl-1* regulated multiple cellular processes at the translational level.

## The translation of genes to regulate neurotransmitter secretion is essential for epidermal cell development

To next compare the transcriptome and translatome between *fbl-2* KD and *egfp* KD controls, 342 genes were increased, and 305 were reduced at the transcriptional level, accompanied by 1524 increased and 948 reduced transcripts at the translational level (Dataset EV4). Similar to the analysis of *fbl-1* group, there was a low correlation between transcription and translation efficiency ($R^2 = 0.0703$) in the *fbl-2* KD groups (Fig. 7A). To gain a more comprehensive understanding of transcription and translation regulation, we identified 765 genes in TE down in *fbl-2* KD animals (Fig. 7B). We then performed the enrichment analysis of transcripts from TE down category that was enriched in epidermal cells (Fig. 7C). The catecholamine secretion pathway, which involves dopamine, and the cell junction assembly pathway were found to be enriched in *fbl-2* KD animals compared to *egfp* KD controls (Fig. EV7A). Both *synaptotagmin 2* (*syt2*) and *neuronal acetylcholine receptor subunit alpha-6* (*chrna6*) were enriched in the catecholamine secretion

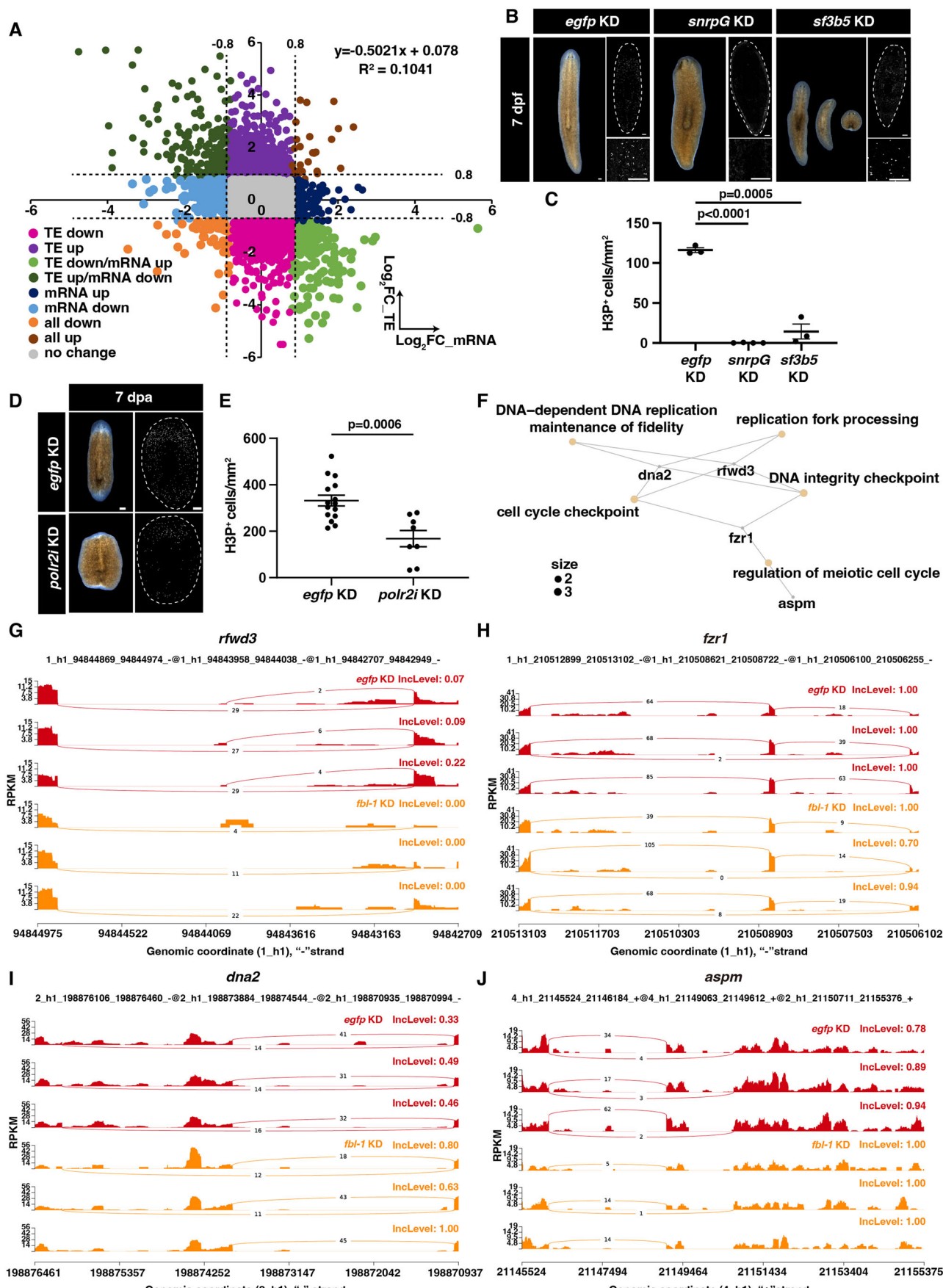

◄ **Figure 6.** *fbl-1* regulates the translation of genes involved in the RNA splicing process in neoblasts and progenitors.

(A) Scatter plot depicts the Log$_2$Fold Change (FC) in total mRNA abundance at 48 hpa on the x-axis and the Log$_2$FC in translational efficiency (TE) on the y-axis upon *fbl-1* KD. The upper right portion of the plot displays the $R^2$ value and the linear formula. (B) Live images and staining of H3P$^+$ cells show defects of *snrpG* KD and *sf3b5* KD animals compared to *egfp* KD controls at 7 dpf. $n = 30$. Scale bar = 200 µm. White dotted lines indicate the boundary of animals. (C) Quantification of H3P$^+$ cells in *egfp* KD ($n = 3$), *snrpG* KD ($n = 4$), and *sf3b5* KD ($n = 3$) animals. Data were represented as mean ± SEM. Each dot represents an individual replicate. Two-tailed unpaired student's *t*-test calculated the *p* values. (D) Live images and staining of H3P$^+$ cells show defects of *polr2i* KD animals compared to *egfp* KD controls at 7 dpa. $n = 30$. Scale bar = 200 µm. White dotted lines indicate the boundary of animals. (E) Quantification of H3P$^+$ cells in *egfp* KD ($n = 15$) and *polr2i* KD ($n = 8$) animals. Data were represented as mean ± SEM. Each dot represents an individual replicate. Two-tailed unpaired student's *t*-test calculated the *p* values. Data were represented as mean ± SEM. (F) Network of genes with altered alternative splicing upon *fbl-1* KD. (G–J) rMATS analysis reveals the changes in alternative skipping exon events of genes, *rfwd3*, *fzr1*, *dna2*, and *aspm*, in *fbl-1* KD animals compared to *egfp* KD controls. Red color represents *egfp* KD group, orange color represents *fbl-1* KD group. Source data are available online for this figure.

pathway. Previous studies identified that *syt2* as a Ca$^{2+}$ sensor in synapse regulates transmitter release (Jamora and Fuchs, 2002; Qian et al, 2016; Xu et al, 2007), and *chrna6* is a nAChR subunit that responds to nicotine (Qian et al, 2016). KD of *syt2* and *chrna6* caused an increase of *AGAT-1$^+$* cells at the posterior of regenerating trunks, similar to the phenotype of *fbl-2* KD animals, in spite of no obvious regeneration defects (Fig. EV7B,C). Given that in planarians, epidermal cells (*AGAT-1$^+$*) express creatine similar to that in neurons and muscle cells (Eisenhoffer et al, 2008; Tu et al, 2015), our results suggested a role of neurotransmitters in the development of epidermis.

Regarding the enriched pathway of cell junction assembly, KD of *angiotensin-converting enzyme* (*ace*) and *lim zinc finger domain containing 1* (*lims1*) resulted in head regression and slightly regenerative defects, respectively (Fig. EV7B,D). During homeostasis, *ace* KD animals showed reduced expression of *AGAT-1$^+$* cells (Fig. EV7E), whereas *AGAT-1$^+$* cells were unaffected under KD of *lims1* (Fig. EV7C). Cell junction relies on the connection of the cytoskeleton to regulate cell migration and signal transduction (Jamora and Fuchs, 2002). These results further provided an explanation that *fbl-2* KD caused the decreased expression of *vim-3*, a marker of cytoskeleton morphogenesis (Fig. EV3F–H).

Since only a limited number of genes enriched in the GO term analysis, we also included genes with low FDR and high foldchange cutoff (FDR <0.05 and <−0.8 FC) in translation, including *growth factor independent 1* (*gfi1*) and *spac17g8.11c* for functional examination. In the human cochlea, *gfi1* is a zinc finger transcription factor that acts as a repressor to regulate the development of hair cells (Jen et al, 2022). *gfi1* KD in planarians resulted in lysis in the dorsal epidermis and a reduction of *AGAT-1$^+$* cells but an increase of *zpuf-6$^+$* cells at the wound sites (Fig. EV7D–F), which suggested a role of *gfi1* in planarian epidermal development. *spac17g8.11c* is a predicted gene of mannosyltransferase, which is required for epidermal integrity during post-embryonic stages in *C. elegans* (Partridge et al, 2008). *spac17g8.11c* KD planarians could regenerate their lost tissues after amputation, but an increase of *AGAT-1$^+$* cells was observed in *spac17g8.11c* KD animals compared to *egfp* KD control animals (Fig. EV7B,C). This result provided evidence of the role of mannosyltransferase in the planarian epidermis. In addition, the GO term analysis of transcripts from TE up identified similar biological processes, such as cilium assembly and transmembrane transport, between *fbl-1* and *fbl-2* KD compared with *egfp* KD controls (Fig. EV7G,H), indicating similarity between *fbl-1* and *fbl-2* in the regulation of planarian tissue homeostasis and regeneration.

## Expression of Wnt pathway components is disturbed in *fbl-2* KD planarians

To investigate any associated defects following *fbl* KD, we performed a GO term analysis of downregulated transcripts in *fbl* KD compared to *egfp* KD controls, disregarding the translation levels. We found a notable enrichment of the Wnt signaling pathway after *fbl-2* KD, but not after *fbl-1* KD (Fig. 7D), despite abnormal distribution of *wnt-1$^+$* cells in both *fbl-1* KD and *fbl-2* KD planarians at 72 hpa (Fig. 1I). The downregulation of transcripts related to the Wnt signaling pathway was confirmed through FISH. As expected, the expression of *frizzled5/8-4* and *frizzle4* in *fbl-2* KD animals at 48 hpa was lower than those in the *egfp* KD controls (Fig. 7E). By performing dFISH of *fbl-2* and *wnt-1* during homeostasis and regeneration, we observed a close spatial correlation between *fbl-2$^+$* and *wnt-1$^+$* cells during regeneration (Fig. 7F). Moreover, the expression of *wnt-1* relative to the distance of the pole was affected as early as 24 hpa, even though no significant change in cell number was observed (Fig. 7G,H).

To gain insights into the regulatory mechanisms of the two *fbl* homologs in planarians, we employed motif prediction. We utilized the JASPAR online tool and uploaded the 2000 base pair sequences flanking the upstream of *fbl-1* and *fbl-2* for motif analysis (Castro-Mondragon et al, 2022). Five motifs were predicted to regulate the expression of *fbl-1* (Fig. EV8A). On the other hand, eleven motifs were highly predicted upstream of *fbl-2*, with forkhead and homeobox binding motifs being the most prevalent (Fig. EV8A). Recent studies have suggested that promoter regions in differentiated cells exhibit an enrichment of adenine (A), in contrast to those in stem cells (Poulet et al, 2023). The putative motifs of the *fbl-2* promoter have a high adenine composition, and the homeobox binding motifs were representative of the CCAAT box. In contrast, this pattern was absent in the motifs of the *fbl-1* promoter (Fig. EV8A). These findings align with previous evidence that promoter regions in epidermal cells contained the CCAAT box (Poulet et al, 2023), thereby supporting the reliability of our motifs analysis.

To validate the transcription factors predicted for *fbl-2*, we conducted RNAi to knock down these genes (Fig. EV8B). Only animals with *smad2-2* KD showed abnormal morphology at the boundary during homeostasis, along with regenerative defects. Additionally, we confirmed the expression patterns of these genes with *fbl-2* (Fig. EV8C). Notably, *homeobox B13* (*hoxB13*), *forkhead box L1* (*foxL1*), and *nuclear receptor subfamily 4 group A member* (*nr4A*) were found to be expressed in *fbl-2$^+$* cells. The nuclear receptor *nr4A* was found to be required for posterior patterning in planarians, as it regulated the expression of tail positional control genes, such as *wnt-1* and *wnt11-2* (Li et al, 2019). To investigate

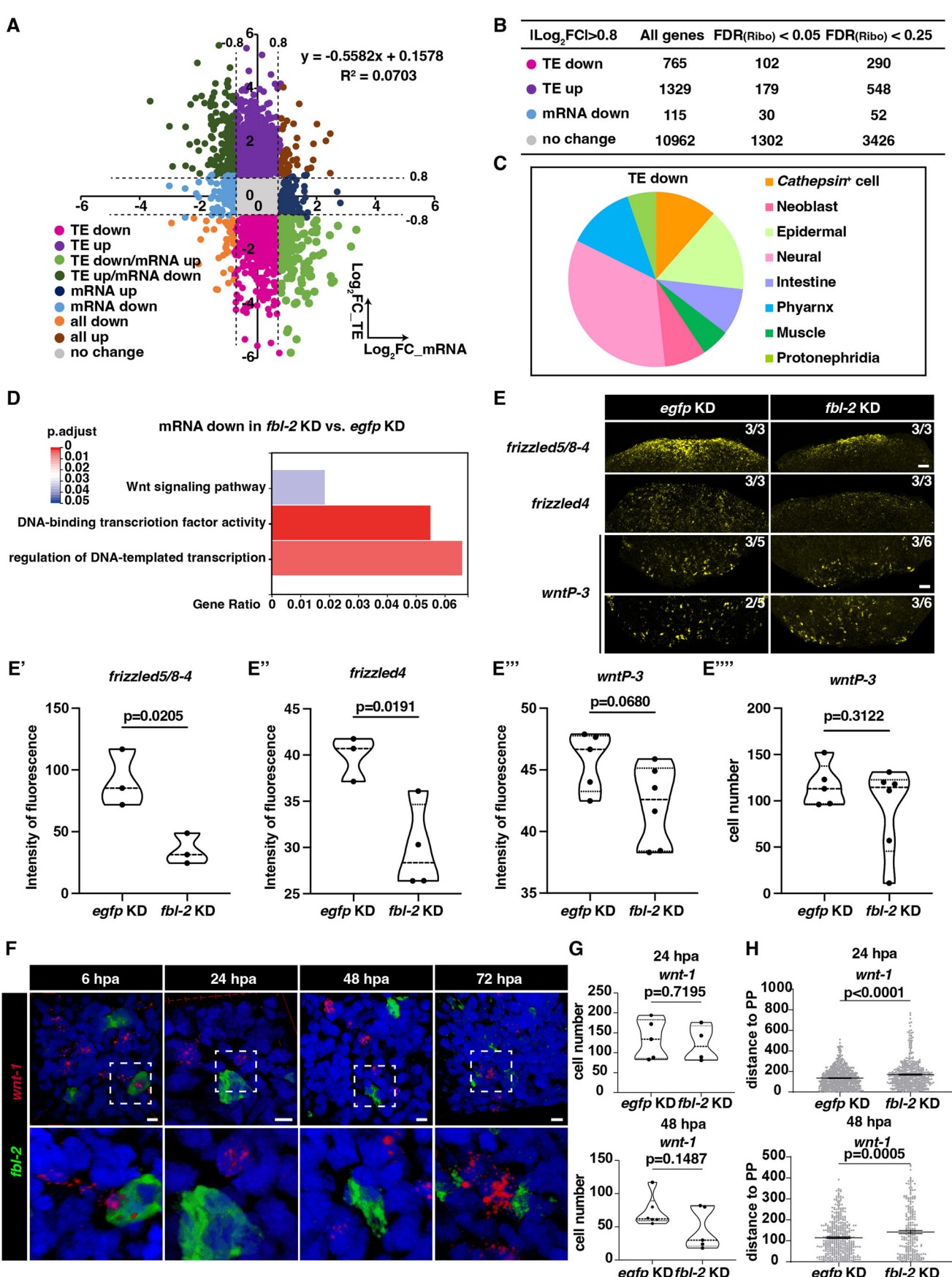

**A** y = -0.5582x + 0.1578, R² = 0.0703

Log₂FC_TE vs Log₂FC_mRNA

- TE down
- TE up
- TE down/mRNA up
- TE up/mRNA down
- mRNA up
- mRNA down
- all down
- all up
- no change

**B**

| \|Log₂FC\|>0.8 | All genes | FDR(Ribo) < 0.05 | FDR(Ribo) < 0.25 |
|---|---|---|---|
| TE down | 765 | 102 | 290 |
| TE up | 1329 | 179 | 548 |
| mRNA down | 115 | 30 | 52 |
| no change | 10962 | 1302 | 3426 |

**C** TE down

- *Cathepsin*⁺ cell
- Neoblast
- Epidermal
- Neural
- Intestine
- Phyarnx
- Muscle
- Protonephridia

**D** mRNA down in *fbl-2* KD vs. *egfp* KD

p.adjust 0 – 0.05

- Wnt signaling pathway
- DNA-binding transcriotion factor activity
- regulation of DNA-templated transcription

Gene Ratio 0 0.01 0.02 0.03 0.04 0.05 0.06

**E**

| | *egfp* KD | *fbl-2* KD |
|---|---|---|
| *frizzled5/8-4* | 3/3 | 3/3 |
| *frizzled4* | 3/3 | 3/3 |
| *wntP-3* | 3/5 | 3/6 |
| | 2/5 | 3/6 |

**E'** *frizzled5/8-4* — Intensity of fluorescence, p=0.0205, *egfp* KD vs *fbl-2* KD

**E''** *frizzled4* — Intensity of fluorescence, p=0.0191, *egfp* KD vs *fbl-2* KD

**E'''** *wntP-3* — Intensity of fluorescence, p=0.0680, *egfp* KD vs *fbl-2* KD

**E''''** *wntP-3* — cell number, p=0.3122, *egfp* KD vs *fbl-2* KD

**F** 6 hpa / 24 hpa / 48 hpa / 72 hpa

*wnt-1* / *fbl-2*

**G** 24 hpa *wnt-1* — cell number, p=0.7195, *egfp* KD *fbl-2* KD

48 hpa *wnt-1* — cell number, p=0.1487, *egfp* KD *fbl-2* KD

**H** 24 hpa *wnt-1* — distance to PP, p<0.0001, *egfp* KD *fbl-2* KD

48 hpa *wnt-1* — distance to PP, p=0.0005, *egfp* KD *fbl-2* KD

**Figure 7.** *fbl-2* was predicted to regulate the expression of *wnt-1*.

(A) Scatter plot depicts the log2fold change (FC) in total mRNA abundance at 48 hpa on the x-axis and the $Log_2FC$ in translational efficiency (TE) on the y-axis upon *fbl-2* KD. The upper right portion of the plot displays the $R^2$ value and the linear formula. (B) Table shows transcripts in each category in A that show a $Log_2FC$ with a cut-off value of ±0.8 in transcription or translational efficiency and false discovery rate (FDR) of translation cut-off values of 0.05 and 0.25. (C) Pie chart shows translational efficiency downregulated mRNA in multiple cell types upon *fbl-2* KD. (D) Bar plot shows the gene ontology of transcriptionally downregulated mRNA-enriched signaling pathway upon *fbl-2* KD. (E) FISH images show genes relative to the Wnt signaling pathway downregulated at 48 hpa upon *fbl-2* KD. The numbers indicate the displayed cases in all the examined samples. Scale bar = 50 μm. (E'-E''') Violin plot of ROI of *frizzled5/8-4*, *frizzled4*, *wntP-3* signals at wound sites. (E'''') Violin plot of the quantification of the number of *wntP-3*+ cells. Each dot represents an individual replicate. n = 3–6. Two-tailed unpaired student's *t*-test calculated the *p* values. Data were represented as mean ± SEM. (F) 3D views show the closed location of *fbl-2*+ and *wnt-1*+ cells at 6, 24, 48, and 72 hpa. The green color indicates *fbl-2* signals, and the red color indicates *wnt-1* signals. The dashed squares indicate the amplified image shown below. n = 4–5. Scale bar = 10 μm. (G) Violin plot of the quantification of the number of *wnt-1*+ cells at the posterior tip in heads of *egfp* and *fbl-2* KD animals at 24 and 48 hpa. n = 4–5. Each dot represents the cell number measured from an individual animal. Two-tailed unpaired student's *t*-test calculated the *p* values. Data were represented as mean ± SEM. (H) Violin plot of the quantification of the distance between *wnt-1*+ cell and the posterior tip in heads of *egfp* and *fbl-2* KD animals at 24 and 48 hpa. n = 4–6. Each dot represents the cell distance measured from an individual *wnt-1*+ cell from *egfp* KD (n = 5) and *fbl-2* KD animals (n = 4) at 24 hpa, and *egfp* KD (n = 6) and *fbl-2* KD animals (n = 5) at 48 hpa. Two-tailed unpaired student's *t*-test calculated the *p* values. Data were represented as mean ± SEM. Source data are available online for this figure.

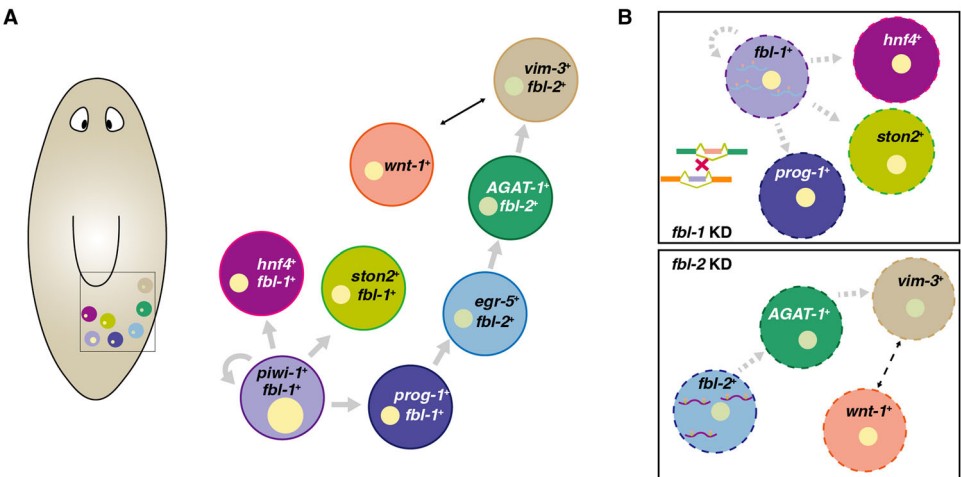

**Figure 8.** Model of regulation of cell differentiation by *fbl* homologs.

(A) *fbl-1* participates in stem cell renewal and progenitor generation, and *fbl-2* regulates the development of epidermal cells. The gray arrows indicate the cell developmental processes (proliferation or differentiation). The black arrow indicates cell-cell communication. (B) Molecular mechanism of regeneration regulation by *fbl*. Splicing processes in neoblasts regulated by *fbl-1* via site-specific rRNA modification. Additional regulation of *fbl-2* expression was relative to *wnt-1* cells and unknown regulation between two *fbls*. During regeneration, KD of *fbl-1* results in disturbances of stem cell proliferation and progenitor generation, while KD of *fbl-2* causes reduced epidermal cell maturation. The gray dashed arrows indicate the reduced cellular process. The black dashed arrow indicates decreased cell-cell communication. The dashed cell membrane indicates a low population of cells.

whether defects in the *wnt-1*+ cells of *fbl-2* KD were accompanied by suppression of these genes, we quantified the number and positional distance from the pole of *notum*+ and *wnt-1*+ cells in KD animals. Our analysis revealed significant defects in the number and distribution of *wnt-1*+ cells after *nr4A* KD, without significant change in *notum*+ cells (Fig. EV8D–F). In addition, *hoxB13* KD animals displayed an abnormal distribution of *notum*+ and *wnt-1*+ cells without significant change in their cell number (Fig. EV8D–F). These findings suggested that the regulation of *fbl-2* in planarians was associated with the transcription factors *nr4A* and *hoxB13* and correlated the body polarity with the Wnt signaling.

## Discussion

Our study discovered two FBL homologs in planarian, *fbl-1* and *fbl-2*, each with unique expression patterns and roles. This observation

suggests that after the duplication event, these two *fbl* genes evolved to carry out different functions in planarian biology. Our data indicates that *fbl-1* is involved in regulating stem cell proliferation and progenitor cell differentiation, whereas *fbl-2* is implicated in late-stage epidermal lineage specification (Fig. 8A). This suggests that the two *fbl* homologs, despite sharing a common ancestral gene, have diverged in their functions and regulate distinct cellular processes. The differential expression of *fbl-1* and *fbl-2* in the early and late stages of epidermal cell differentiation also highlights their unique roles in cell lineage development. In-depth analysis using Ribo-seq and RNA-seq showed that the suppression of *fbl-1* or *fbl-2* caused disturbance of early and late stages of cell development by affecting splicing processes and cell-cell communication (Fig. 8B). This provides evidence that the varying 2'-O-methylation patterns in specific cell types regulate the expression of specific mRNA in planarians. We acknowledge that confirming the alteration of 2'-O-methylation sites on rRNA and its consequent dysregulation of

targeted mRNA translation is challenging in the physiology of complex organisms. However, studying the physiological consequences that link the rRNA modification to translational control in the current study of *fbl-1* and *fbl-2* with cell-type-specificity holds significant importance.

Changes in 2′-*O*-methylation of rRNA have been associated with cancers, different cancer cell lines, and developmental stages of zebrafish and frogs (Delhermite et al, 2022; Khoshnevis et al, 2022; Krogh et al, 2016; Ramachandran et al, 2020). rRNA modification sites installed in ribosome functional sites to regulate translation in human cells, which may be the reason why 2′-O-methylation of rRNA was linked to some pathological diseases or cancer (Dimitrova et al, 2019). More and more evidence indicated the alternation of site-specific rRNA 2′-O-methylation could cause defects in ribosome function, even specific mRNA translation (Häfner et al, 2023). Studies in *C. elegans* suggest that Cm1638 located at the decoding site can form a non-classical base pairing with m⁶A1832 to influence ribosome conformation (Rong et al, 2020; Sloan et al, 2017). In planarians, rRNA 2′-O-methylation sites were first captured by RiboMeth-seq in this study. Our study showed that the methylation frequency varies among individual sites and during regeneration. The differentially methylated frequency is different between *fbl-1* KD and *fbl-2* KD animals. Together with the Ribo-seq analyses, our findings suggest a ribosomal heterogeneity in various planarian cells.

Regeneration requires a precise response to the missing tissue. Previous studies on wound response gene expression have revealed that gene expressions in at least three cell types are activated in response to tissue damage, including neoblasts, epidermal cells, and muscle cells, with cell-type-specificity (Wurtzel et al, 2015). In line with these findings, we found that *fbl-2*⁺ epidermal cells responded to wounds, and the regenerative capacity was impaired after *fbl-2* inhibition. Given the multiple distinct steps of the epidermal lineage development, this discovery may explain why having an additional homolog of *fbl* provides an advantage in efficiently regulating the sequential requirement of ribosome heterogeneity for further differentiation beyond *egr-5*⁺ cells. Additionally, we also observed a reduction in H3P⁺ cells after *fbl-2* suppression, which may support the hypothesis of feedback regulation between epidermis and neoblasts (Tu et al, 2015; Zhu et al, 2015).

The gene duplication has been classified into four major categories to delineate the relationship between evolution and phenotype deriving from duplicated genes (Innan and Kondrashov, 2010). In many organisms, a gene duplication is a common event that can lead to functional diversification, and this appears to be the case with the *fbl* homologs in planarians. Although gene duplication of *fbl* in various species genomes has been recognized, the physiological functions have been primarily focused on one *fbl* in vertebrates (Pereira-Santana et al, 2020). This knowledge gap in our understanding of *fbl* gene duplication, particularly in non-mammalian species, is a significant hindrance to comprehending the full scope of functional diversification that can arise from such events. While it is clear that mammalian *fbl-like-1* has been largely overlooked due to its low expression levels (Pereira-Santana et al, 2020), it may still play crucial, yet subtle, roles in cellular functions and organismal physiology. Underlining the significance of *fbl* duplication throughout the animal kingdoms, studies on *fbl* homologs in planarians gave significant hints to study the other copy of *fbl* in mammals.

In conclusion, our study illuminates how gene duplication and subsequent functional diversification of the two *fbl* genes contribute to the complex biological processes in planarians. Further studies are required to understand how rRNA site modification is regulated by different *fbl* and how the mRNA translation by cell-type-specific ribosomes is achieved for cell fate decisions. Besides, techniques to identify and explore the function of snoRNAs in planarians need to be developed.

# Methods

**Reagents and tools table**

| Reagent/resource | Reference or source | Identifier or catalog number |
| --- | --- | --- |
| **Experimental models** | | |
| Asexual S. mediterranea (strain CIW4) | Sánchez Alvarado laboratory | |
| **Antibodies** | | |
| Anti-Digoxigenin-POD | Roche | 11207733910 |
| Anti-Digoxigenin-AP | Roche | 11093274910 |
| Anti-Fluorescein-POD | Roche | 11426346910 |
| Anti-phospho-Histone H3 (Ser10) (H3P) antibody | Abcam | ab32107 |
| Anti-Smed-PIWI-1 antibody | Gift from Dr. Jochen Rink | |
| Anti-NST antibody | Gift from Dr. Peter Reddien | |
| Anti-puromycin antibody (clone 12D10) | Millipore | MABE343 |
| Alexa 555-conjugated goat anti-rabbit antibody | Abcam | ab150086 |
| Peroxidase (HRP)-conjugated goat anti-mouse antibody | Immunoway | RS0001 |
| **Oligonucleotides and other sequence-based reagents** | | |
| *Escherichia coli* DH5a | | |
| *Escherichia coli* HT115 | | |
| Primers | This study | Dataset EV2 |
| **Chemicals, enzymes, and other reagents** | | |
| Formaldehyde | Sigma | F8775 |
| RIPA | Genstar | E122-01 |
| IGEPAL (NP40) | Sigma | I3021 |
| Tween-20 | Sigma | P1379-100ML |
| Triton X-100 | Sigma | T8787 |
| Proteinase K | Invitrogen | 25530049 |
| 50×Dehardts | Thermo Scientific | D2532 |
| DTT (Dithiothreitol) | Thermo Scientific | R0862 |
| Deionized Formamide | Thermo Scientific | AM9344 |
| Formamide | Sangon | A600212 |
| Heparin | Sigma | H3149 |
| Torula Yeast RNA | Sigma | R6625 |
| Western Blocking Reagent | Roche | 11921681001 |

| Reagent/resource | Reference or source | Identifier or catalog number |
|---|---|---|
| Horse Serum | Hyclone | N/A |
| RNAse-Free H$_2$O | Invitrogen | AM9937 |
| TRIzol reagent | Invitrogen | 10296010 |
| Cycloheximide (CHX) | MCE | HY-12320 |
| Ponceau S | Biosharp | BL519A |
| Dextran Sulfate | Sangon | A600160 |
| Maleic acid | aladdin | M108866 |
| *N*-acetyl-L-cysteine (NAC) | Sigma | A7250 |
| NBT | Roche | 11383213001 |
| BCIP | Roche | 11383221001 |
| Tyramide-conjugated Cy3 | Sigma-Aldrich | PA13101 |
| Tyramide-conjugated Cy5 | Sigma-Aldrich | PA15101 |
| DAPI | Thermo Fisher Scientific | D3306 |
| Phusion High-Fidelity DNA Polymerase | NEB | M0530L |
| T7 RNA Polymerase | Promega | P207E |
| RNase A | Thermo Scientific | EN0531 |
| TURBO™ DNase | Invitrogen | AM2238 |
| RNasin ribonuclease inhibitor | Promega | N2515 |
| M-MLV reverse transcriptase | Promega | M1705 |
| DIG RNA Labeling Mix | Roche | 11277073910 |
| Fluorescein RNA Labeling Mix | Roche | 11685619910 |
| SurePAGE 4–20% Bis-Tris gel | GenScript | M00657 |
| MOPS running buffer | GenScript | M00138 |
| BioTraceNTnitrCellulose | Cytiva | 66485 |
| **Software** | | |
| Fiji | | v2.3.0 |
| Adobe Photoshop 2024 | | |
| Adobe Illustrator 2024 | | |
| **Other** | | |
| LSM 900 with Airyscan | Zeiss | |
| Nikon C2Si | Nikon | |
| Leica M205 FA fluorescence stereomicroscope | Leica | |
| RNA Clean & Concentrator-25 kit | Zymo Research | R1017 |
| VAHTS Total RNA-seq (H/M/R) Library Prep Kit | Vazyme | NR603 |

## Animal maintenance and irradiation

The planarian *Schmidtea mediterranea* CIW4 strain was cultured in 1× Montjuïc water at 20 °C. Prior to any experiments, the animals were starved for 7–10 days. For irradiation experiments, planarians of 4–5 mm were exposed to 100 Gy of radiation dose using an RS2000 Pro X-ray irradiation apparatus.

## RNA extraction, gene cloning, and expression analysis

Total RNA was extracted with TRIzol (Invitrogen, 10296010) and subsequently converted into cDNA utilizing ABScript II-RT Mix for qPCR with gDNA Remover (ABclonal, RK20403). This cDNA was then used for gene cloning. Primer sequences used in this study are provided in Dataset EV2. Primers for gene cloning were designed based on the transcriptome database available at https://planosphere.stowers.org/, while primers for qPCR were designed using an online tool found at https://simrbase.stowers.org/cgi-bin/primer3.pl. The expression of genes of interest was calculated relative to the expression of *tubulin*. The specific primers used were as follows: *fbl-1*: 5′ TTTCCAGCATCAGGTGAAAG and 3′ ATCTCCAAATCCTCCCCTAC; *fbl-2*: 5′ AAGGCCTCTTGTATC-GATTC and 3′ CCATTGACAGCCAAGATTTC; 18S rRNA: 5′ AACGGCTACCACATCC and 3′ ACCAGACTTGCCCTCC; and 28S rRNA: 5′ CGGATTGTTTGAGAATGCA and 3′ CAAAGTTCTTTTCAACTTTCCC. *tubulin*: 5′ TGGCTGCTTG TGATCCAAGA and 3′ AAATTGCCGCAACAGTCAAATA. The genes were cloned into the pT4P vector which was used for riboprobe synthesis and the preparation of RNA interference food. Three biological repeats were collected from each group for RNA extraction and gene expression analysis.

## Phylogenetic analysis

A search for homologs of the human FBL sequence in the planarian *Schmidtea mediterranea* was conducted using protein sequence alignment. The protein sequences of the FBL homologs were sourced from the websites https://www.uniprot.org/, https://planosphere.stowers.org/ and referred to a previous study (Pereira-Santana et al, 2020). MEGA 6 (Tamura et al, 2013) was used to generate the FASTA files of these sequences. All the protein sequences were aligned using the online tool MAFFT version 7 with the L-INS-i method (https://mafft.cbrc.jp/alignment/server/index.html) (Katoh et al, 2019; Kuraku et al, 2013). The aligned sequences were then trimmed using Gblocks 0.91b. Maximum likelihood analyses were performed using the online web server IQ TREE (http://iqtree.cibiv.univie.ac.at/) (Hoang et al, 2018; Nguyen et al, 2015). This analysis involved 1000 ultrafast bootstrap replicates, the application of LG + G4 model, the estimation of four substitution rate categories, and the calculation of the proportion of invariable sites based on the dataset. Finally, the phylogenetic tree resulting from these analyses was visualized using MEGA 6.

## RNAi feedings

pT4P with the genes of interest was transferred into the *E. coli* HT115 strain, and a single colony was cultured for 16 h as a starter culture in 2× YT medium. The starter culture was allowed to grow tenfold before induction with 1 mM IPTG for two hours until the OD600 was ~0.8. The 4× RNAi food for KD of *fbl-1* was prepared by mixing 50 mL of cultured bacteria with 125 μL of liver homogenate (90% liver paste, 9% 1× Montjuic water, and 1% red food coloring). The 2× concentration of RNAi food was prepared for KD of *fbl-2*. The feeding schedule was arranged according to practical necessity (to produce the desired phenotype but no

lethality). The *fbl-1* KD and *fbl-2* KD planarians were fed five every three days, and the samples were collected after 7-day starvation.

## WISH and FISH

Planarians were killed using 5% *N*-acetylcysteine (NAC) solution for 5 min, followed by fixation in 4% formaldehyde (FA) for 45 min. This was proceeded by reduction and dehydration. The worms were then incubated in 100% methanol for 2 h, after which they were prepared for WISH and FISH experiments. The worms were hybridized with riboprobe in Hybe (50% (v/v) Deionized Formamide (Thermo Scientific, AM9344), 5× Saline-sodium citrate (SSC), 1× Denhardt's Solution (Sigma; D2532), 100 µg/mL Heparin from porcine intestine (Sigma; H3149), 0.25 mg/mL yeast torula RNA (Sigma; R6625), 1% (v/v) Tween-20 (sigma, P1379-100ML), and RNAse-Free H$_2$O (Invitrogen; AM9937)) for up to 16 h at 56 °C after being bleached in 5% formamide in 0.5× SSC under direct light for 2 h, and then permeabilized with proteinase K for 10 min. For WISH experiments, the worms were incubated with anti-DIG conjugated with Alkaline Phosphatase (AP) at 4 °C overnight. The color was subsequently developed using NBT and BCIP. For FISH experiments, the worms were incubated with anti-DIG/DNP/FL antibodies conjugated with Peroxidase (POD) or Horseradish Peroxidase (HRP) at 4 °C overnight. The fluorescent signal was developed by incubating the worms with tyramine labeled with a fluorescence tag at room temperature for an hour.

## Antibody staining

The anti-H3P (Abcam, ab32107) antibody staining was performed after FISH. The anti-H3P was used at 1: 1000 dilution in PBSTx0.5. The anti-Smed-PIWI-1 antibody was a gift from Dr. Jochen Rink and was used at a 1:10,000 dilution. The anti-NST antibody was a gift from Dr. Peter Reddien and was used at a 1:2000 dilution.

## Library preparation of RNA-seq and Ribo-seq

The experiments and analysis referred to previous studies (Luan et al, 2022). Thirty worms for one replicated group were collected at 48 hpa before being treated with cycloheximide (CHX, final concentration 0.1 mg/mL, MCE, HY-12320) for 10 min and washed with 1× PBS twice. The samples were covered by RNA Keeper Tissue Stabilizer (Vazyme, R501) and frozen by liquid nitrogen. Frozen samples were lysed with 600 µL Polysome buffer (20 mM Tris, pH 7.4,150 mM NaCl, 5 mM MgCl$_2$,1 mM DTT, 1% Triton X-100), fresh added fresh 1 U/µL DNase I (NEB, M0303S), 50 mg/ml CHX. For each sample, the lysate was divided into two aliquots (450 µL for Ribo-seq and 150 µL for RNA-seq). For Ribo-seq, the lysates were treated with 5 µg/ml of RNase A (Thermo Scientific, EN0531) for each A260 of lysate (Thermo Scientific, EN0531) for 45 min at room temperature. Nuclease digestion was stopped by adding 10 µL of SUPERase·In RNase Inhibitor (Ambion, AM2696). Lysates were loaded onto MicroSpin S-400 HR spin columns (GE Healthcare, 27-5140-01) and purified using RNA Clean & Concentrator-25 kit (Zymo Research, R1017). Then the rRNA was depleted with 5'Biotin DNA probes and separated on a 10% TBE–urea polyacrylamide gel. Ribosome-protected fragments with lengths between 30 nt and 40 nt were selected to construct libraries. The RPF library was constructed by End Repair,

linker ligation, Reverse Transcription, Circularize the cDNA, and PCR amplification. Finally, 8% PAGE to recover the desired products ranging from 140 to 160 bp amplicons. For RNA-seq, total RNA was purified with a Zymo RNA Clean & Concentrator-25 Kit (Zymo Research). rRNA was depleted with 5'Biotin DNA probes, and libraries were constructed with a VAHTS Total RNA-seq (H/M/R) Library Prep Kit (Vazyme, NR603) according to the manufacturer's instructions. All libraries were subjected to PE150 sequencing in an Illumina HiSeq ×10 or a NovaSeq 6000 system.

## Data processing

The raw sequence reads were demultiplexed using CASAVA (v1.8.2), and the 3′-end adapter was clipped using Cutadapt (v1.8.1) (with parameters "-a AGATCGGAAGAGCACACGTCTGAACTCCAG TCA -match-read-wildcards -m 6"). Low-quality sequences were trimmed using Sickle (v1.33) (with parameters "-q 20"). The trimmed reads were filtered by length to retain reads in the range [20, 50] bp. Reads mapping to *Schmidtea mediterranea* rRNA references were removed, and the remaining clean reads were aligned to the *Schmidtea mediterranea* reference transcriptome smed_2010614, downloaded from https://planosphere.stowers.org/ using Tophat2 (v2.0.14) (Kim et al, 2013) with the following parameters: "tophat2 -g 20 -N 2 --transcriptome-index [index file] -G [gtf file] [fastq file] -o [output directory]". Only uniquely mapped reads were extracted for subsequent gene expression analysis. Read counts per gene were obtained using the R package featureCounts (v1.6.2) [PMID: 30783653], and then converted to transcripts per kilobase million (TPM).

## Differential expression analysis

Differentially expressed genes were identified by integrating Ribo-seq and RNA-seq read count data using the deltaTE method (Chothani et al, 2019). The deltaTE algorithm models the proportional changes in ribosome footprints versus RNA levels to detect differences in translational efficiency between conditions. Genes with FDR <0.05 were considered significantly differentially transcribed or translated between conditions.

## RiboMeth-Seq and analysis

RNA (3 µg) extracted from 48 hpa samples was used to quantify 2′-*O*-methylated residues, following methodologies described in previous studies (Ayadi et al, 2018; Marchand et al, 2016). The STAR software (version 2.7.9a) was used to create a reference genome for the 18S and 28S rRNA of the planarian *Schmidtea mediterranea*. The sequencing data were then processed for adapter trimming using fastp (version subread/2.0.2), which effectively removed most sequencing adapters. Subsequently, the trimmed sequencing data from twelve sets were aligned to the constructed reference genome using STAR, generating *.bam files. Next, the bedtools suite (version 2.30.0) was used to compute the counts for each aligned locus within these *.bam files, thereby obtaining the counts for every locus. To calculate the Methscore for each locus, the count value at a given locus "j" was compared to the average count value of the six preceding and six succeeding loci. This iterative process was performed for each locus to derive its respective Methscore, following a published method

(Birkedal et al, 2015). This calculation enabled the quantification of local methylation status across the genomic loci.

## RTL-PCR

For the detection of specific 2′-*O*-methylated sites in planarian, RTL-PCR was performed according to a method described previously (Barros-Silva et al, 2023). RNA was extracted from wild-type samples during homeostasis (intact) and regeneration at 24, 48, and 72 hpa, as well as from *egfp* KD, *fbl-1* KD, and *fbl-2* KD at 48 hpa using TRIzol (Invitrogen, 10296010). Genome DNA was removed using the TURBO™ DNase (Invitrogen, AM2238). 40 ng RNA with 1 µM RT-primers (RT-A and RT-U, or RT of Am28, Dataset EV2) was mixed to be annealed at 70 °C for 10 min and then chilled on ice. For the RT step, a 20 µL mixture containing 200 U M-MLV reverse transcriptase (Promega, M1705) was prepared with a low (2 µM) and high (1 mM) concentration of dNTPs, respectively. The RT reaction was incubated at 42 °C for 1 h and subsequently heated at 70 °C for 15 min to deactivate the reverse transcriptase. About 2 µL cDNA was thereafter amplified using PCR with a mixture of specific PCR primers (Dataset EV2) and 2× PCR mix (ABclonal, RK20719). The PCR reactions were performed as follows: one cycle at 94 °C for 3 min, followed by 40 cycles at 94 °C for 30 s, 54 °C for 30 s, and 72 °C for 20 s. The PCR products were then equally loaded and separated on 1% agarose gels, stained with YeaRed dye (Yeasen, 10203ES76), and visualized using UV trans-illumination. The gel images were captured by a gel image system (Tanon 2500) and quantified using ImageJ. The bands of each group were selected using the rectangle tool in the gels analysis feature. The images were inverted to display black bands on a white background. The area of each peak was measured to quantify the density of the bands. The relative intensity of the bands was normalized for each condition of one Nm detection.

## Northern blot

Total RNA was extracted from planarian samples using TRIzol reagents, as mentioned above. Digoxigenin (DIG)-labeled antisense probes for ITS1, ITS2, 18S, 28S rRNA, and *gapdh* were synthesized using specific primers based on the rDNA sequences in planarians (Dataset EV2) (Carranza et al, 1999). For gel electrophoresis, 0.04–4 µg RNA was loaded on a 1% formaldehyde agarose gel and run at 65 V for 4.5 h. Subsequently, the RNA was transferred onto a nylon membrane (Amersham Hybond-N+, RPN303B) for 16–22 h. The transferred RNA was cross-linked to the membrane using a UVP cross-linker (Analytik Jena, CL1000) at a setting of $4000 \times 100$ µJ/cm². The membrane was incubated with DIG-labeled probes (1:2000 dilution) in the hybridization solution (6× SSC, 0.5% SDS, 100 µg/mL Salmon DNA, 50% Deionized formamide, 5× Denhardt's solution) overnight after prehybridization at 56 °C for 1 h. Detection of DIG-labeled probes was achieved by incubating the membrane with anti-DIG antibody conjugated with Peroxidase POD (1:2000 dilution) overnight at 4 °C or 2 h at RT, following visualization of the POD signal by ECL (NCM Biotech, P10300). The membranes were imaged using the ImageQuant 800 system (cytiva). The 0.03% methylene blue in 0.3 M sodium acetate buffer (pH 5.2) was used to stain 18S and 28S rRNA. The intensity of bands was quantified using ImageJ.

## Puromycin labeling and immunoblot

Planarians were incubated with 0.5 mg/mL puromycin for 24 h before sample collection. The control groups either received no puromycin treatment or were treated with puromycin supplemented with 0.1 mg/mL CHX (MCE, HY-12320). Each group contained 15 worms with three biological repeats. For puromycin labeling analysis at 48 hpa, the incubation started at 24 hpa and samples were collected at 48 hpa. The worms were dissected in lysis buffer composed of RIPA buffer, proteinase inhibitor, PMSF, phosphatase inhibitor, and 1 mM DTT. The resulting mixture was ground and incubated on ice for 15 min. For gel electrophoresis, an equal amount of total protein was separated by 4–20% gel (GenScript, M00657). After 1-h blocking in 5% skim milk at RT, the membrane was incubated with an anti-puromycin antibody (clone 12D10, Millipore, MABE343) diluted 1:5000 in blocking buffer overnight at 4 °C. After washed with TBSTw 0.05%, the membrane was incubated in anti-mouse HRP conjugated antibody diluted 1:20,000 overnight at 4 °C. ECL (NCM Biotech, P10300) was used to detect the signal of puromycin-conjugated protein. Ponceau S (Biosharp, BL519A) was used for total protein detection. The images were quantified using ImageJ. The lanes of each group were selected by rectangle using gel analysis after background subtraction. The profile plot was used to represent the relative density of the contents of the rectangle was measured to quantify the density of the detected protein.

## rMATS

RNA-Seq reads from *fbl-1* KD and *egfp* KD control groups were mapped to the *Schmidtea mediterranea* genome (PRJNA885486). The software rMATS v4.1.2 was used to identify the splicing events. For visualization of these splicing events, the rmats2sashimiplot was used.

## Image acquisition and analysis

WISH samples and live worms were imaged on a Leica M205 FA fluorescence stereomicroscope equipped with a 0.63× objective (N.A. = 0.35). FISH samples were imaged on Nikon C2Si and ZEISS LSM900 inverted confocal microscopes. The objectives, including 10× objective (N.A. = 0.3), 20× objective (N.A. = 0.8), and 63× objective (N.A. = 1.4), were used to capture images. Images were processed using software Fiji v2.3.0, Adobe Photoshop 2024, and Adobe Illustrator 2024.

## Statistics

Statistical analysis, including the calculation of mean values and standard errors of the mean (SEM), was conducted using the software GraphPad Prism 10. The calculation of $p$ values is mentioned in the figure legends.

# Data availability

The code in this paper is available at https://github.com/leilabteam/cjjprojectfbl. The raw sequencing data in this paper have been

deposited and available in the Gene Expression Omnibus (GEO) with accession number GSE255595.

The source data of this paper are collected in the following database record: biostudies:S-SCDT-10_1038-S44318-024-00315-x.

# Peer review information

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

## Acknowledgements

We express our gratitude to Xiaochun Yu for the helpful discussion and help, Peter Reddien for sharing the anti-NST antibody, Jochen Rink for sharing the anti-PIWI-1 antibody, Weike Pei, Pengfei Xu, Feng He, and Peng Li for their technical help, and all the members of the Lei lab for the stimulating discussion. We acknowledge all members of the High-Performance Computing Center (Ling Yang, Nan Li), Flow Cytometry, and Microscopy Core Facilities at Westlake University for technical support. K.L. was supported by the National Natural Science Foundation of China (32470878, 32122032, and 31970750), the Zhejiang Provincial Key Laboratory Construction Project, and the start-up fund from the Westlake Education Foundation.

## Author contributions

**Jiajia Chen**: Conceptualization; Data curation; Formal analysis; Validation; Investigation; Visualization; Methodology; Writing—original draft; Project administration; Writing—review and editing. **Yucong Li**: Data curation; Visualization; Methodology; Writing—original draft. **Yan Wang**: Data curation; Investigation; Methodology. **Hui Wang**: Investigation; Methodology. **Jiaqi Yang**: Investigation; Methodology. **Xue Pan**: Investigation; Methodology. **Yun Zhao**: Validation; Investigation. **Hao Xu**: Validation; Investigation. **Penglei Jiang**: Methodology. **Pengxu Qian**: Supervision. **Hongwei Wang**: Data curation; Formal analysis; Methodology. **Zhi Xie**: Supervision. **Kai Lei**: Conceptualization; Resources; Data curation; Formal analysis; Supervision; Funding acquisition; Validation; Investigation; Methodology; Project administration; Writing—review and editing.

Source data underlying figure panels in this paper may have individual authorship assigned. Where available, figure panel/source data authorship is listed in the following database record: biostudies:S-SCDT-10_1038-S44318-024-00315-x.

## Disclosure and competing interests statement

The authors declare no competing interests.

# Expanded View Figures

**Figure EV1. Identification of *fbl-1* and *fbl-2* and their function in planarians.** ▶

(**A**) Phylogenetic tree of fbl and fbl-like 1 protein homologs from 14 species. The posterior probability value is shown on each branch. *Homo sapiens*, Hs; *Rattus norvegicus*, Rn; *Mus musculus*, Mm; *Macaca fascicularis*, Mf; *Xenopus laevis*, Xl; *Ambystoma mexicanum*, Am; *Danio rerio*, Dr; *Schmidtea mediterranea*, Smed; *Biomphalaria glabrata*, Bg; *Octopus bimaculoides*, Ob; *Caenorhabditis elegans*, Ce; *Hofstenia miamia*, Hm; *Drosophila melanogaster*, Dm; *Hydra vulgaris*, Hv. Proteins of human FBL, planarian FBL-1 and FBL-2 are illustrated with glycine-arginine-rich (GAR) and methyltransferase (MTase) protein domains. (**B**) Sequence alignment of protein homologs shows the residue conservation on the GAR domain among 14 species (Robert and Gouet, 2014). (**C**) Relative mRNA level of *fbl-1* and *fbl-2* in *fbl-1* KD animals measured by quantitative PCR and whole-mount in situ hybridization (WISH). $n = 3$ replicates in each of the two independent experiments. Each dot represents an individual replicate. Two-tailed unpaired student's *t*-test calculated the *p* values. Data were represented as mean ± SEM. Scale bar = 500 μm. (**D**) Relative mRNA level of *fbl-1* and *fbl-2* in *fbl-2* KD animals measured by quantitative PCR and WISH. $n = 3$ replicates in each of the two independent experiments. Each dot represents an individual replicate. Two-tailed unpaired student's *t*-test calculated the *p* values. Data were represented as mean ± SEM. Scale bar = 500 μm. (**E**) Dual FISH of neurons (*pc2*, green) and intestinal tissue (*smed30004557*, magenta) in trunks of *egfp* KD, *fbl-1* KD, and *fbl-2* KD animals at 7 dpa. $n = 6$. Scale bar = 200 μm. (**F**) Dot plot shows the ratio of brain length to body length in trunks of *egfp* KD, *fbl-1* KD, and *fbl-2* KD animals at 7 dpa. Each dot represents the ratio measured from an individual animal. $n = 6$. Data were represented as mean ± SEM. Two-tailed unpaired student's *t*-test calculated the *p* values.

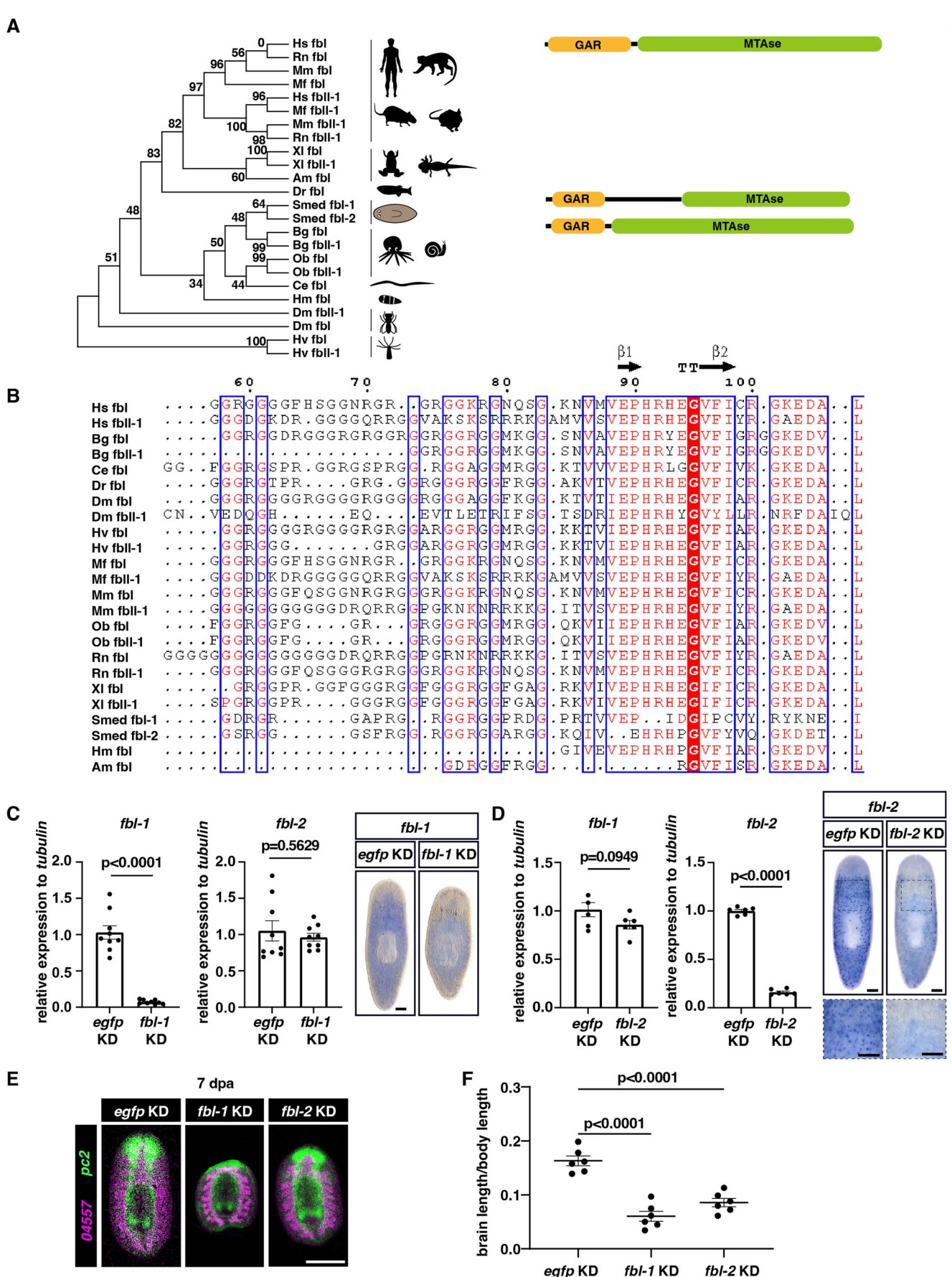

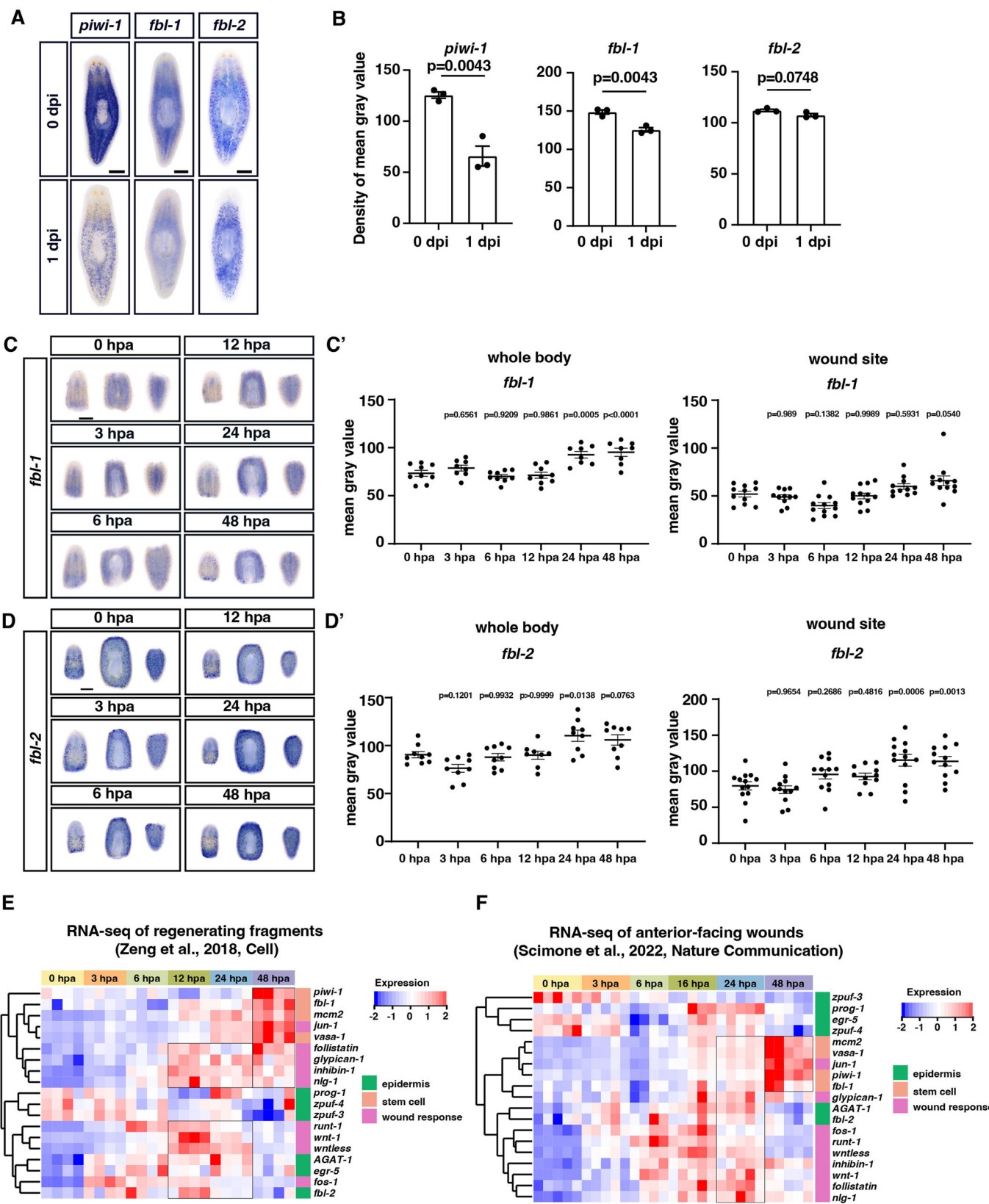

**Figure EV2.  *fbl-1* and *fbl-2* expression in homeostasis and regeneration.**

(**A**) Colorimetric WISH images show staining with probes for *piwi-1*, *fbl-1*, and *fbl-2* in intact asexual planarians at 0-day post irradiation (dpi) and 1 dpi. $n = 3$. Scale bar $= 500\,\mu m$. (**B**) Quantification of WISH images in (**A**). Each dot represents an individual replicate. $n = 3$. Two-tailed unpaired student's *t*-test calculated the *p* values. Data were represented as mean ± SEM. (**C**) Colorimetric WISH of *fbl-1* transcripts during regeneration at 0, 3, 6, 12, 24, and 48 hpa. $n = 3$. Scale bar $= 500\,\mu m$. (**C'**) Quantification of *fbl-1* signals during regeneration at 0, 3, 6, 12, 24, and 48 hpa. Each dot represents an individual replicate. Two-tailed unpaired student's *t*-test calculated the *p* values. Data were represented as mean ± SEM. (**D**) Colorimetric WISH of *fbl-2* transcripts during regeneration at 0, 3, 6, 12, 24, and 48 hpa. $n = 3$. Scale bar $= 500\,\mu m$. (**D'**) Quantification of *fbl-2* signals during regeneration at 0, 3, 6, 12, 24, and 48 hpa. Each dot represents an individual replicate. Two-tailed unpaired student's *t*-test calculated the *p* values. Data were represented as mean ± SEM. (**E**) Heatmap shows the expression levels of *fbl-1*, *fbl-2*, stem cell markers, epidermis markers, and wound response genes at various time points, including 0, 3, 6, 12, 24, and 48 hpa based on a published RNA-seq data (Zeng et al, 2018). Black boxes indicate the differentially expressed genes (adjusted *p* < 0.05) compared to the expression levels at 0 h. (**F**) Heatmap shows the expression levels of *fbl-1*, *fbl-2*, stem cell markers, epidermis markers, and wound response genes at various time points, including 0, 3, 6, 16, 24, and 48 hpa based on a published RNA-seq data (Scimone et al, 2022). Black boxes indicate the differentially expressed genes (adjusted *p* < 0.05) compared to the expression levels at 0 h.

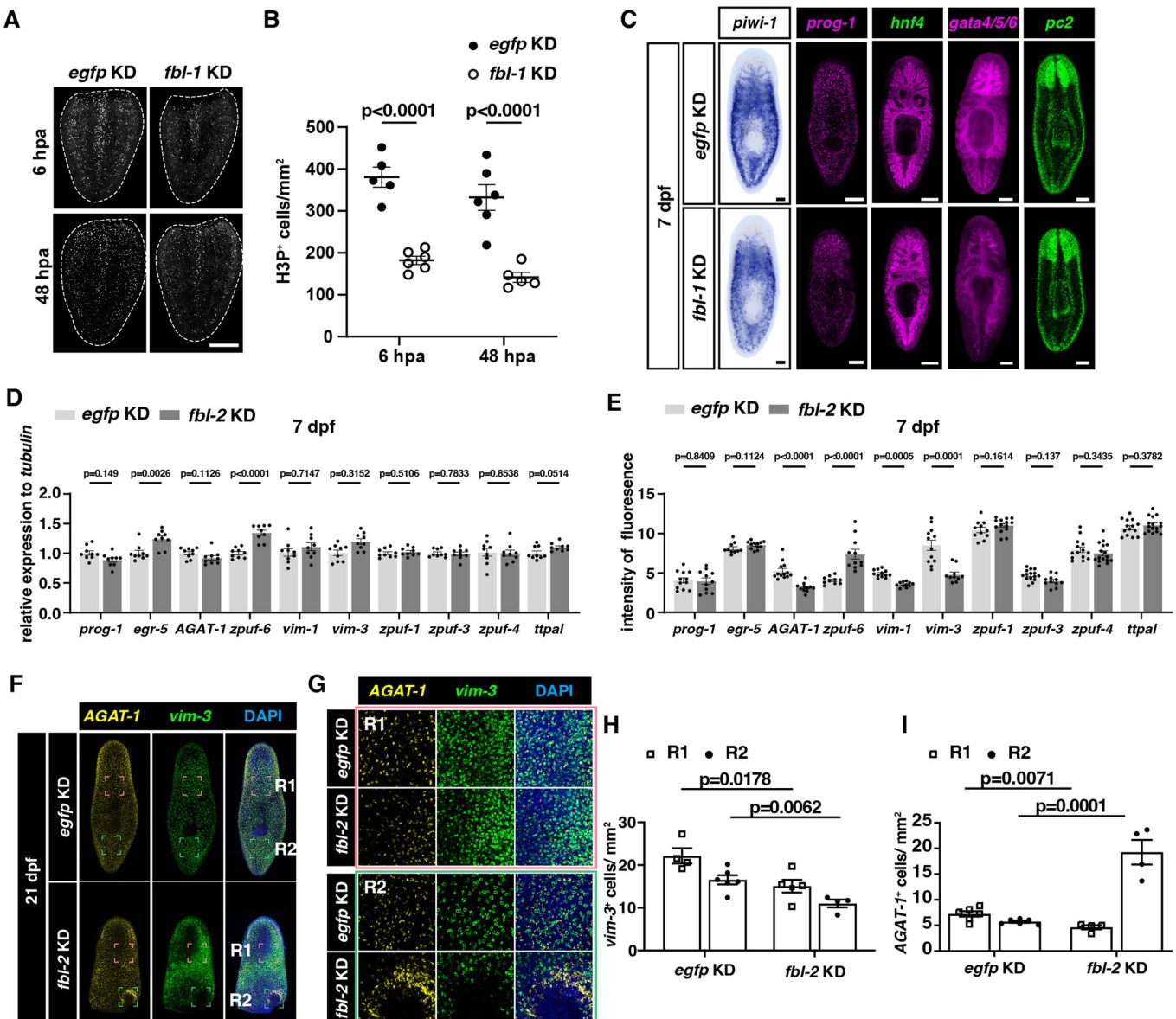

**Figure EV3.  Various cell lineage development in *fbl-1* KD and *fbl-2* KD planarians compared with *egfp* KD control animals.**

(A) Representative immunofluorescent images of proliferating cells (H3P+) in *egfp* KD control and *fbl-1* KD animals at 6 and 48 hpa. *n* = 5–6. Scale bar = 200 μm. White dotted lines indicate the boundary of animals. (B) Quantification of proliferating cells (H3P+) in *egfp* KD control and *fbl-1* KD animals at 6 and 48 hpa. Data were represented as mean ± SEM. Each dot represents an individual replicate. *n* = 5–6. Two-tailed unpaired student's *t*-test calculated the *p* values. (C) Expression of stem cell marker *piwi-1*, epidermal progenitor marker *prog-1*, intestinal markers *hnf4* and *gata4/5/6*, and neuronal marker *pc2* in *egfp* KD control and *fbl-1* KD animals at 7 dpf. *n* = 3–4. Scale bar = 200 μm. (D) Bar plot shows the quantitative real-time PCR for the expression levels of epidermal cell signatures (*prog-1, egr-5, AGAT-1, zpuf-6, vim-1, vim-3, zpuf-1, zpuf-3, zpuf-4*, and *ttpal*) in *fbl-2* KD animals compared with *egfp* KD controls at 7 dpf. *n* = 9. Each dot represents an individual replicate. Two-tailed unpaired student's *t*-test calculated the *p* values. Data were represented as mean ± SEM. (E) Bar plot shows the quantitation of the mean intensity of epidermal cell signatures (*prog-1, egr-5, AGAT-1, zpuf-6, vim-1, vim-3, zpuf-1, zpuf-3, zpuf-4*, and *ttpal*) from FISH in *fbl-2* KD animals compared with *egfp* KD controls at 7 dpf. *n* > 10. Each dot represents an individual replicate. Two-tailed unpaired student's *t*-test calculated the *p* values. Data were represented as mean ± SEM. (F) FISH images for *vim-3+* cells and *AGAT-1+* cells in *fbl-2* KD animals compared with *egfp* KD controls at 21 dpf. Scale bars = 200 μm. (G) FISH images indicate the enlarged regions in the pink (R1) and the green (R2) dashed square in panel (F). Scale bars = 20 μm. (H) Quantification of *vim-3+* cells at the R1 and R2 regions at 21 dpf after *fbl-2* KD. *n* = 3. Each dot represents an individual replicate. Two-tailed unpaired student's *t*-test calculated the *p* values. Data were represented as mean ± SEM. (I) Quantification of *AGAT-1+* cells at the R1 and R2 regions at 21 dpf after *fbl-2* KD. *n* = 3. Each dot represents an individual replicate. Two-tailed unpaired student's *t*-test calculated the *p* values. Data were represented as mean ± SEM.

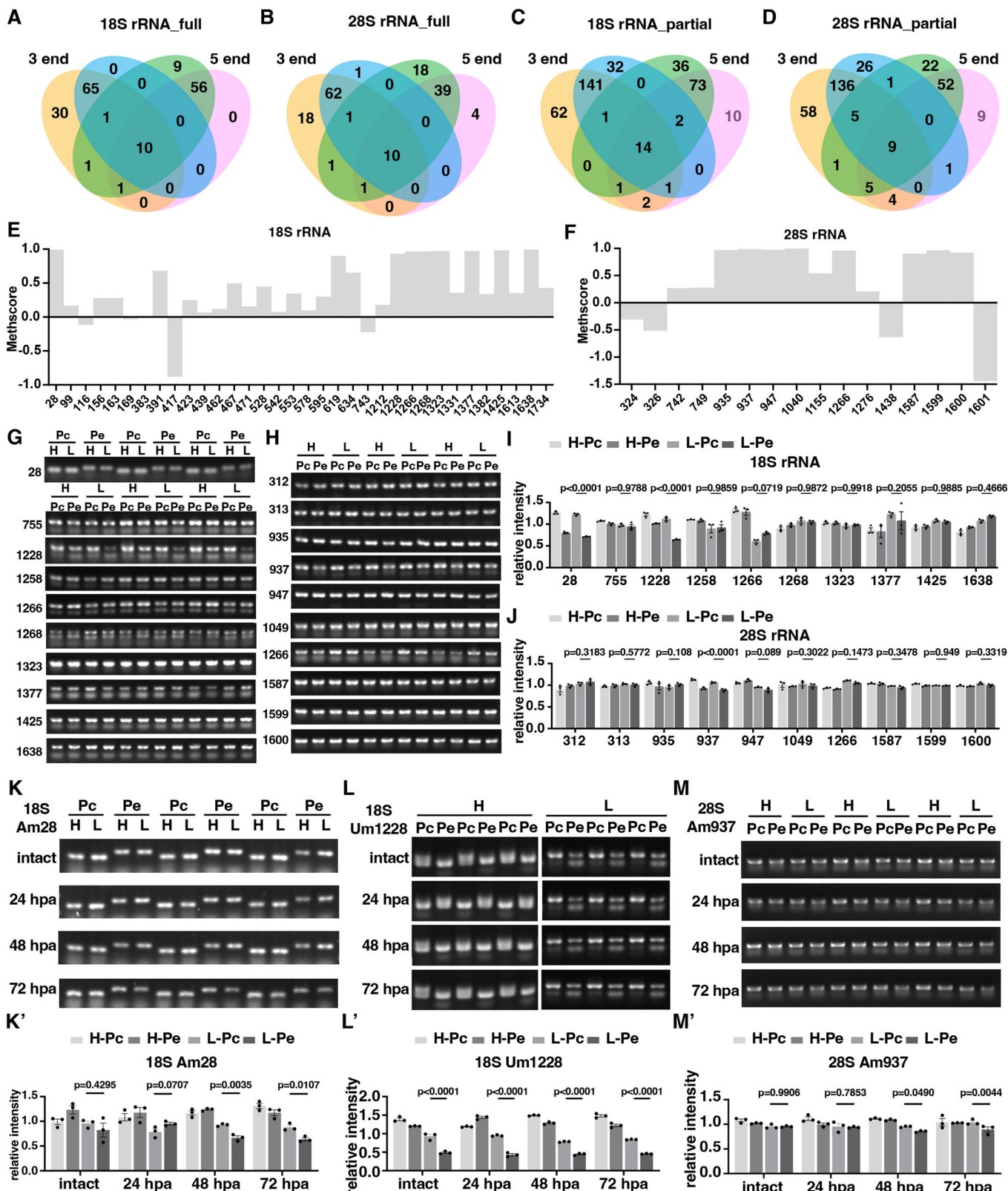

**Figure EV4. Identification of rRNA modification sites in planarians.**

(A) Venn diagram shows the fully 2′-*O*-methylated sites in 18S rRNA by overlapping sites with Methscore >0.85 and <1, in 5′ and 3′ end from *egfp* KD control. (B) Venn diagram shows the fully 2′-*O*-methylated sites in 28S rRNA by overlapping sites with Methscore >0.85 and <1, in 5′ and 3′ end from *egfp* KD control. (C) Venn diagram shows the partially 2′-*O*-methylated sites in 18S rRNA by overlapping sites with Methscore >0.65 and <0.85, in 5′ and 3′ end from *egfp* KD control. (D) Venn diagram shows the partially 2′-*O*-methylated sites in 28S rRNA by overlapping sites with Methscore >0.65 and <0.85, in 5′ and 3′ end from *egfp* KD control. (E) Planarian rRNA 2′-*O*-methylation sites in 18S rRNA known in human cells. (F) Planarian rRNA 2′-*O*-methylation sites in 28S rRNA known in human cells. (G) The detection of fully 2′-*O*-methylated sites in 18S rRNA by RTL-PCR during regeneration (48 hpa). H high dNTP, L low dNTP, Pc primer for control (RT-A anchored reverse transcription primers, FD forward downstream primer), Pe primer for examination (RT-U unanchored reverse transcription primers, FU forward upstream primer). (H) The detection of fully 2′-*O*-methylated sites in 28S rRNA by RTL-PCR during regeneration (48 hpa). (I) Quantification of the band intensity of fully 2′-*O*-methylated sites in 18S rRNA by RTL-PCR in (G). Each dot represents an individual replicate. $n = 3$. Two-tailed unpaired student's *t*-test calculated the *p* values. Data were represented as mean ± SEM. (J) Quantification of the band intensity of fully 2′-*O*-methylated sites in 28S rRNA by RTL-PCR in (H). Each dot represents an individual replicate. Two-tailed unpaired student's *t*-test calculated the *p* values. Data were represented as mean ± SEM. (K−M) The detection of Am28 in 18S rRNA (K), Um1228 in 18S rRNA (L), and Am937 in 28S rRNA (M) by RTL-PCR during homeostasis (intact) and regeneration (24, 48, and 72 hpa). (K′−M′) Quantification of the band intensity of RTL-PCR in K (K′), L (L′), and M (M′). Each dot represents an individual replicate. $n = 3$. Two-tailed unpaired student's *t*-test calculated the *p* values. Data were represented as mean ± SEM.

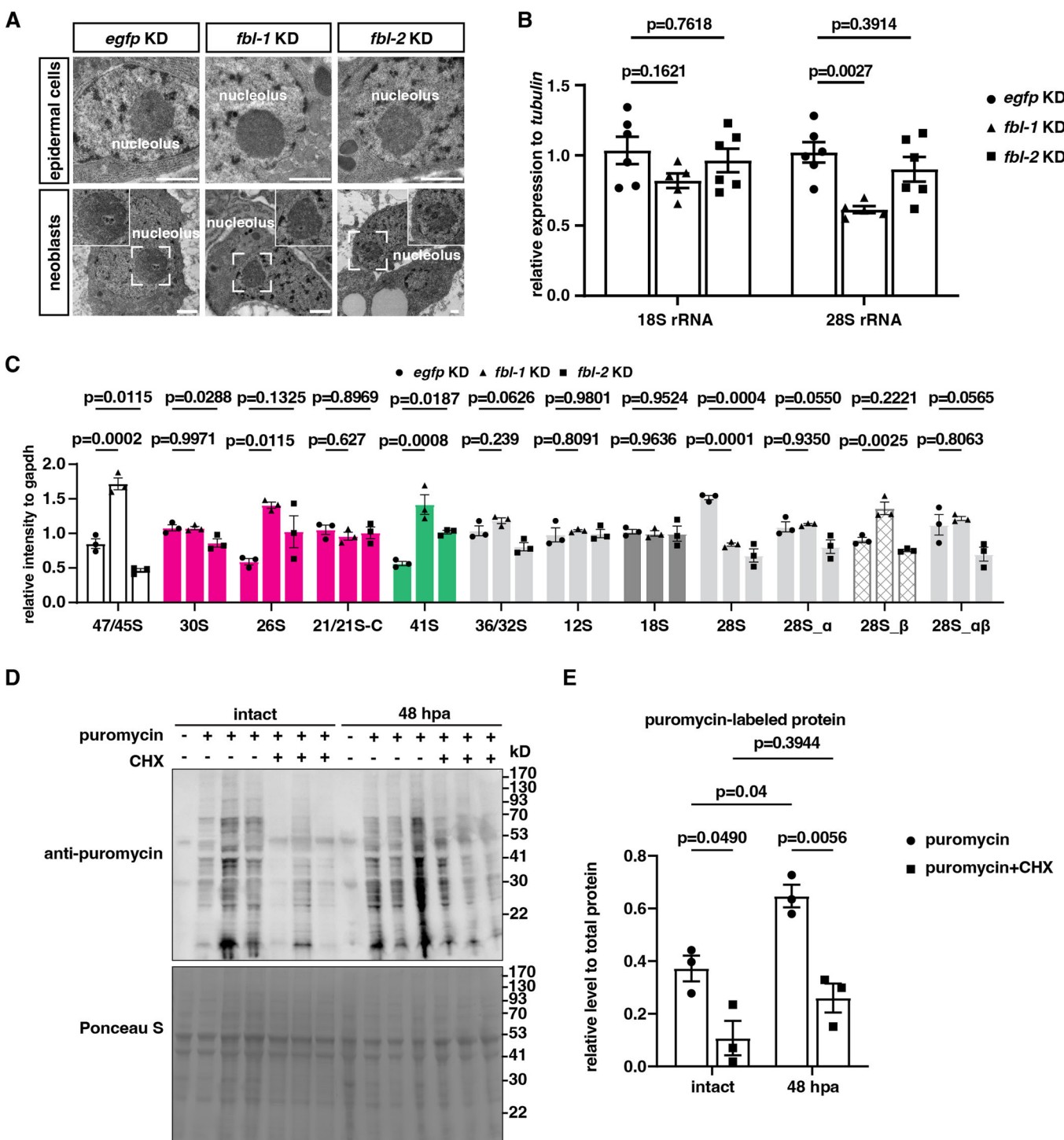

**Figure EV5. Nucleolar structure and relative expression levels of 18S rRNA and 28S rRNA after KD of *fbl-1* and *fbl-2*.**

(A) Transmission electron microscope images show the nucleolar structure in epidermal cells (*n* = 6 for each condition) and stem cells (*n* = 8 for each condition) upon *fbl-1* and *fbl-2* KD compared to *egfp* KD. Scale bar = 1 nm. (B) Relative expression levels of 18S rRNA and 28S rRNA after KD of *fbl-1* and *fbl-2*. *n* = 5–6 biological replicates. Each dot represents an individual animal. One-way ANOVA with Dunnett's multiple comparisons calculated adjusted p values. Data were represented as mean ± SEM. (C) Quantification of rRNA intermediates and mature rRNAs. Precursor rRNA 47/45S, intermediates 30S, 26S, 41S, 36/32S, 18S, and 28S α and β rRNAs are normalized to *gapdh*. *n* = 3 biological replicates. One-way ANOVA with Sidak's test calculated adjusted *p* values. Data were represented as mean ± SEM. (D) Overall protein synthesis rates during homeostasis (intact) and regeneration (48 hpa) under the indicated treatment by labeling puromycin. *n* = 3. (E) Quantification of protein synthesis rates normalized to total protein during homeostasis (intact) and regeneration (48 hpa). *n* = 3. Each dot represents an individual replicate. Two-way ANOVA with Sidak's multiple comparisons tests calculated the *p* values. Data were represented as mean ± SEM.

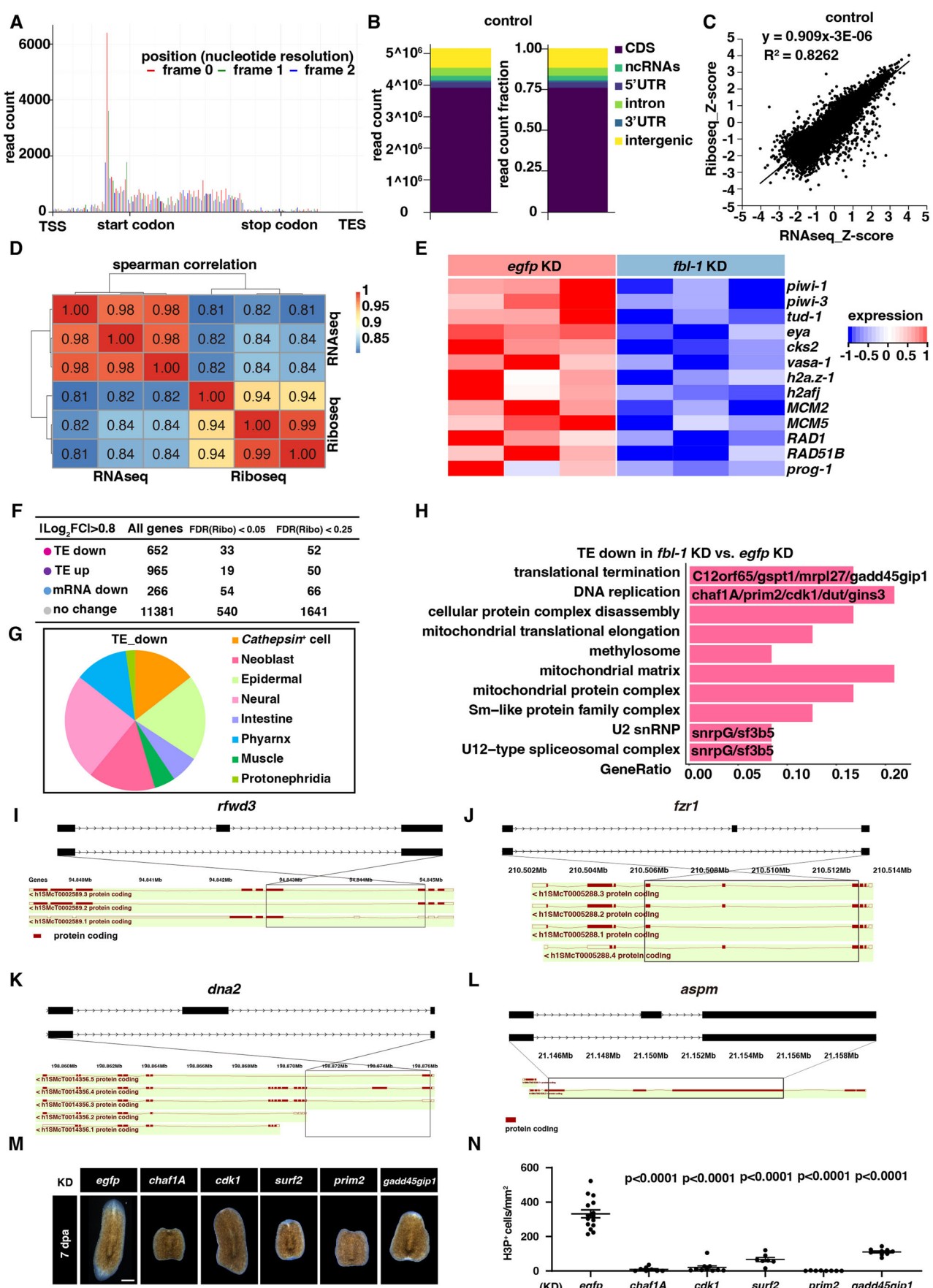

◀

**Figure EV6.  rMATS analysis upon *fbl-1* KD.**

(A) Bar plot shows the metagene read distribution of control Ribo-seq around the start codon and stop codon. Read positions relative to open reading frames (ORF0, 1, 2) are shown in different colors. (B) Read distribution at 5′UTR, CDS, and 3′UTR regions in Ribo-seq analysis of *egfp* KD control. (C) Quantile-quantile plot shows genes in transcriptional level (RNA-seq) and translational level (Ribo-seq) in the *egfp* KD control group. The upper portion of the plot displays the $R^2$ value and the linear formula. (D) The Spearman correlation between RNA-seq and Ribo-seq in the *egfp* KD control group. (E) Heatmap shows differentially expressed genes (adjusted $p < 0.05$) enriched in neoblasts and epidermal early progenitors, including *piwi-1* and *prog-1* in *fbl-1* KD planarians compared to *egfp* KD controls at 48 hpa. (F) Table shows transcripts in each category in Fig. 6A that show a $Log_2$Fold Change (FC) with a cut-off value of ±0.8 in transcription or translational efficiency and false discovery rate (FDR) of translation cut-off values of 0.05 and 0.25. (G) Pie chart shows translational efficiency downregulated mRNA in multiple cell types upon *fbl-1* KD. (H) Gene ontology of translational efficiency downregulated mRNAs in neoblasts, through an overlay of our data with previously published single-cell RNA-seq, revealed the pathway enrichment upon *fbl-1* KD. (I–L) Gene isoforms of *rfwd3*, *fzr1*, *dna2*, *aspm* referring to WormBase ParaSite 18: Schmidtea mediterranea (PRJNA885486). Assembly: schMedS3_haplotype1. Region of *rfwd3*: Scaffold 1_h1:94,839,052-94,845,427. Region of *fzr1*: Scaffold 1_h1:210,501,355-210,514,356. Region of *dna2*: Scaffold 2_h1:198,858,635-198,876,981. Squares indicate gene region with alternative splicing. Region of *aspm*: Scaffold 4_h1:21,144,144-21,159,115. IncLevel, inclusion level. (M) Live images show regenerative defects of worms with indicated KD treatment. $n = 30$. Scale bar $= 500$ μm. (N) Quantification of H3P$^+$ cells in worms with indicated KD treatment. Each dot indicates one sample. $n = 7$–15. Two-tailed unpaired student's $t$-test calculated the $p$ values. Data were represented as mean ± SEM.

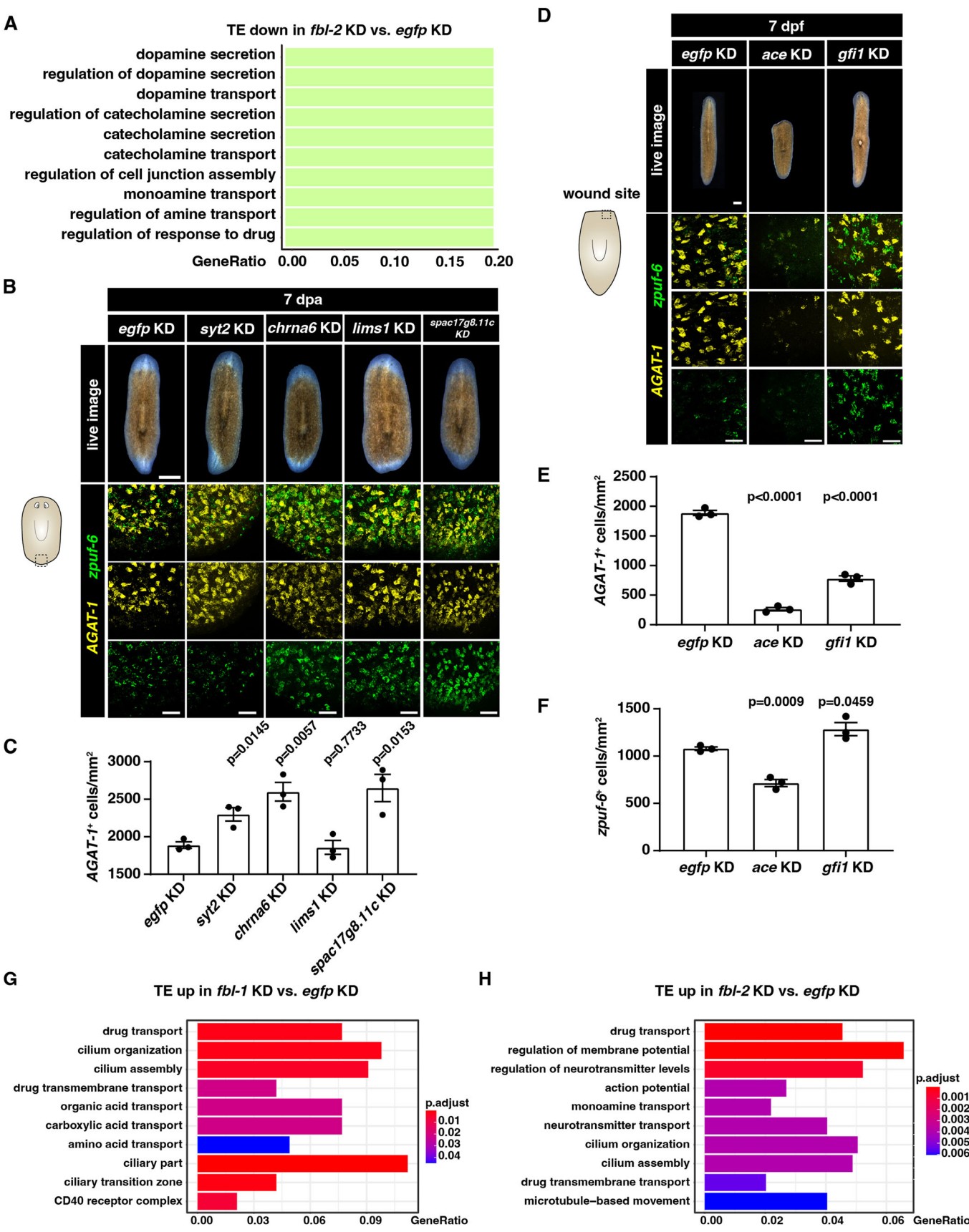

◄ **Figure EV7. Phenotype analysis after KD of genes in TE down upon *fbl-2* KD.**

(A) Gene ontology of translational efficiency downregulated mRNAs in the epidermis through an overlay of our data with previously published single-cell RNA-seq revealed pathway enrichment upon *fbl-2* KD. (B) Live images (n = 30) and FISH show regenerative defects of *syt2* KD, *chrna6* KD, *lims1* KD and *spac17g8.11c* KD animals compared to *egfp* KD controls at 7 dpa. Scale bar = 500 μm in live images and =20 μm in FISH images. Cartoon illustrations show the displayed regions. (C) Bar plot shows the quantification of the density of *AGAT-1*⁺ cells at the posterior regions of regenerating trunks after KD of *egfp*, *syt2*, *chrna6*, *lims1*, and *spac17g8.11c*. n = 3. Data were represented as mean ± SEM. Each dot represents an individual replicate. Two-tailed unpaired student's *t*-test calculated the *p* values. Data were represented as mean ± SEM. (D) Live images (n = 30) and FISH show homeostatic defects of *ace* KD and *gfi1* KD animals compared to *egfp* KD controls at 7 dpf. Scale bar = 500 μm in live images and =20 μm in FISH images. Cartoon illustrations show the displayed regions. (E) Bar plot shows the quantification of the density of *AGAT-1*⁺ cells at the wound sites in *egfp* KD, *ace* KD, and *gfi1* KD animals. n = 3. Data were represented as mean ± SEM. Each dot represents an individual replicate. Two-tailed unpaired student's *t*-test calculated the *p* values. Data were represented as mean ± SEM. (F) Bar plot shows the quantification of the density of *zpuf-6*⁺ cells at the wound sites in *egfp* KD, *ace* KD, and *gfi1* KD animals. n = 3. Data were represented as mean ± SEM. Each dot represents an individual replicate. Two-tailed unpaired student's *t*-test calculated the *p* values. (G) Bar plot shows the gene ontology of translational efficiency upregulated mRNA (TE up) upon *fbl-1* KD. GO terms with *p* adjust <0.05. Fisher's Exact test with Benjamini–Hochberg for multiple test corrections. (H) Bar plot shows the gene ontology of translational efficiency upregulated mRNA (TE up) upon *fbl-2* KD. GO terms with *p* adjust <0.05. Fisher's Exact test with Benjamini–Hochberg for multiple test corrections.

 

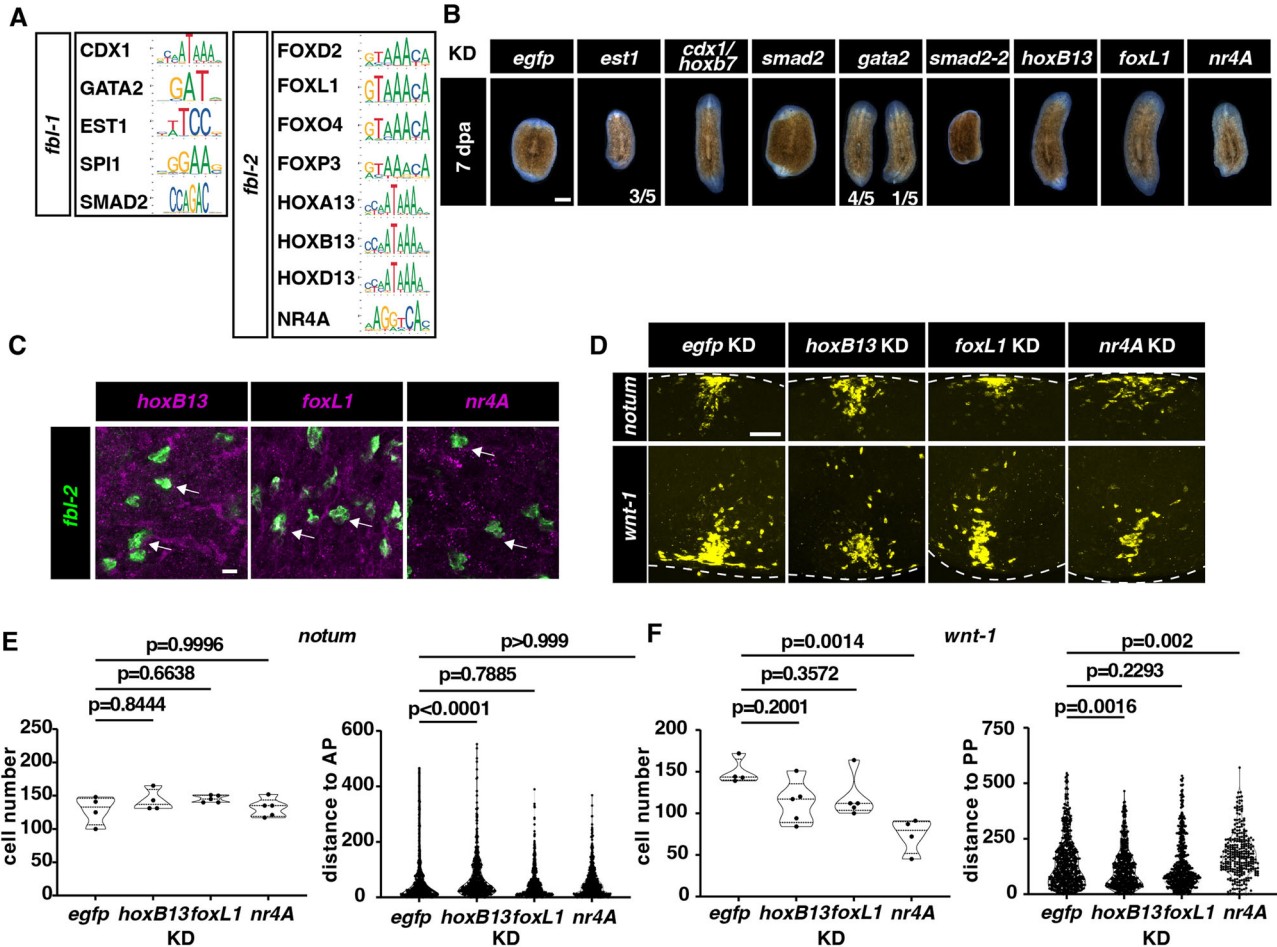

**Figure EV8. Motif analysis of promoters of *fbl-1* and *fbl-2*.**

(A) Binding motif sequences of putative transcription factors predicted from the promoters of *fbl-1* and *fbl-2*. (B) Live images of worms with indicated KD treatment at 7 dpa. $n = 5$. Scale bar = 500 μm. (C) FISH images show the coexpression of putative transcription factors in *fbl-2*$^+$ cells. The white arrow indicated double positive cells. Scale bar = 10 μm. (D) FISH images show *notum* and *wnt-1* signals at the anterior and posterior pole of regenerated tissue upon *egfp* KD, *hoxB13* KD, *foxL1* KD, and *nr4A* KD. Scale bar = 20 μm. (E) Violin plot of quantification of *notum*$^+$ cell number and the distance between *wnt-1*$^+$ cell and the anterior tip in tails of *egfp* KD, *hoxB13* KD, *foxL1* KD, and *nr4A* KD animals at 72 hpa. Each dot represents the cell number and cell distance measured from an individual animal and individual *notum*$^+$ cell in the left and right panels, respectively. $n = 4$–5. One-way ANOVA with the Tukey test calculated adjusted *p* values. Data were represented as mean ± SEM. (F) Violin plot of quantification of *wnt-1*$^+$ cell number and the distance between *wnt-1*$^+$ cell and the posterior tip in heads of *egfp* KD, *hoxB13* KD, *foxL1* KD, and *nr4A* KD animals at 72 hpa. Each dot represents the cell number and cell distance measured from an individual animal and individual *wnt-1*$^+$ cell in the left and right panels, respectively. $n = 4$–5. One-way ANOVA with the Tukey test calculated adjusted *p* values. Data were represented as mean ± SEM.

