## [Peer Review File · The EMBO Journal]

Fibrillarin homologs regulate translation in divergent cell lineages during planarian homeostasis and regeneration

Jiajia Chen, Yucong Li, Yan Wang, Hui Wang, Jiaqi Yang, Xue Pan, Yun Zhao, Hao Xu, Penglei Jiang, Pengxu Qian, Hongwei Wang, Zhi Xie, and Kai Lei

Corresponding author(s): Kai Lei (leikai@westlake.edu.cn)

Review Timeline:

Submission Date:	8th Aug 22
Editorial Decision:	9th Aug 22
Appeal:	13th Feb 24
Editorial Decision:	20th Mar 24
Appeal:	8th Apr 24
Editorial Decision:	26th Apr 24
Revision Received:	21st Jun 24
Editorial Decision:	5th Aug 24
Revision Received:	1st Oct 24
Editorial Decision:	22nd Oct 24
Revision Received:	28th Oct 24
Accepted:	6th Nov 24

Editor: Stefanie Boehm / Ieva Gailite

Transaction Report:

Prof. Kai Lei
Westlake University
School of Life Sciences
No. 18 Shilongshan Street
Hangzhou, Zhejiang 310024
China

9th Aug 2022

Re: EMBOJ-2022-112327
Fibrillarin-1 and Fibrillarin-2 are required for divergent cell lineage development in planarian

Dear Prof. Lei,

Thank you for submitting your manuscript (EMBOJ-2022-112327) to The EMBO Journal. I have now read your study carefully and discussed the work with the other members of the editorial team. However, I regret to inform you that we have decided not to pursue publication of this manuscript in The EMBO Journal.

We appreciate that you report the identification of two planarian fibrillarin (FBL) homologs *smcd-fbl-1* and *smcd-fbl-2*, and find differential cell-type specific expression. Moreover, the phenotypes of *smcd-fbl-1* and *smcd-fbl-2* depletion differ, and you conclude that FBL-1 functions in epidermal lineage development, while FBL-2 plays a role in regulating proliferation and differentiation of stem and progenitor cells. However, while we recognize that this report of planarian FBL homologs will raise interest in the field, we overall find that their specific function in this context and regulation remains to be defined on a molecular level. Thus, also in light of previous studies on fibrillarin in other organisms, we have concluded that the additional functional insight is not sufficient to provide the broader conceptual advance that would be required for further consideration for publication in The EMBO Journal.

That being said, we appreciate the value of the findings to the scientific community and believe that your study is an excellent candidate for our partner journal Life Science Alliance (<http://www.life-science-alliance.org/>; our broad scope Open Access journal published in partnership between the EMBO-, Rockefeller University-, and Cold Spring Harbor Laboratory Presses). The editors of Life Science Alliance would be pleased to send your manuscript for in-depth peer review; no reformatting is required. We very much hope you will be interested in this option: please follow the link below for transfer.

Thank you for giving us the opportunity to consider your manuscript. I am sorry that we cannot be more positive on this occasion and I hope you will be interested in the transfer option.

Kind regards,

Stefanie Boehm

Stefanie Boehm
Editor
The EMBO Journal

*** As a service to authors, The EMBO Journal offers the possibility to directly transfer declined manuscripts to another EMBO Press title (EMBO Reports, EMBO Molecular Medicine, Molecular Systems Biology) or to the open access journal Life Science Alliance launched in partnership between EMBO Press, Rockefeller University Press and Cold Spring Harbor Laboratory Press. The full manuscript (including reviewer comments, where applicable and if chosen) will be automatically forwarded to the receiving journal, to allow for fast handling and a prompt decision on your manuscript. For more details of this service, and to transfer your manuscript to another EMBO title please follow this link:
Link Not Available

Feb. 12, 2024

Dear Editor,

We are delighted to submit our manuscript entitled “Fibrillarin homologs regulate divergent cell lineage in planarian homeostasis and regeneration” for publication in the *EMBO Journal*.

Our study aims to advance our understanding of the mechanisms of adult tissue renewal and regeneration. Adult tissue renewal and regeneration involve complex cellular processes in cell proliferation, fate determination, and differentiation. While many studies focus on gene expression and signaling changes in adult tissue renewal and regeneration, more and more studies have suggested the significance of translational regulation in these processes. Specifically, the translational regulation by rRNA in these processes remains largely unknown. Renowned for remarkable tissue regeneration, Planarians provide an ideal model for studying this. Our current study delves into the role of Fibrillarin (FBL), an RNA 2-O'-methyltransferase crucial for rRNA processing, in planarian *Schmidtea mediterranea* and identified two FBL homologs: *Smed-fbl-1 (fbl-1)* and *Smed-fbl-2 (fbl-2)*. Both are essential for homeostasis and regeneration but play distinct roles. *fbl-1* is crucial for progenitor cell differentiation, while *fbl-2* is important for late-stage epidermal lineage specification.

One and a half years after the initial submission in 2022, we have conducted substantial mechanistic studies to enhance the strength of our study. Our current version has made several notable advances in knowledge. Therefore, we still believe this study is highly suitable for publication in your esteemed journal.

First, we present evidence indicating that different types of rRNA 2-O-methylation are tightly regulated in different cell types. Notably, we observed distinct changes in nucleolus morphology and 2-O'-methylation patterns following *fbl-1* and *fbl-2* knockdown, suggesting their roles in specific mRNA translation during regeneration. **Second**, we report the first examination of planarian Riboseq analysis, which provides insight into differential translational regulation in stem cells and epidermal cells involved in tissue homeostasis and regeneration. Importantly, we find that *fbl-1* knockdown reduced the translation of genes associated with alternative splicing, cell cycle, and DNA replication. Alternatively, *fbl-2* knockdown reduced the translation of genes involved in neurotransmitter secretion. Furthermore, we show evidence of how *fbl-2* regulates the Wnt signaling pathway. **Third**, our study uncovers a significant observation in the animal kingdom regarding the evolution of Fibrillarin (FBL), which has formed two distinct clusters, FBL and FBL-like, suggesting the importance of FBL duplication across diverse species. Our findings suggest that these *fbl* gene duplications may have compensatory roles in specific cell lineage development, emphasizing the necessity for a more comprehensive investigation into the function of FBL homologs in regenerative organisms. Above all, this research significantly contributes to the largely unexplored field of

translational regulation in animal tissue renewal and regeneration. Moreover, it sets the stage for future studies on regulatory mechanisms at multiple levels, through the comparative analysis of transcriptomics, translomics, and proteomics.

Thank you for considering our manuscript for publication. We look forward to answering any questions from you and the reviewers.

Sincerely,
Kai Lei, Ph.D.

Assistant Professor and Principal Investigator
School of Life Sciences
Westlake University

Dear Dr. Lei,

Thank you for submitting your manuscript for consideration by The EMBO Journal. We have now received two reviewer reports on your manuscript, which are included below for your information. Since the third reviewer was delayed in report submission, I am taking the decision based on the input at hand. Based on these comments, we unfortunately had to conclude that the study is not a sufficiently strong candidate for publication in The EMBO Journal.

As you can see, while the reviewers find the topic per se of interest, they also indicate multiple substantial concerns with the experimental approach, depth of analysis and conclusiveness of the findings. Given these opinions from good experts in the research field and since a major experimental revision beyond our usual 3-month timeframe and with an uncertain outcome would be needed to address the main referee concerns, I am afraid that we cannot offer further proceedings towards publication in The EMBO Journal.

While we cannot pursue this manuscript further, I would like to suggest a transfer of your study to our not-for-profit open-access sister journal, Life Science Alliance (LSA). We shared your manuscript and the accompanying reviews with LSA Executive Editor, Eric Sawey, who is interested in these findings, and would like to invite further consideration of this manuscript at LSA pending the following revisions:

- Address Reviewer 1's comments.
- Address Reviewer 2's Major concerns #1.1, 1.3, 1.4, 1.6, and the Minor comments.

We understand that such a revision might need to be re-reviewed, in which case, Dr. Sawey will walk the Reviewers through our transfer process. Please use the link below to transfer your manuscript to LSA:

emboj.msubmit.net/cgi-bin/main.plex?el=A3li6vtP1B2Dnrk6X2A9ftdwoYpB6epjg3G5Cy0gFkQY

You do not need to revise the manuscript before transferring it to LSA. Once you transfer, Dr. Sawey will email you an invitation to revise and resubmit, listing the same revision requests as mentioned above. Please feel free to reach out at e.sawey@life-science-alliance.org if you have any questions about the LSA journal, the transfer process or the revisions requested.

Thank you in any case for the opportunity to consider this manuscript. I regret that I could not communicate more positive news this time, but I nevertheless hope that you will find our reviewers' comments helpful for further improvement of the manuscript.

Yours sincerely,

Ieva Gailite

Referee #1:

The work by Chen et al characterizes the function of the planarian fbl homologs. The major claims of the paper are that (1) fbl-1 is required for progenitor cell differentiation, (2) fbl-2 is required for epidermal lineage specification, (3) identifying the impacts of either fbl-1 and fbl-2 suppression on translation, and (4) the role of rRNA modifications in translational regulation. The inhibition of either fbl-1 and fbl-2 results in a striking phenotype, which the authors then characterize. The he myriad roles of fbl-1 and fbl-2 render these phenotypes difficult to dissect. I outline my main critique below, but I strongly suggest that the authors: (1) carefully edit the manuscript, re-evaluate what parts do not contribute to the story and reduce its clarity. (2) Present the data in the figures more clearly.

Major:

The paper requires substantial editing. I had difficulties understanding some of the authors' observations and the basis of their interpretations. For example, the analyses described in lines 303-317 are unclear, and the language used is inaccurate (e.g.,

using the word "proof" should be avoided; X1 is mentioned but not described). This is a general comment that requires attention and it is not limited to this example.

Authors should be clearer on whether their interpretations of phenotypes (e.g., reduction in H3P; reduction in epidermal progenitors) represent direct or indirect effects of *fbl-1/2* suppression. The text is ambiguous in this respect, and both the results and discussion should present the likely scenario that the effects observed are indirect.

Several major claims are based on limited data, which could be further tested using data that is already reported in the paper. The initial characterization of the *fbl-1/fbl-2* suppression phenotype will benefit from the gene expression data that they obtained by RNAseq. For example, considering the *fbl-2* (RNAi) phenotype shown in figure 2J-L, reduction in the expression of many genes expressed in AGAT-1+ is expected (see Tu et al, ELife, 2015). If that is not the case, the authors need to discuss the discrepancy between the FISH and RNAseq data.

Fig 5A/6A: data shown in figure is unclear and the data presentation is incomplete. The authors need to include data showing that the riboseq works on controls / wildtype, and the correlation between RNAseq and translation efficiency. The analysis is generally not clear

The authors hypothesize that epidermal development relies on communication between *fbl-2+* cells and neurons, but there is no evidence presented to establish this hypothesis.

The observation of spatial proximity between *fbl-2+* and *wnt-1+* cells described by the authors could be a consequence of both genes being injury-induced (Petersen, PNAS 2009; and the data shown by the authors in Fig EV2C). The authors should consider using published data to test whether *fbl-2* is injury induced (see Wenemoser, Genes & Dev, 2012).

Minor

Fig 2A: Suggesting to add a sagittal section to show the DV distribution (low imp)

Fig 2C-D: Explain why the proportion of *fbl-2/egr-5* drops from 89% to 0% from homeostasis to regeneration.

Fig 4E: unclear what is shown, and what kind of differences were evaluated?

Line 41: Language editing is required (e.g., *fbl-2+* cells correlated with *wnt-1+* cells).

99: influent

EV1B: reference format {Robert, 2014 #1340}.

Fig EV1E - Difficult to see. Produce higher resolution image and quantification of the results.

Fig EV2B - Very difficult to see difference in expression between unirradiated and irradiated animals. Provide better quality image and preferably quantification

Fig EV2B-C - missing comparison to 0 hpa. Provide images of similar fragments at 0 hpa for both genes.

Fig EV3A - The expression of *hnf4* appears completely downregulated by 7 dpf. How come that these cells disappear so quickly? Do you see evidence for that in RNAseq/riboseq?

Fig. 3C-G unclear what exactly was counted. Were ovo progenitors counted?

Lines 133-134: Not sure what evolution retention is. The results indicate that both genes are required.

Referee #2:

Dear Editor,

The manuscript by Chen et al. presents a series of experiments to prove the presence of two fibrillarlin homologs in planarian and their distinct roles in homeostasis and regeneration via regulating the translation by differential rRNA modifications. We focus here predominantly on the aspects relating to rRNA modification and translation. There are major concerns that need to be addressed before considering the manuscript for publication.

1. Major concerns:

1.1. Global translation and ribosome biogenesis: The authors should provide analysis of the effects on global translation as well as examine potential effects on ribosome biogenesis. Could the results seen in regeneration capacity and in the ribo-seq experiments result from diminished global translation?

1.2. RiboMethSeq analysis: In the Ribo-meth-seq, the authors zoom in on rRNA 2'Ome sites by comparing to sites known to be methylated in human cells. While this approach may identify conserved sites, it would completely miss planarian-specific sites, which one could argue would be more relevant for, for instance, regeneration. The authors should conduct a careful analysis of the ribo-meth-seq data to identify all potential methylation sites on the planarian rRNA. Identifying bonafide sites over noise may be guided by snoRNA expression levels. Once identified, it would be relevant to investigate 2'Ome dynamics over regeneration to identify sites of potential relevance. The current ribo-meth-seq analysis is from 1 timepoint only. Intriguingly, all sites shown appear to be fully methylated in the control samples. This contrasts with what has been reported for yeast, mouse, and human cells, and could indicate that none of the sites are regulatory.

1.3. Supplementary figure 4: The Fig. EV4 is difficult to understand. For instance, position 18S-28S is claimed to be fully methylated but the bars have the same height as all the neighboring presumably unmethylated sites?

1.4. Classification of 2'-O-me sites: The classification of sites according to Natchair et al. (2017) has been criticized as several of the reported sites turned out to be wrong interpretations of the cryoEM data and were in fact Mg ions (discussed on pubpeer).

1.5. Gene knock-down experiments: More importantly, what can be deduced from gene disruption or knockdown experiments? While KD or KO experiments can be used to imply importance of a particular gene for correct instigation of particular gene expression programs, global manipulation of the fbl-levels do not mimic a naturally occurring process. Hence, analyzing the outcome of ribo-seq experiments likely more reflects the result of a process drifting off course rather than a specific reprogramming. As such, the deregulation of individual genes or gene categories may not be relevant. Unless the authors can demonstrate that fbl1 and 2 are specifically regulated during regeneration resulting in altered rRNA methylation patterns, the ribo-seq results can likely not be interpreted as a specific reprogramming.

1.6. Discussion and Results Discrepancy: The discussion refers to data not presented in the results section.

2. Minor comments:

2.1. Time points clarification: In Figure 1A, the 5 feedings seems to occur over between day 2 and day 7 (2 d and 7 d in the figure). However, in the legend it states, 'schedule of RNAi feeding every three days...'. Please adjust the figure or legend to clarify the conditions.

2.2. Homeostasis definition: The definition of the term homeostasis used in the manuscript is currently unclear and imprecise. For example, for Figure 1B a live image shows the three worms are morphologically similar 7 days post feeding but it's not clear how 'homeostatic maintenance' is assessed using the images. Using terms such as 'viability' or 'morphologically similar to wild-type' would be clearer to the reader.

2.3. KD Confirmation: Why fbl-1 KD and fbl-2 KD groups are not used for fbl-1 and fbl-2 staining? So, it can be used for confirmation of KD.

2.4. Missing Information: The number of used animals for some experiments like survival curves (Fig. 1C) should be included. Based on the figure, these numbers appear to be imbalanced. The authors should explain why there were different numbers used between conditions.

2.5. Biological Replicates and Quantification: Biological replicates and their subsequent quantification are missing for FISH staining images, like Fig. 2 and Fig. 3C, Fig. 6E.

2.6. Negative Controls: Several negative controls are missing. For example, negative control in staining of Fig. EV2A as a background. Timepoints of post-amputation experiments are not same as each other for fbl-1 and fbl-2 staining in Fig. EV2B and Fig. EV2C.

2.7. Non-convincing conclusions: Several conclusions are not convincing due to lack of supportive data. For example, in Fig. EV2B, irradiation sensitivity cannot be concluded without quantification and comparison with control groups.

2.8. Rephrase: Rephrase sentences starting on lines 278 and 333 for clarity. 'To determine which transcripts are regulated...' and 'Within the catecholamine secretion pathway...' respectively.

2.9. Missing Reference to figure: There isn't any reference to Fig. 6C in the manuscript.

2.10. Data Presentation Inconsistency: The result section refers to data not presented. For example, in lines 395, "we confirmed the expression patterns of these genes with fbl-1 and fbl-2, respectively (Fig. EV7C)." However, coexpression of only three genes, not all the selected transcription factors, with only fbl-2, not for both fbls is shown in Fig. EV7C.

2.11. Expected Citation: Fincher, Christopher T., et al. "Cell type transcriptome atlas for the planarian *Schmidtea mediterranea*." *Science* 360.6391 (2018): eaaq1736.

** As a service to authors, EMBO Press provides authors with the possibility to transfer a manuscript that one journal cannot offer to publish to another EMBO publication or the open access journal *Life Science Alliance* launched in partnership between EMBO Press, Rockefeller University Press and Cold Spring Harbor Laboratory Press. The full manuscript and if applicable, reviewers' reports, are automatically sent to the receiving journal to allow for fast handling and a prompt decision on your manuscript. For more details of this service, and to transfer your manuscript please click on Link Not Available. **

Thank you very much for reading our manuscript and providing us the constructive suggestions and comments. We highly appreciate the opportunity to enhance our study and manuscript. We have performed further experiments and revised our manuscript according to your suggestions. The following are the point-by-point responses to address each of your comments.

Referee #1:

The work by Chen et al characterizes the function of the planarian fbl homologs. The major claims of the paper are that (1) fbl-1 is required for progenitor cell differentiation, (2) fbl-2 is required for epidermal lineage specification, (3) identifying the impacts of either fbl-1 and fbl-2 suppression on translation, and (4) the role of rRNA modifications in translational regulation. The inhibition of either fbl-1 and fbl-2 results in a striking phenotype, which the authors then characterize. The he myriad roles of fbl-1 and fbl-2 render these phenotypes difficult to dissect. I outline my main critique below, but I strongly suggest that the authors: (1) carefully edit the manuscript, re-evaluate what parts do not contribute to the story and reduce its clarity. (2) Present the data in the figures more clearly.

Major:

1. The paper requires substantial editing. I had difficulties understanding some of the authors' observations and the basis of their interpretations. For example, the analyses described in lines 303-317 are unclear, and the language used is inaccurate (e.g., using the word "proof" should be avoided; X1 is mentioned but not described). This is a general comment that requires attention and it is not limited to this example.

Response: Thank you very much for your advice. We have made efforts to enhance the clarity and readability of the manuscripts by refining the language and providing explanations for specific terms. We have first revised “prove” to “suggest” in the original lines 303-317, and confirmed similar usage throughout the manuscript. In the 2nd paragraph of the Discussion section, we revised the sentence as “In planarians, rRNA 2'-O-methylation sites were first captured by RiboMeth-seq in this study. Our study showed that the methylation frequency varies among individual sites and during regeneration. The differentially methylated frequency is different between *fbl-1* KD and *fbl-2* KD animals. Together with the Ribo-seq analyses, our findings suggest a ribosomal heterogeneity in various planarian cells”.

We agree that it is important to clarify that X1 cells are a cell population categorized within planarian cells. We have added a sentence in lines 401-403 for this term “Based on flow cytometry analysis, planarian cells were broadly categorized as X1, X2, and Xins, in which X1 cells enrich the proliferating stem cells at the S/G2/M cell cycle phases (Reddien *et al.*, 2005; Hayashi *et al.*, 2006)”.

Also suggested by the other reviewer, the writings in lines 303-317 and the two sections related to the translational regulation need to be improved for clarity. We have included a description in lines 397-417 that elaborates on the splicing events analysis after *fbl-1* KD: “KD of *small nuclear ribonucleoprotein polypeptide G (snrpG)* and *splicing factor 3b subunit 5 (sf3b5)* resulted in head regression (Fig. 6B). KD of RNA polymerase II subunit I (*polr2i*) led to delayed planarian regeneration (Fig. 6D).

KD of *snrpG*, *sf3b5*, and *polr2i* all caused the reduction of stem cell proliferation as shown by staining of H3P⁺ cells (Fig. 6C, E).....". This revision also improved the presentation for the data in the figures.

We have also marked the revisions in red throughout the manuscript.

2. Authors should be clearer on whether their interpretations of phenotypes (e.g., reduction in H3P; reduction in epidermal progenitors) represent direct or indirect effects of fbl-1/2 suppression. The text is ambiguous in this respect, and both the results and discussion should present the likely scenario that the effects observed are indirect.

Response: We agree with your suggestion to discuss on the direct or indirect effects caused by *fbl-1/2* KD. Before observing the phenotypes resulting from *fbl-1/2* KD, we identified the expression patterns of *fbl-1* and *-2*, which were distinctly expressed in neoblasts and epidermal cells, respectively. As a result, the presented phenotypes were associated with cell-type-specific expression. However, we acknowledge the concerns raised regarding the potential indirect effects of *fbl-1/2* suppression. For example, we also observed a reduction in H3P⁺ cells in the *fbl-2* KD animals. Previous studies have indicated that the epidermal lineage cells have a feedback influence to neoblast proliferation (Tu *et al.*, 2015; Zhu *et al.*, 2015), which is the reason why we did not emphasize this result. To discuss the possibility of indirect effects caused by *fbl-2*, we have added sentences in three places.

(1) We add a sentence in lines 209-211: "Since the expression of *fbl-2* is enriched in the epidermal lineage, this effect is likely to be an indirect feedback influence from the defects in epidermal cells, similar to the previous studies (Tu *et al.*, 2015; Zhu *et al.*, 2015)".

(2) After the phenotype examination, we add two sentences in lines 241-244: "The defective phenotypes in the reduction of late-stage differentiated cells in *fbl-1* KD and *fbl-2* KD animals may be the outcome of the defects in progenitor cells at earlier stages. Our findings emphasize the essential functions of *fbl-1* and *fbl-2* in a physiological context."

(3) We have also referenced previous studies in the 3rd paragraph of the Discussion section in the manuscript: "Additionally, we also observed a reduction in H3P⁺ cells after *fbl-2* suppression, which may support the hypothesis of feedback regulation between epidermis and neoblasts (Tu *et al.*, 2015; Zhu *et al.*, 2015)".

3. Several major claims are based on limited data, which could be further tested using data that is already reported in the paper. The initial characterization of the fbl-1/fbl-2 suppression phenotype will benefit from the gene expression data that they obtained by RNAseq. For example, considering the fbl-2 (RNAi) phenotype shown in figure 2J-L, reduction in the expression of many genes expressed in AGAT-1+ is expected (see Tu et al, ELife, 2015). If that is not the case, the authors need to discuss the discrepancy between the FISH and RNAseq data.

Response: Thanks for the constructive suggestion. We have taken your advice and conducted an analysis of gene expression changes in *fbl-1* KD and *fbl-2* KD compared to *egfp* KD controls. As shown in the figure below, the RNA-seq analysis of

fbl-1 KD at 48 hpa reveals a significant decrease in the expression levels of *piwi-1*, *prog-1*, and multiple cell cycle-related genes (in green square), which aligns with the results obtained from FISH experiments. We have incorporated the bulk RNA-seq analysis of *fbl-1* KD vs. *egfp* KD in the revised Figure EV6.

Because the time point of bulk RNA-seq is not the same as the detection of AGAT-1+ cells at 7 dpf, we then examined the expression and cell number of those genes (*prog-1*, *egr-5*, *AGAT-1*, *zpuF-6*, *vim-1*, *vim-3*, *zpuF-1*, *zpuF-3*, *zpuF-4*, and *ttpal*) in AGAT-1+ cells by qPCR and FISH. We found that while the expression of *egr-5* and *zpuF-6* increased, most of the other genes expressed in AGAT-1+ cells were not changed according to the qPCR analysis (Fig. EV3D). However, using FISH, we detected a decrease in the cell number of AGAT-1+, *vim-1*+, and *vim-3*+ cells, but an increase of the *zpuF-6*+ cells following *fbl-2* KD (Fig. EV3E).

D Bar plot shows the quantitation real-time PCR for the expression levels of epidermal cell signatures (*prog-1*, *egr-5*, *AGAT-1*, *zpuF-6*, *vim-1*, *vim-3*, *zpuF-1*, *zpuF-3*, *zpuF-4*, and *ttpal*) in *fbl-2* KD animals compared with *egfp* KD controls at 7 dpf. n = 9. Two-tailed unpaired student's *t*-test calculated the p values. **, p < 0.01; ****, p < 0.0001; ns, no significance.

E. Bar plot shows the quantitation of mean intensity of signals of epidermal cell signatures (*prog-1*, *egr-5*, *AGAT-1*, *zpuF-6*, *vim-1*, *vim-3*, *zpuF-1*, *zpuF-3*, *zpuF-4*, and *ttpal*) from FISH in *fbl-2* KD animals compared with *egfp* KD controls at 7 dpf. n > 10. Two-tailed unpaired student's *t*-test calculated the p values. ***, p < 0.001; ****, p < 0.0001; ns, no significance.

It was worth noting that these genes were not completely expressed in the same

cell populations. The difference in the results of qPCR and FISH could be due to the different sensitivity in the detection of small population cells out from the whole-body cells, and on the levels of transcripts versus cell numbers for each analysis.

We then conducted bulk RNA-seq analysis of the *fbl-2* KD animals at 48 hpa and also quantified the *AGAT-1*⁺ and *zpuif6*⁺ cell numbers at the injury region of *fbl-2* KD with *egfp* KD controls. It is consistent that the *egr-5* displayed upregulation in RNA-seq and qPCR (A, B). By FISH, the *vim-3*⁺ was decreased, whereas the *egr-5*⁺ was not significantly changed (C). We further quantified the expression of *egr-5* and *vim-3* at regions of the wound and far away wound, respectively (D, E). The *egr-5*⁺ was increased and *vim-3*⁺ was decreased at wound sites after *fbl-2* KD. It was consistent with the results that at 21 dpf, the damaged tails of *fbl-2* KD exhibited reduced expression of *vim-3* at both intact and damaged regions (Fig. EV3F-H). In contrast, *AGAT-1*⁺ cells were consistently decreased in the intact region, but an accumulation of *AGAT-1*⁺ cells was observed in the damaged region (Fig. EV3F, G, I). This observation suggested that the expression of *AGAT-1* was not directly regulated by *fbl-2*, or the induced expression of *AGAT-1* by injury was regulated by an independent mechanism. Based on the expression pattern of *fbl-2*, our findings suggested that *fbl-2* might promote the differentiation of a subset of epidermal cell lineage consisting of *egr-5*⁺*fbl-2*⁺ into *vim-3*⁺ cells. These observations further suggested that the difference in the results of qPCR and FISH could be due to the different sensitivity in the detection of small population cells out from the whole-body cells, and on the levels of transcripts versus cell numbers for each analysis.

A. Heatmap shows differential genes expressed in *AGAT-1*⁺ cells from RNA-seq data after *fbl-2* KD at 48 hpa. Green star indicates the significant differential expression genes.

B. Bar plot shows the quantitation real-time PCR for the expression levels of epidermal cell signatures (*prog-1*, *egr-5*, *AGAT-1*, *zpuif-6*, *vim-1*, and *vim-3*) in *fbl-2* KD

animals compared with *egfp* KD controls at 48 hpa. $n = 9$. Two-tailed unpaired student's *t*-test calculated the *p* values. **, $p < 0.01$; ****, $p < 0.0001$; ns, no significance.

C. Bar plot shows the quantitation of mean intensity of epidermal cell signals (*prog-1*, *egr-5*, *AGAT-1*, *zpuF-6*, *vim-1*, and *vim-3*) from FISH in *fbl-2* KD animals compared with *egfp* KD controls at 48 hpa. $n > 10$. Two-tailed unpaired student's *t*-test calculated the *p* values. **, $p < 0.01$; ****, $p < 0.0001$.

D. Bar plot shows the quantitation of mean intensity of *egr-5* signals at wound sites and far away wound sites from FISH in *fbl-2* KD animals compared with *egfp* KD controls at 48 hpa. $n > 5$. Two-way ANOVA with sidak's multiple comparisons tests calculated the *p* values. *, $p < 0.05$; **, $p < 0.01$; ns, no significance.

E. Bar plot shows the quantitation of mean intensity of *vim-3* signals at wound sites and far away wound sites from FISH in *fbl-2* KD animals compared with *egfp* KD controls at 48 hpa. $n > 5$. Two-way ANOVA with sidak's multiple comparisons tests calculated the *p* values. *, $p < 0.05$; ***, $p < 0.001$; ****, $p < 0.0001$; ns, no significance.

Based on the above analysis, we believe that utilizing FISH staining is a more dependable approach for evaluating alterations in the cell number of the epidermal cell lineage. Considering the complexities involved in comparing the RNA-seq, qPCR, and FISH at 48 hpa, we have decided not to include the analysis at 48 hpa in the manuscript. It does not influence the conclusion and will improve the readability of the manuscript. To reflect the new analysis, we have made revisions in lines 208-236, which have been highlighted in red.

4. Fig 5A/6A: data shown in figure is unclear and the data presentation is incomplete. The authors need to include data showing that the riboseq works on controls / wildtype, and the correlation between RNAseq and translation efficiency. The analysis is generally not clear

Response: It is a good suggestion to present the Ribo-seq analysis of the *egfp* KD controls prior to conducting the comparative analysis between *fbl-1/2* RNAi and control conditions. In our experiments, we obtained the worms at 48 hours post-amputation for both Ribo-seq and RNA-seq library construction to minimize the discrepancy. The worm samples were divided into a 25% portion for RNA-seq and a 75% portion for Ribo-seq, with three replicates. Subsequently, we carried out an integrated analysis of two datasets. Quality control analysis confirmed that the Ribosome protected frame exhibited typical Ribo-seq features, including the expected mapping rates, the depletion of signals from 3'UTRs, and the presence of the characteristic 3-nucleotide (nt) periodicity as described in other Ribo-seq studies (A-B). After careful examination of the data, we showcased the correlation between RNA-seq and translation efficiency in the figure below (C).

In response to your previous comment, we have thoroughly revised the two sections about the *fbl-1/2* Ribo-seq analyses as follows.

(1) We have included the description of the quality control analysis and the correlation between RNA-seq and translation efficiency on *egfp* KD controls in the

revised manuscript lines 364-374 “To gain a comprehensive understanding of the role of *fb1* in translational control, we performed Ribo-seq to identify the transcripts being translated, and we evaluated changes in transcription level using RNA-seq at 48 hpa. We obtained the worms at 48 hpa for both Ribo-seq and RNA-seq library construction to minimize the discrepancy. The worm samples were divided into a 25% portion for RNA-seq and a 75% portion for Ribo-seq, with three replicates. Quality control analysis confirmed that the ribosome protected frame exhibited typical Ribo-seq features, including the expected mapping rates, the depletion of signals from 3’UTRs, and the presence of the characteristic 3-nucleotide (nt) periodicity as described in other Ribo-seq studies (Fig. EV6A, B). The data between RNA-seq and Ribo-seq of control group also presented high correlation (Fig. EV6C)”. We have added the results in the revised Figure EV6.

(2) We have added the bulk RNA-seq analysis as you suggested in the previous comment in the revised Figure EV6D. The statement is also included in lines 377-379 “Additionally, the expression levels of transcripts enriched in neoblasts and epidermal early progenitors were decreased in *fb1-1* KD compared with *egfp* KD controls (Fig. EV6D), supporting our FISH results”.

(3) We have revised the paragraphs you suggested in your first comment to clarify the logic flow of the RNAi analysis for the candidate genes identified from the Ribo-seq analysis. The revisions are marked in red.

- A. Bar plots showing the metagene read distribution of control Riboseq around the start codon and stop codon. Read positions relative to ORF reading frames (0, 1, 2) are shown in different colors.
- B. Read distribution at 5'UTR, CDS, and 3'UTR regions of the control Ribo-seq.
- C. The correlation between RNA-seq and Ribo-seq in the control group

5. *The authors hypothesize that epidermal development relies on communication between *fbl-2*+ cells and neurons, but there is no evidence presented to establish this hypothesis.*

Response: We apologize for the confusion in making the hypothesis in the original manuscript without experimental validation. Through Ribo-seq data analysis, our study identified two specific down-regulated genes expressed in the epidermal cells: *neuronal acetylcholine receptor subunit alpha-6 (chrna6)* and *synaptotagmin 2 (syt2)*. These two genes are involved in neurotransmitter transport. Previous studies have shown a close association between the epidermal cells and muscular cells as well as neurons (see Eisenhoffer et al., 2008; Tu et al., 2015). Based on this evidence, we proposed that epidermal development relies on communication between *fbl-2*⁺ cells and neurons.

We agree with your comment that we should conduct experiments to assess the plausibility of the statement. To validate this claim, we have carried out RNAi experiment to knock down each of these two genes. After the suppression of *syt2* and *chrna6*, the animals regenerated the anterior and posterior tissues at 7 dpa but showed an increase in *AGAT-1*⁺ cells at the posterior of regenerating trunks of *chrna6* KD planarians, suggesting a role of neurotransmitter signals in epidermal lineage development. We agree with your concerns that this hypothesis needs more data to support beyond the scope of this study. We therefore think it better to describe the results, but remove the hypothesis from the revised manuscript.

We have revised the manuscript in lines 440-451 as “The catecholamine secretion pathway, which involves dopamine, and the cell junction assembly pathway were found to be enriched in *fbl-2* KD animals compared to *egfp* KD controls (Fig. EV7A). Both *synaptotagmin 2 (syt2)* and *neuronal acetylcholine receptor subunit alpha-6 (chrna6)* were enriched in the catecholamine secretion pathway. Previous studies identified that *syt2* as a Ca²⁺ sensor in synapse regulates transmitter release (Xu et al, 2007), and *chrna6* is a nAChR subunit that responds to nicotine (Qian et al, 2016). KD of *syt2* and *chrna6* caused an increase of *AGAT-1*⁺ cells at the posterior of regenerating trunks, similar to the phenotype of *fbl-2* KD animals, in spite of no obvious regeneration defects (Fig. EV7B, C). Given that in planarians, epidermal cells (*AGAT-1*⁺) express creatine similar to that in neurons and muscle cells (Eisenhoffer et al., 2008; Tu et al., 2015), our results suggested a role of neurotransmitter in the development of epidermis”.

6. *The observation of spatial proximity between *fbl-2*+ and *wnt-1*+ cells described by the authors could be a consequence of both genes being injury-induced (Petersen, PNAS 2009; and the data shown by the authors in Fig EV2C). The authors should consider using published data to test whether *fbl-2* is injury induced (see Wenemoser, Genes & Dev, 2012).*

Response: Thank you for your comment regarding the relationship between *fbl-2*+

and *wnt-1*⁺ cells. The previous study by Wenemoser et al. used microarray method without *fbl-2* in the array panel. As shown in figures below, we, therefore, have examined RNA-seq data obtained from regenerating fragments (Zeng et al., 2018, Cell), which provided the evidence for the induction of *fbl-2* expression following tissue damage at 12 hpa and 24 hpa, while the expression of *wnt-1* was induced as early as 6 hpa and persisted until 48 hpa. In addition, a recent RNA-seq data from planarian wound sites (blastema area) (Scimone et al., 2022, Nature Communication), indicates that the expression of *wnt-1* was increased as early as 3 hpa, whereas the upregulated expression of *fbl-2* commenced at 24 hpa. These datasets indicate that the time point at which *fbl-2* expression increased was later than the time point at which *wnt-1* expression was upregulated. Although we acknowledge the possibility you raised, our main focus lies on examining the function of *fbl-2* on the normal distribution of *wnt-1*⁺ cells rather than its expression level, as shown in the original Fig. 6G-H.

E. Heatmap shows the expression levels of *fbl-1*, *fbl-2*, stem cell markers, epidermis markers, and wound response genes at various time points, including 0, 3, 6, 12, 24, and 48 hpa based on the RNA-seq data of Zeng et al. (Cell, 2018). Black boxes indicate the significant differential expression compared to the expression levels at 0 h.

F. Heatmap displays the expression levels of *fbl-1*, *fbl-2*, stem cell markers, epidermis markers, and wound response genes at various time points, including 0, 3, 6, 16, 24, and 48 hpa based on the RNA-seq data from Scimone et al. (Nature Communication, 2022). Black boxes indicate the significant differential expression compared to the expression levels at 0 h.

Minor

Fig 2A: Suggesting to add a sagittal section to show the DV distribution (low imp)

Response: It is a good suggestion. The signal was not strong enough to present on the sagittal sections. We found that it is better to show the expression pattern of *fbl-1/2* in the orthogonal view for the DV distribution in Fig. 2B.

Fig 2C-D: Explain why the proportion of *fbl-2/egr-5* drops from 89% to 0% from homeostasis to regeneration.

Response: We do not have a concise answer, but it is quite an interesting observation. The previous study by Tu et al. 2015 found that the *egr-5*⁺ cells in the epidermal cell lineage development give rise to *vim-3*⁺ cells. This differential proportion of *fbl-2*⁺*egr-5*⁺ between homeostasis and regeneration, with an opposing proportion for *fbl-2*⁺*vim-3*⁺, suggests that *fbl-2* is involved in epidermal cell development and has a distinct function during regeneration at 48 hpa. Another possibility could be the transit expression shift, which drives the differentiation of *fbl-2*⁺*egr-5*⁺ cells to *fbl-2*⁺*vim-3*⁺ cells that will gradually return to a high *fbl-2*⁺*egr-5*⁺ ratio in later stages of regeneration.

To validate these possibilities, we have performed the dual FISH at multiple time points during regeneration to quantitatively examine the dynamic changes. Our results aligned with the possibility that there is a transit expression shift, which drives the differentiation of *fbl-2*⁺*egr-5*⁺ cells to *fbl-2*⁺*vim-3*⁺ cells that will gradually return to a high *fbl-2*⁺*egr-5*⁺ ratio in later stages of regeneration. We have incorporated the results in the revised manuscript lines 184-189 as “The differentiation from *fbl-2*⁺*egr-5*⁺ cells to *fbl-2*⁺*vim-3*⁺ cells gradually returns to a high ratio of *fbl-2*⁺*egr-5*⁺ cells in later stages of regeneration (Fig. 2E, F). This differential proportion of *fbl-2*⁺*egr-5*⁺ cells between homeostasis and regeneration, coupled with an opposing proportion for *fbl-2*⁺*vim-3*⁺ cells, suggests that *fbl-2* is involved in the development of epidermal cells and possesses a distinct function during regeneration.”. Meanwhile, I have also added the results in the revised Figure 2F.

F. Quantification of *fbl-2+egr-5+* and *fbl-2+vim-3+* cells during regeneration at 0 dpa, 1 dpa, 2 dpa, 3 dpa, 5 dpa, and 7 dpa. n = 6 animals for each time point.

Fig 4E: unclear what is shown, and what kind of differences were evaluated?

Response: We show the nucleolus in the nucleus with TEM images. By comparing the morphology of the nucleolus, we did not observe a difference among *egfp* KD control, *fbl-1* KD, and *fbl-2* KD. We have added the labels in the figure and description in the manuscript: “However, no significant differences in the morphology of the nucleolus were observed after *fbl-1* and *fbl-2* KD”. The original Fig 4E was moved to Fig. EV5A.

A. Transmission electron microscope images show the nucleolar structure in epidermal cells (n = 6 for each condition) and stem cells (n = 8 for each condition) upon *fbl-1* and *fbl-2* KD compared to *egfp* KD. Scale bar = 1 nm.

Line 41: Language editing is required (e.g., fbl-2+ cells correlated with wnt-1+ cells).

Response: We have revised the sentences as “*fbl-2*⁺ cells displayed a spatial correlation with *wnt-1*⁺ cells, suggesting a possible involvement of *fbl-2*⁺ cells in the localization of *wnt-1*⁺ cells”.

99: influent

Response: We have revised “influent” to “influenced”.

EV1B: reference format {Robert, 2014 #1340}.

Response: We feel sorry about this carelessness. We have corrected it in the figure legend.

Fig EV1E - Difficult to see. Produce higher resolution image and quantification of the results.

Response: The low-resolution images came from the transformation of the documents into the combined pdf supplemental file. We have submitted the high-resolution figures this time. Fig. EV1F is the quantification of the image results of EV1E.

Fig EV2B - Very difficult to see difference in expression between unirradiated and irradiated animals. Provide better quality image and preferably quantification

Response: The reason is the same as the above comment. The low-resolution images came from the transformation of the documents into the combined pdf supplemental file. We have submitted the high-resolution figures this time. Meanwhile,

we have conducted the quantification and showed the results in the revised Figure EV2B.

Fig EV2B-C - missing comparison to 0 hpa. Provide images of similar fragments at 0 hpa for both genes.

Response: It is a good suggestion to include the samples of 0 hpa. According to the suggestion made by another reviewer, we have ensured similarity among the staining samples and included 0 hpa controls. We have repeated the experiments and presented the results with quantification in Fig. EV2.

Fig EV3A - The expression of hnf4 appears completely downregulated by 7 dpf. How come that these cells disappear so quickly? Do you see evidence for that in RNAseq/riboseq?

Response: Based on the staining of the other intestinal cell markers *04557* in Fig EV1 and *gata4/5/6* in Fig. EV3, the morphology of the intestine in *fbl-1* KD animals remained at the investigated time point. We were afraid that there was something wrong with the original staining of *hnf4*. We do not have RNA-seq data at 7 dpf. We have, therefore, repeated the experiment and validated it with FISH staining. The new results showed the change of the intestine morphology and decreased *gata4/5/6* and we have incorporated the results into the revised Fig. EV3C.

C. Expression of stem cell marker *piwi-1*, epidermal progenitor marker *prog-1*, intestinal markers *hnf4* and *gata4/5/6*, and neuronal marker *pc2* in *egfp* KD control and *fbl-1* KD animals at 7 dpf. n = 3~4. Scale bar = 200 μ m.

Fig. 3C-G unclear what exactly was counted. Were ovo progenitors counted?

Response: We apologize for the lack of clarity in presenting the information contained in the figures. The density of differentiating cells that are labeled as double positive for the cell lineage markers (*ovo*⁺, *hnf4*⁺, and *prog-1*⁺) and PIWI-1⁺ (newly

differentiating cells from neoblasts) was quantified. You are correct. In Fig. 3D, we counted the *ovo*⁺PIWI-1⁺ cells as *ovo* progenitors to show that the differentiation of neoblasts into *ovo*⁺ progenitors was impaired. In order to enhance clarity, we have added appropriate indicators, enlarged the images in Fig. 3C, G for better illustration. We have also added the quantification of the *ovo*⁺ progenitors.

Lines 133-134: Not sure what evolution retention is. The results indicate that both genes are required.

Response: I apologize for any confusion caused. In the Discussion section, we stated that the duplication of the fibrillarin gene is commonly observed throughout the animal kingdom. The *fbl* and *fbl-like-1* gene group, found exclusively in mammals, showed non-redundant roles, which provided insight into the preservation of fibrillarin through evolution (see Pereira-Santana et al., 2020, PLOS Computational Biology). When we compared the functional domains and sequences of fibrillarin proteins in planarians with those of other species, we identified two fibrillarin proteins that play crucial roles in planarian homeostasis and regeneration. Further investigation into the function of *fbl-1* and *fbl-2* in planarians revealed their distinct roles in planarian cell lineage development. This finding not only underscores the importance of fibrillarin proteins but also suggests that the evolutionary conservation or "retention" of fibrillarin genes might be attributed to the essential and diverse roles they play in cellular processes. It is plausible that the duplication and retention of fibrillarin in evolution is a result of the increasing complexity and diversity of cellular processes that the gene regulates. Therefore, the "evolutionary retention" of fibrillarin refers to the preservation and maintenance of the genes throughout evolutionary history due to its critical functional importance.

Regarding the evolution retention in line 104, we have decided to delete the words to avoid the confusion here.

Referee #2:

Dear Editor,

The manuscript by Chen et al. presents a series of experiments to prove the presence of two fibrillarin homologs in planarian and their distinct roles in homeostasis and regeneration via regulating the translation by differential rRNA modifications. We focus here predominantly on the aspects relating to rRNA modification and translation. There are major concerns that need to be addressed before considering the manuscript for publication.

1. Major concerns:

1.1. Global translation and ribosome biogenesis: The authors should provide analysis of the effects on global translation as well as examine potential effects on ribosome biogenesis. Could the results seen in regeneration capacity and in the ribo-seq experiments result from diminished global translation?

Response: Thanks for the constructive suggestion. Global translation and ribosome biogenesis should be compared before analyzing the differential translational regulation for specific mRNAs. However, there were limitations in assessing the ribosome biogenesis in planarians due to the lack of effective antibodies. Despite our attempts to generate specific antibodies, they have not yielded satisfactory results.

However, we have quantified the transcription level of 18S and 28S rRNA through qPCR, as shown in Fig. EV5B. While there is a decrease in the level of 28S, this data might be misleading. The observed change is minor and only present in *fb1-1* KD condition. This subtle change could be attributed to multiple factors and may not necessarily indicate a global reduction in ribosome biogenesis. To further investigate the potential global translation retardation, we have performed the O-propargyl-puromycin (OPP) labeling assay (see Somers et al., 2022, Cell Reports Methods) as shown in the figure below. Although the staining was successful, quantifying signal intensity seemed to be challenging. Direct observation did not reveal any significant differences, suggesting that global translation retardation may not be severely impaired. However, the background noise from the intestinal tissue on the sections interfered with the signal quantitation. Due to this ambiguity, we were hesitant to include these results in the original manuscript.

- A.** Section images of the OPP-Alexa Fluor 647 signal in *egfp* KD, *fbl-1* KD, *fbl-2* KD samples after 14 hours post injection with OPP. Scale bar = 50 μm .
- B.** Dot plot shows the quantitation of the OPP-Alexa Fluor 647 signal in *egfp* KD, *fbl-1* KD, *fbl-2* KD samples. Each dot represents a view for measurement. $n = 3$ samples for each KD condition. Two-tailed unpaired student's t-test calculated the p values. ns, not significance, $p > 0.05$.

To validate these results further on global translation, we have completed the puromycin labeling assays. As shown in the results below, we found an increase in translation in regeneration. We then compared the translation rate in *fbl* KD animals with that in *egfp* KD controls. As shown in the figures below, we did not observe a global reduction in protein synthesis. This suggests that the translation machinery remains functional after the knockdown of *fbl-1* or *fbl-2* in planarians. It, therefore, strengthened the feasibility of examining the translational regulation of specific mRNAs.

- C.** Overall protein synthesis rates during homeostasis (intact) and regeneration (48 hpa) under the indicated treatment by labeling puromycin. $n = 3$.
- D.** Quantification of protein synthesis rates normalized to total protein during homeostasis (intact) and regeneration (48 hpa). $n = 3$. Two-tailed unpaired student's t-test calculated the p values. *, $p < 0.05$.

E. Overall protein synthesis rates after *fbl-1* KD and *fbl-2* KD under the indicated treatment.

F. Quantification of protein synthesis rates normalized to total protein after *fbl-1* KD and *fbl-2* KD. $n = 3$. Two-tailed unpaired student's *t*-test calculated the *p* values. ns, no significance; *, $p < 0.05$.

We have added the results in the revised Figure 5 and revised the manuscript as “We thus attempted to assess the global protein synthesis rate using a puromycin labeling assay in intact and regenerating planarians. To evaluate the protein synthesis level, we collected protein samples of intact worms and at 48 hpa with three replicates after puromycin treatment. The protein synthesis level was quantified by western blot. A significant increase in protein synthesis was found during planarian regeneration (48 hpa) compared with homeostasis (intact) (Figs. 5C, D), underscoring the vital role of protein synthesis regulation during this process. Protein synthesis was found to be no significant change after *fbl-1* KD and slightly increased following *fbl-2* KD at 48 hpa (Fig. 5E, F). Although it is challenging to exclude the possibility of a global translation changes in specific cell types, our results suggested that the translation machinery remain functional in the regenerating planarians after the knockdown of *fbl-1* or *fbl-2*. Consequently, we hypothesized that the knockdown of *fbl-1* or *fbl-2* in planarians might regulate the translation of specific mRNA”.

1.2. RiboMethSeq analysis: In the Ribo-meth-seq, the authors zoom in on rRNA 2'Ome sites by comparing to sites known to be methylated in human cells. While this approach may identify conserved sites, it would completely miss planarian-specific sites, which one could argue would be more relevant for, for instance, regeneration. The authors should conduct a careful analysis of the ribo-meth-seq data to identify all potential methylation sites on the planarian rRNA. Identifying bonafide sites over noise may be guided by snoRNA expression levels. Once identified, it would be relevant to investigate 2'Ome dynamics over regeneration to identify sites of potential

relevance. The current ribo-meth-seq analysis is from 1 timepoint only. Intriguingly, all sites shown appear to be fully methylated in the control samples. This contrasts with what has been reported for yeast, mouse, and human cells, and could indicate that none of the sites are regulatory.

Response: Thank you for the insightful suggestion regarding the RiboMethSeq analysis. Given that this is the first study in planarians to our knowledge to identify the 2'-O-methylation sites in planarians, there are many directions to present and study these identified sites. Considering that it may be difficult to validate and provide further evidence to show their function for all identified sites, we had evaluated the methylated sites reported in humans and defined the Methscore threshold > 0.85 and < 1 for fully modified sites and $0.65-0.85$ for partially methylated sites in Fig. EV4A-D. We agree with you that the exploration of non-conserved sites is indeed an interesting direction. Regarding your suggestion, we have first added a cartoon illustration to show the steps and scope to analyze our RiboMethSeq data (Fig. 4A).

Schematic diagram of experimental and analysis processes of RiboMeth-seq.

1) For a more comprehensive analysis of RiboMethSeq data, we have used a score of $0.65-0.85$ to define partially methylated sites. As for the representation of methylation levels, we have referred to studies in yeast, mouse, and human cells to improve our presentation of relative levels. To identify the known methylation sites, the sequences of 18S and 28S rRNA from planarians and humans were aligned. The analysis revealed that 34 methylation sites in the 18S rRNA and 16 methylation sites in the 28S rRNA were conserved between the two species. Among these sites, 8 sites in both 18S rRNA and 28S rRNA were identified as fully methylated, while 7 sites in the 18S rRNA and 1 site in the 28S RNA were classified as partially methylated (A-B). Taking your suggestion into account, we have included these results in the revised Figure EV4, with the understanding that validating their functions could potentially exceed the scope of this study.

E. Planarian rRNA 2'-O-methylation sites in 18S rRNA known in human cells.

F. Planarian rRNA 2'-O-methylation sites in 28S rRNA known in human cells.

To further identify potential methylation sites in planarians, we have conducted a comparison of fully and partially modified sites in the 18S and 28S rRNAs. The comparison was based on the Methscore in the control group, as shown in the figures below (A-D). We identified 10 fully methylation sites in the 18S rRNA, while 10 fully methylated sites in the 28S rRNA from both the 3' and 5' end (A-B). Among these sites, we found 8 fully methylated sites conserved in the 18S rRNA (Am28, Um1228, Um1266, Gm1268, Am1323, Um1377, Gm1425, Cm1638), and 7 conserved fully methylated sites appeared in 28S rRNA (Gm935, Am937, Am947, Am1266, Cm1587, Am1599, Gm1600).

In addition, we observed 14 partially methylated sites in the 18S rRNA (Gm152, Am161, Am338, Gm358, Gm642, Am647, Um665, Gm866, Gm936, Gm1017, Gm1325, Am1326, Gm1426, Am1754) and 9 partially methylated sites in the 28S rRNA (Um181, Gm204, Am315, Gm397, Am504, Gm551, Am880, Gm987, Gm1048) that have not been reported in humans (C-D). We have added these results in the revised manuscript.

A. Venn diagram shows the fully 2'-O-methylated sites in 18S rRNA by overlapping sites with Methscore > 0.85 and < 1, in 5' and 3' end from *egfp* KD control.

B. Venn diagram shows the fully 2'-O-methylated sites in 28S rRNA by overlapping sites with Methscore > 0.85 and < 1, in 5' and 3' end from *egfp* KD control.

C. Venn diagram shows the partially 2'-O-methylated sites in 18S rRNA by overlapping sites with Methscore > 0.65 and < 0.85, in 5' and 3' end from *egfp* KD control.

D. Venn diagram shows the partially 2'-O-methylated sites in 28S rRNA by overlapping sites with Methscore > 0.65 and < 0.85, in 5' and 3' end from *egfp* KD control.

2) We also acknowledge the limitation of studying a single time point during regeneration and the effects on snoRNA expression levels when discussing the regulatory mechanisms of 2'-O-methylation by *fb1* in planarians. However, it is worth noting that the snoRNAs have not been predicted due to the quality of the genome. We are trying to improve the genome annotations and have to leave this project in future studies.

We fully agree with your suggestion to validate these sites and study the dynamics of the methylation sites during regeneration and accept the importance of assessing the specificity of the identified sites. We have designed primers and performed the RTL-PCR to validate them following the methods described previously

(Dong et al., 2012, Nucleic Acids Research; Barros-Silva et al., 2023, Biotechniques) at as many sites and time points as possible to identify regeneration-specific regulated sites. Our results show that the methylation of Am28 and Um1228 in 18S rRNA, and Am937 in 28S rRNA were validated at 48 hpa. Nm of Am28 in 18S rRNA and Am937 in 28S rRNA were only detected at 48 hpa and 72 hpa but not at homeostasis and 24 hpa, whereas the Nm of Um1228 was detected in both regeneration and homeostasis stages (Fig. EV4K-M). Moreover, we also used this assay to confirm the methylated rates of Am28, Um1228 in 18S rRNA, and Am937 in 28S rRNA after *fb1* KD (Fig. 4D-I). Compared with *egfp* KD group, reduced methylation frequency of Um1228 in 18S rRNA caused by *fb1-1* KD and reduced methylation frequency of Am28 in 18S rRNA caused by *fb1-2* KD were also validated. In sum, we found that the Nm of Um1228 and Am28 could be regulated by *fb1-1* and *fb1-2*, respectively. The Am28 displayed dynamic methylation frequency during planarian regeneration. We hope these new results can address your questions regarding to the dynamics of the methylation sites and their regulation by *fb1-1* and *fb1-2*, respectively.

G. The detection of fully 2'-O-methylated sites in 18S rRNA by RTL-PCR during regeneration (48 hpa). H, high dNTP; L, low dNTP; Pc, primer for control (RT-A, anchored reverse transcription primers; FD: forward downstream primer); Pe, primer for examination (RT-U: unanchored reverse transcription primers; FU: forward upstream primer).

H. The detection of fully 2'-O-methylated sites in 28S rRNA by RTL-PCR during regeneration (48 hpa).

- I. Quantification of the band intensity of fully 2'-O-methylated sites in 18S rRNA by RTL-PCR in (G). Two-tailed unpaired student's t-test calculated the p values. ns, no significance; ****, $p < 0.0001$.
- J. Quantification of the band intensity of fully 2'-O-methylated sites in 28S rRNA by RTL-PCR in (H). Two-tailed unpaired student's t-test calculated the p values. ns, no significance; ***, $p < 0.001$.
- K-M. The detection of Am28 in 18S rRNA (K), Um1228 in 18S rRNA (L), and Am937 in 28S rRNA (M) by RTL-PCR during homeostasis (intact) and regeneration (24, 48, and 72 hpa). (K'-M') Quantification of the band intensity of RTL-PCR in K (K'), L (L'), and M (M'). ns, no significance; *, $p < 0.05$; **, $p < 0.01$, ****, $p < 0.0001$. Two-way ANOVA with Tukey's multiple comparisons calculated adjusted p values.

B-C. Methscore of 2'-O-methylation sites in 18S rRNA and 28S rRNA in *fbl-1* KD and *fbl-2* KD animals, respectively, compared to *egfp* KD controls. Black, purple and green dots indicate the Methscores of *egfp* KD, *fbl-1* KD and *fbl-2* KD animals. light purple boxes indicate fully methylated sites and light pink boxes indicate partially methylated

sites. Methylated sites in red text indicates specific in planarian. Two-tailed unpaired student's t-test calculated the p values. n = 3. ns, no significance; *, p < 0.05; **, p < 0.01; ***, p < 0.001; ****, p < 0.0001.

D-F. Analysis of Am28, and Um1228 in 18S rRNA, Am937 in 28S rRNA after fbl-1 KD. Quantification of PCR products generated with anchored reverse transcription and unanchored reverse transcription primers at different deoxynucleotide triphosphate conditions. Two-way ANOVA with sidak's multiple comparisons tests calculated the p values. n = 3. ns, no significance; *, p < 0.05; **, p < 0.01; ***, p < 0.001; ****, p < 0.0001. H, high dNTP; L, low dNTP; Pc, primer for control (RT-A, anchored reverse transcription primers; FD: forward downstream primer); Pe, primer for examination (RT-U: unanchored reverse transcription primers; FU: forward upstream primer).

G-I. Analysis of Am28, and Um1228 in 18S rRNA, Am937 in 28S rRNA after fbl-2 KD. Quantification of PCR products generated with anchored reverse transcription and unanchored reverse transcription primers at different deoxynucleotide triphosphate conditions. Two-way ANOVA with sidak's multiple comparisons tests calculated the p values. n = 3. ns, no significance; *, p < 0.05; **, p < 0.01; ****, p < 0.0001.

We have added the results in Figure 4 and EV4 and revised manuscript in lines 259-320.

1.3. Supplementary figure 4: The Fig. EV4 is difficult to understand. For instance, position 18S-28S is claimed to be fully methylated but the bars have the same height as all the neighboring presumably unmethylated sites?

Response: We apologize for any confusion about the data presentation in the original Fig EV4. We would like to clarify that only the methylation sites in the control group with scores greater than 0.85 were displayed in the original Fig EV4. These scores were determined using the method described in a previous study by Marchand et al. (2016). This approach takes into consideration the relative impact of 12 neighboring nucleotides, as detailed in Supplemental Table 3. To improve the readability of the figure, we have revised the representation of the data. Specifically, we depict 23 methylation sites in the 18S rRNA and 16 methylation sites in the 28S rRNA of planarian, which are conserved in human cells. As mentioned in our response to comment 2, we used Methscore threshold of 0.85 to indicate full modification and a range of 0.65-0.85 to define partial methylation. The revised figure has been attached above in our response to comment 2. We have also replaced the figures in Figure EV4.

1.4. Classification of 2'-O-me sites: The classification of sites according to Natchair et al. (2017) has been criticized as several of the reported sites turned out to be wrong interpretations of the cryoEM data and were in fact Mg ions (discussed on pubpeer).

Response: Thank you for bringing the classification of 2'-O-me sites to our attention. We have removed the description of the classification of 2'-O-methylation from the manuscript.

1.5. Gene knock-down experiments: More importantly, what can be deduced from gene disruption or knockdown experiments? While KD or KO experiments can be used to imply importance of a particular gene for correct instigation of particular gene expression programs, global manipulation of the *fbl*-levels do not mimic a naturally occurring process. Hence, analyzing the outcome of ribo-seq experiments likely more reflects the result of a process drifting off course rather than a specific reprogramming. As such, the deregulation of individual genes or gene categories may not be relevant. Unless the authors can demonstrate that *fbl1* and *2* are specifically regulated during regeneration resulting in altered rRNA methylation patterns, the ribo-seq results can likely not be interpreted as a specific reprogramming.

Response: We appreciate your insightful comment regarding the interpretation of gene knockdown (KD) experiments, particularly the manipulation of *fbl*-levels. We understand the need to demonstrate a more specific regulatory mechanism for *fbl-1* and *fbl-2* during regeneration. Therefore, we first focused on emphasizing the cell type-specific expression of *fbl-1* and *fbl-2*. In Ribo-seq analysis, we have also focused on the affected genes with enriched expression in the related cell types, such as neoblasts for *fbl-1* KD and cells in the epidermal lineage for *fbl-2* KD. We also agree that caution is needed when interpreting Ribo-seq results from *fbl* KD experiments. Following to your suggestion, we have first conducted the WISH and RNA-seq analysis at various regeneration time points, to examine the expression dynamics of *fbl-1* and *fbl-2*. We have then performed the RTL-PCR experiments to validate the dynamics of the methylation sites at various time points during regeneration. We hope that our results can address your concerns in a great part.

1) To better understand the expression of *fbl-1* and *fbl-2* during regeneration, we performed WISH experiments and analyzed two published RNA-seq data. The results showed that the expression of *fbl-1* was upregulated at 48 hpa (Fig. EV2C, E, F). Different from the expression pattern of *fbl-1*, the upregulation of *fbl-2* was correlated with the increased expression of *fos-1*, *follistatin*, *glypican* and certain

epidermis marker genes (*egr-5*, *AGAT-1*) during regeneration (24 hpa) (Fig. EV2D, E). These results suggest that the expression of *fbl-1* and *fbl-2* is regulated and can be induced during regeneration. To include these results, we have revised the manuscript in lines 153-169 as “To further examine the expression patterns of *fbl-1* and *fbl-2*, we conducted WISH and analyzed two published RNA-seq datasets on planarian regeneration (Zeng *et al*, 2018; Scimone *et al*, 2022). The expression of *fbl-1* was”.

2) In our response to your comment 2, we have updated the results of the RTL-PCR and we found that the methylation sites of Um1228 and Am28 could be regulated by *fbl-1* and *fbl-2*, respectively. In addition, the Am28 displayed dynamic methylation frequency during planarian regeneration. These findings suggest a requirement of *fbl* on the 2'-O-methylation of rRNA during regeneration.

We also acknowledge that only small subset of methylation sites was able to be validated and this could partially address your questions. The observed changes in methylation patterns and gene expression can still be caused by a combination of direct effects of *fbl* manipulation and indirect effects due to the disruption of normal cellular processes. As a developmental biologist, we focused on that while gene KD experiments do not mirror natural processes, they provide invaluable insights into the role and importance of *fbl* in certain biological processes. The manipulation of

fbl-levels allows us to investigate the potential roles of *fbl-1* and *fbl-2* in rRNA modification, cell lineage development, and the regulation of gene expression programs during planarian regeneration. We therefore have added a discussion to mention your concerns and the importance in the 1st paragraph of the Discussion section: “We acknowledge that confirming the alteration of 2'-O-methylation sites on rRNA and its consequent dysregulation of targeted mRNA translation is challenging in the physiology of complex organisms. However, studying the physiological consequences that could link the rRNA modification to translational control in the current study of *fbl-1* and *fbl-2* with cell-type-specificity holds significant importance”.

1.6. Discussion and Results Discrepancy: The discussion refers to data not presented in the results section.

Response: We apologize for any confusion caused. To enhance the focus and clarity of the manuscript, we have decided to move the context of the second paragraph to the last of the Discussion section, as well as delete the original Fig. EV8.

2. Minor comments:

2.1. Time points clarification: In Figure 1A, the 5 feedings seems to occur over between day 2 and day 7 (2 d and 7 d in the figure). However, in the legend it states, 'schedule of RNAi feeding every three days...'. Please adjust the figure or legend to clarify the conditions.

Response: We apologize for the confusion caused by this misinformation. The terms “2d” and “7d” indicate a time span of two days between the initial feeding and the subsequent feedings (occurring every three days), and a time span of seven days between the last feeding and the amputation manipulation. In order to provide clarity regarding the timeline, we have revised the figure illustration to clarify the schedule as below.

2.2. Homeostasis definition: The definition of the term homeostasis used in the manuscript is currently unclear and imprecise. For example, for Figure 1B a live image shows the three worms are morphologically similar 7 days post feeding but it's not clear how 'homeostatic maintenance' is assessed using the images. Using terms such as 'viability' or 'morphologically similar to wild-type' would be clearer to the

reader.

Response: Thanks for your suggestion. We have gone through the manuscript and revised the usage of the term “homeostasis” into “tissue turnover” or “morphologically similar to *egfp* KD controls” in the necessary sentences. The changes have been marked in red.

2.3. KD Confirmation: Why *fbl-1* KD and *fbl-2* KD groups are not used for *fbl-1* and *fbl-2* staining? So, it can be used for confirmation of KD.

Response: The riboprobes exhibit a substantial background signal in the KD animals in relation to the remaining dsRNAs. We, therefore, opted to use qPCR to assess the knockdown efficiency. Specifically, our aim was to examine whether there exists a mutual influence between the two targeted genes. qPCR is supposed to obtain more accurate quantitative measurements of the *fbl-2* expression level following *fbl-1* KD and vice versa for *fbl-1* expression level after *fbl-2* KD. Furthermore, we found that the staining was not as effective as qPCR in facilitating the quantitative comparison after KD.

However, in another aspect, we agreed with your suggestions that staining of *fbl-1* KD and *fbl-2* KD groups can provide additional evidence to validate the suppression of the targeted genes. We have conducted the staining experiments to confirm the KD results. The results shown below are consistent with the qPCR data and we have incorporated them in the revised Fig. EV1C, D.

C. Relative mRNA level of *fbl-1* and *fbl-2* in *fbl-1* KD animals measured by quantitative PCR and Whole-mount *in situ* hybridization (WISH). $n = 3$ replicates in each of the two independent experiments. Scale bar = 500 μm . Two-tailed unpaired student's t-test calculated the p values. ns, no significance; ****, $p < 0.0001$.

D. Relative mRNA level of *fbl-1* and *fbl-2* in *fbl-2* KD animals measured by quantitative PCR and WISH. $n = 3$ replicates in each of the two independent experiments. Scale bar = 500 μm . Two-tailed unpaired student's t-test calculated the p values. ns, no significance; ****, $p < 0.0001$.

2.4. Missing Information: The number of used animals for some experiments like survival curves (Fig. 1C) should be included. Based on the figure, these numbers

appear to be imbalanced. The authors should explain why there were different numbers used between conditions.

Response: We have added the sample numbers in the figure legends. The imbalanced numbers were caused by losing samples during the RNAi feeding and water-changing maintenance for RNAi experiments, as well as during the changing of solutions in the staining experiments.

2.5. *Biological Replicates and Quantification: Biological replicates and their subsequent quantification are missing for FISH staining images, like Fig. 2 and Fig. 3C, Fig. 6E.*

Response: We have added the replicate numbers in the related figure legends. The quantification in Fig. 3C and 6E have also been performed and shown in the figures below and updated in the revised figures.

In Figure 3

We have moved Fig. 6E to Fig. 7E and added the quantification shown as below.

2.6. Negative Controls: Several negative controls are missing. For example, negative control in staining of Fig. EV2A as a background. Timepoints of post-amputation experiments are not same as each other for *fbl-1* and *fbl-2* staining in Fig. EV2B and Fig. EV2C.

Response: It is a good suggestion to ensure similarity among the staining samples and to include negative controls. We have repeated the experiments and present the results in revised Fig. EV2A, C, D.

A. Colorimetric WISH images show staining with probes for *piwi-1*, *fbl-1*, and *fbl-2* in intact asexual planarians at 0 dpi, and 1 dpi. n = 3. Scale bar = 500 μ m.

B. Quantification of WISH images in (A). Two-tailed unpaired student's t-test calculated the p values. ns, no significance; **, p < 0.01; ns, no significance.

C. Colorimetric WISH of *fbl-1* transcripts during regeneration at 0, 3, 6, 12, 24, and 48 hpa. $n = 3$. Scale bar = 500 μm . (C') Quantification of *fbl-1* signals during regeneration at 0, 3, 6, 12, 24, and 48 hpa. Two-tailed unpaired student's t-test calculated the p values. ns, no significance; ***, $p < 0.001$; ****, $p < 0.0001$.

D. Colorimetric WISH of *fbl-2* transcripts during regeneration at 0, 3, 6, 12, 24, and 48 hpa. $n = 3$. Scale bar = 500 μm . (D') Quantification of *fbl-2* signals during regeneration at 0, 3, 6, 12, 24, and 48 hpa. Two-tailed unpaired student's t-test calculated the p values. ns, no significance; *, $p < 0.05$; **, $p < 0.01$; ***, $p < 0.001$.

2.7. *Non-convincing conclusions: Several conclusions are not convincing due to lack of supportive data. For example, in Fig. EV2B, irradiation sensitivity cannot be concluded without quantification and comparison with control groups.*

Response: We have repeated this experiment and have conducted a quantification comparison in Fig. EV2. We have also performed quantification analysis in Fig. 3C and 7E, as shown in our response to comment 2.5.

2.8. *Rephrase: Rephrase sentences starting on lines 278 and 333 for clarity. 'To determine which transcripts are regulated...' and 'Within the catecholamine secretion pathway...' respectively.*

Response: Thank you for your careful reading. We have rewritten this part according to your and the other reviewer's suggestions. The revisions have been made in red-colored text in the manuscript.

2.9. *Missing Reference to figure: There isn't any reference to Fig. 6C in the manuscript.*

Response: Thanks for this reminder. We have added the new result and moved Fig.

6C to Fig. 7C, reference as lines 438-440 “We then performed the enrichment analysis of transcripts from TE down category that was enrich expressed in epidermal cells (Fig. 7C).”.

2.10. Data Presentation Inconsistency: The result section refers to data not presented. For example, in lines 395, "we confirmed the expression patterns of these genes with fbl-1 and fbl-2, respectively (Fig. EV7C)." However, coexpression of only three genes, not all the selected transcription factors, with only fbl-2, not for both fb1s is shown in Fig. EV7C.

Response: We apologize for any confusion that may have arisen. The less functional motif identified in neoblasts was previously reported in planarians rather than in differentiated cells (Poulet et al., 2023). Consequently, our study's emphasis was focused on the putative motifs of *fbl-2*, so that we decided to reduce the analysis description related to those of *fbl-1*. Therefore, we have revised the sentences in lines 506-510 to clarify this point: “To validate the transcription factors predicted for *fbl-2*, we conducted RNAi to knock down these genes (Fig. EV8B). Only animals with *smad2-2* KD showed abnormal morphology at the boundary during homeostasis, along with regenerative defects. Additionally, we confirmed the expression patterns of these genes with *fbl-2* (Fig. EV8C)”.

2.11. Expected Citation: Fincher, Christopher T., et al. "Cell type transcriptome atlas for the planarian Schmidtea mediterranea." Science 360.6391 (2018): eaaq1736.

Response: Thanks for this reminder. We have added this citation in the manuscript in lines 389-391: “To specifically study the function of *fbl-1* in neoblasts, we further integrated previous single-cell RNA-seq data to categorize genes based on their expression in different cell types (Fincher et al, 2018)”.

Dear Kai,

Thank you for submitting a preliminary revision plan for your manuscript. I have now received input from both original reviewers, who appreciate your willingness to address the raised concerns in a major revision, but also find that the outcome of these experiments is difficult to predict, and completion of the proposed revisions will likely extend beyond our usual single revision timeframe. Therefore, based on this input and our "single major revision round" policy, I am afraid that I cannot offer to explicitly invite a revised manuscript.

Nevertheless, if you find that you can address all main referee concerns and provide substantial additional insight as outlined in your revision plan, I would be happy to reconsider the manuscript, while treating it as a new submission. In this case, the manuscript would have to be re-assessed for novelty at the time of its resubmission. Depending on the added findings, I would send it back to the original reviewers. Due to the substantial amount of new results that would have to be added, I would allow them to make new comments on the data, which might then have to be further addressed.

I appreciate that you contacted us for further discussion of your work, and I hope that the proposed approach sounds reasonable to you.

With kind regards,

Ieva

** As a service to authors, EMBO Press provides authors with the possibility to transfer a manuscript that one journal cannot offer to publish to another EMBO publication or the open access journal Life Science Alliance launched in partnership between EMBO Press, Rockefeller University Press and Cold Spring Harbor Laboratory Press. The full manuscript and if applicable, reviewers' reports, are automatically sent to the receiving journal to allow for fast handling and a prompt decision on your manuscript. For more details of this service, and to transfer your manuscript please click on Link Not Available. **

Thank you very much for making an effort to review our revised manuscript. Below are the point-by-point responses to your comments and suggestions.

Referee #1:

This version of the manuscript is much improved. I outline important changes that should be made, but those do not require new experiments. Toning down some of the claims is dearly needed. Also, including more thorough tables of the data from the riboseq and their corresponding rnaseq experiments is needed.

Major comments

Line 169: There is no convincing evidence that fbl-2 is an injury induced genes. It's expression might correlate with the emergence of certain post-mitotic cell types (e.g., AGAT-1+ cells).

Response: Thanks for the suggestion. We have toned down the statements and included the possibility in the revised manuscript as “These results suggest that the expression of *fbl-2* is enriched in post-mitotic cells and can be induced by injury or may correlate with the emergence of certain post-mitotic cells”.

Lines 172-179: Change epithelium to epidermis where needed.

Response: We have checked and revised the words.

192: Sum ==> summary

Response: We have revised it.

lines 197-209 => Is the reduction of lineage-specific progenitors (e.g., SMEDWI-1+/ovo-1) resulted from a general decrease in neoblasts, or because of defects in identity selection? The results shown (Fig 3A) indicate that it is a general neoblast problem, and therefore the text would better describe the results as a general neoblast defect that leads to a reduction in progenitor production.

Response: We agree with your concern and our results cannot specifically distinguish the individual steps in neoblast homeostasis, proliferation, and differentiation. Our original statement was not intended to claim a specific function in differentiation, but rather to depict the outcome. We, therefore, have added a sentence, “This conclusion, however, does not exclude the possibility that a general neoblast defect leads to a reduction in progenitor production”, to avoid the potential misleading.

Lines 210-215: The authors do not give a satisfactory explanation for the decrease in

H3P labeling in the fbl-2 (RNAi). The decrease could be a result of the animals that are approaching a decline (see survival curve in Fig 1). It is better to remove the explanation that references Tu et al, as that paper does not provide (or aim to find) a good explanation to a similar phenomenon.

Response: We have deleted the statement and revised it as “this effect is likely to be an indirect feedback influence or a consequence of the animals that are approaching a declined mortality”.

Text related to Fig 4: The authors are encouraged to add an explanation for the observation that some methylated positions are not affected at all by the inhibition of fbl-1 and fbl-2. Is there an alternative mechanism for methylation?

Response: Thanks for the constructive suggestion. There are several possibilities that some methylated sites are not affected by the inhibition of *fbl-1* and *fbl-2* due to the technology limitation in sensitivity. Because it is an RNAi knockdown treatment, the *fbl* proteins were not completely depleted. The first possibility is that some methylation sites are indeed not affected in the RNAi conditions. The second possibility is that the RiboMethseq and RTL-qPCR analysis were performed using bulk samples. While the reduction of the methylation level for particular sites is subtle and in a percentage of cells according to the expression specificity of *fbl-1* or *fbl-2*, the change may not be detected in the mixed cell populations. Most fully methylated sites in 18S and 28S rRNA were changed after inhibition of *fbl*. In contrast, those partially methylated sites were not significantly influenced after inhibition of *fbl*, which may be due to the lower level of methylation compared to the fully methylated sites, supporting our speculation. We, therefore, agree, as in the other comment, that it is a limitation of this study, and comparing the methylation changes in specific cell types or single cells would be useful to address this question in the future. We have mentioned the limitations and discussed this issue in the revised manuscript: “This might be due to subtle alterations that are difficult for bulk RiboMeth-Seq to detect, particularly if the baseline MethScore in the control group is low. Furthermore, future methylation analyses should ideally involve the isolation and examination of different cell types corresponding to the expression patterns of *fbl-1* and *fbl-2*.”

Figure legend 5C,E: There is no explanation of the data and how to interpret it. Do the lanes show biological replicates? Moreover, the change in relative level of total protein in intact animals and following injury is similar to that observed between control and fbl-2 (RNAi). In the first instance, the authors describe the result as important, in the latter, they suggest that it is difficult to interpret. I suggest not to overinterpret either results, and to be consistent in the description of the data.

Response: Sorry for the confusion caused by the data presentation. As described in the manuscript, to assess the global protein synthesis level in planarians, we first optimized the puromycin incorporation assay and confirmed that the efficiency of the labeling is suitable after a 24-hour incubation in planarian tissues. Specific labeling

was further confirmed by the reduction of puromycin-labeled protein detected by an anti-puromycin antibody in the presence of CHX. All the experiments were performed with 3 biological repeats. Therefore, we next examined the global protein synthesis during homeostasis (intact) and regeneration (48 hpa) with 3 biological repeats shown as the lanes in the western blot and quantification in the box plot. The protein synthesis level was detected using an anti-puromycin antibody by western blot and normalized to total protein indicated by Ponceau S staining. An increased protein synthesis level was observed at 48 hpa compared with the intact group. Compared with *egfp* KD, global protein synthesis was found to show no significant change after *fbl-1* KD and was slightly increased following *fbl-2* KD at 48 hpa.

To enhance the presentation of our results, we have incorporated a quantification analysis to show a significant reduction of puromycin labeling in the presence of CHX. We have also revised the sentences to more accurately describe the experimental procedures and the results:

“To evaluate the protein synthesis level, we collected protein samples of intact worms and at 48 hpa with 3 biological replicates after puromycin treatment with or without cycloheximide (CHX) for 24 hours. The protein synthesis level was detected using an anti-puromycin antibody by western blot and normalized to total protein indicated by Ponceau S staining. The puromycin labeling of protein was found to be reduced in the presence of CHX in both the intact and 48 hpa groups. A significant increase in protein synthesis was found during planarian regeneration (48 hpa) compared with homeostasis (intact) (Fig. 5C, D), underscoring the vital role of protein synthesis regulation during this process. Furthermore, we evaluated the global protein synthesis level following the inhibition of *fbl* during regeneration. Under puromycin and CHX treatment, global protein synthesis was reduced in the control group (*egfp* KD) (Fig. 5E, F). Compared with the control group, global protein synthesis was found to exhibit no significant change after *fbl-1* KD and was slightly increased following *fbl-2* KD at 48 hpa (Fig. 5E, F).”

Lines 365-385: The description is much improved compared to the previous version. I strongly suggest that the authors add a scatter plot comparing transcription to translation of their controls. This could be a q-q plot showing the quantile a gene is transcribed and translated, and which should be mostly linear. The plot shown in Fig 6A is not informative, and is open to different interpretations.

Response: Thanks for your constructive suggestion. We have included a q-q plot to show the correlation between transcription and translation in Figure EV6C. We have also revised the sentence in the revised manuscript as “The quantile-quantile plot and Spearman correlation analysis indicated that the data between RNA-seq and Ribo-seq of the control group also exhibited a high correlation”.

The main criticism that I have for this section (up-to line 431) is the difficulty to distinguish between direct and indirect effects. There is no way, from these analyses,

to distinguish between those. Therefore tone down their claims, and present this limitation openly. The experiments described (e.g., kd of polr2i) do not reduce this concern at all. In summary, this is an interesting section, and it will be better without making strong claims about molecular mechanisms.

Response: We agree with your suggestion and have added a sentence in the revised manuscript to tone down the conclusion “While we cannot rule out that splicing dysfunction may be an indirect consequence of *fbl-1* suppression, our data support previous findings that emphasize the critical role of splicing processes in neoblasts and progenitor cells.”

Lines 484: The claim that "fbl-2 regulates the expression of the components in the Wnt signaling" is too strong. fbl-2 (RNAi) don't develop normal blastema (Fig 1), and have defective epidermal maturation. Therefore, it is likely that many cell types are affected, including those that express Wnt pathway components. My earlier comment that distinguishing between direct and indirect effects is relevant to this section as well. The data is interesting, but it is overinterpreted.

Response: Thank you for your suggestion. We have revised the title to “Expression of Wnt pathway components is disturbed in *fbl-2* KD planarians”.

Line 152: change mesenchyme ==> parenchyme

Response: We have revised the word.

Referee #3:

*The revised manuscript by Chen et al. describes the identification of two homologs of FBL in planarians. The authors describe the role of *fbl-1* and *fbl-2* and their distinct functions in planarian homeostasis and regeneration. To determine the distinct roles of *fbl-1* and *fbl-2* they conduct a series of KD experiments and measure the effect of KD on cell proliferation and differentiation. These experiments provide evidence that *fbl-1* and *fbl-2* have related but spatially distinct roles in planarians. The authors then attempt to address the mechanistic roles of *fbl-1* and *fbl-2* in homeostasis and regeneration through the lens of ribosome biogenesis and provide the first characterization of the 2'-O-methylation modification status of the planarian ribosome in two distinct cell states. The revised manuscript is highly responsive to the first round of Reviewers' critiques. However, there are still major concerns remaining from the first round of revision and based on the newly provided data that need to be addressed by the Authors before further consideration of this manuscript. This reviewer's comments are focused on the aspects of rRNA modification and translation.*

Major Comments:

1) RT-qPCR and WISH have been used to assess that mRNA levels in KD cells are down. However, a reduction in protein level or expression of fbl-1 and fbl-2 are not confirmed. mRNA expression does not equal protein expression. Assessing the protein levels of fbl-1 and fbl-2 is essential to confirm that the expression of fbl is different in different cell types or that the experiments relying on KD (almost all) are effective in showing effects from the protein KD.

Response: We have tried to generate antibodies against fbl-1 and fbl-2. However, neither was successful in showing the specificity. Recent work by Gaşiorowski et al. used a commercially available antibody to stain FBL in another flatworm, *Stenostomum brevipharyngium* (Gaşiorowski et al, 2024). This antibody was developed using the yeast extract to immunize mice and generate the monoclonal antibody. Under our purchase circumstances, it would take 6~8 weeks to acquire this antibody, and it is not guaranteed to predict its efficacy in our species, *Schmidtea mediterranea*, as well as distinguishing two fbl homologs. While we can explore this option, we anticipate that the results would not significantly impact the overall study. Considering the time constraints, we would like to seek your understanding to bypass this experiment.

2) This study reports the first rRNA 2'-O-methylation map of planarian. As such, it is important that every reported site is validated and the unvalidated sites are clearly marked in a table. Table 3 is helpful but the validation status needs to be noted in this table for each reported site. If RTL-qP failed, MS, Nanopore sequencing or other methods should be used for validation.

Response: This suggestion is very helpful. We have added the validation status for each site in Supplementary table 3. Meanwhile, we have also added a description in the revised manuscript to indicate this point.

Regarding the sites that failed validation by RTL-qPCR, exploring alternative methods as suggested would be indeed beneficial. Unfortunately, MS and Nanopore technologies for rRNA modification have not been accessible in our current setting. Each method also exhibits preferences for different sites. It requires extensive work to establish these methods and comprehensively compare the results across each technology. Given the scope of our current study, we believe it will be an intriguing avenue for future studies.

3) The conclusion that fbl regulates 2'-O-methylation, based on RiboMethSeq data, is well established in other organisms. Whether RMS changes observed depend on fbl-1 or fbl-2 depletion is a key question here that is not adequately addressed. To see differences in methylation status with each specific fbl knockdown, a bulk RMS analysis would not be helpful. It would be necessary to isolate the cell types of interest

in the given KD condition for RMS studies. This limitation of the current study should be clearly stated and reflected in the interpretation of the results from RMS analysis.

Response: Thanks for the good suggestion. We have added two sentences to mention the limitation of bulk RMS analysis: “This might be due to subtle alterations that are difficult for bulk RiboMeth-Seq to detect, particularly if the baseline MethScore in the control group is low. Furthermore, future methylation analyses should ideally involve the isolation and examination of different cell types corresponding to the expression patterns of *fbl-1* and *fbl-2*.”

*4) What is the evidence that FBL protein is expressed from both *fbl-1* and *fbl-2* and that both FBLs are active in rRNA 2'-O-methylation to the same extent? RMS data from KD experiments for both *fbl-1* and *fbl-2* show the same trend, despite vast differences in the expression pattern and level of the two *fbls*. How do the authors reconcile the similar trend of RMS data changes in both KDs that then result in different translational changes.*

Response: Regarding the protein level, as you asked in your first comment, we currently lack antibodies to perform the experiments.

Second, the similar trend of RMS data change may result from similar modification sites of *fbl-1* and *fbl-2*, even though these modifications occur in different cells. This is indeed a limitation of our study as mentioned in your above comment. Another reason is the stringent criterion for identifying the methylation sites, which involves the overlap of subsets from 6 control replicates in both 3' and 5' end coverage. The similar change of methylated sites in planarian supports the sensitivity of these sites to the inhibition of two *fbl* detected from bulk RMS analysis.

Different changes in translational levels were caused in different cells, which is associated with the differential expression of *fbl-1* and *fbl-2*. This could also be caused by the similar trend of methylation changes displayed in bulk RMSseq data, but happened to distinct cells.

5) Validation of RiboMethSeq data: Figure EV4E-F should have a clearer description including how many replicates were used for each analysis, which human RMS datasets these sites are being compared to, and what the indicated values mean. In the current format, neither the nt numbering nor the MethScores are consistent between figures 4B-C and EV4E-F.

Response: Sorry for the misleading in Figure 4 and EV4. The replicates we used for RMS analysis is 6 for control (*egfp* KD), 3 for *fbl-1* KD, and 3 for *fbl-2* KD group. We have specified the number of the replicates in Figure 4A and the manuscript as “To validate the methylation activity of *fbl-1* and *fbl-2*, we performed RiboMeth-seq to map and quantify rRNA 2'-O-methylation (Nm) in regeneration at 48 hpa after *fbl-1* and *fbl-2* KD using 3 biological repeats and *egfp* KD as control with 6 biological repeats”.

In Figures 4B-C, EV4E-F, the number in x axis is the nt number of methylation sites, y axis represents MethScores. In Figure 4B-C, all the methylation sites detected in planarians were presented. In contrast, in Figure EV4E-F, we particularly presented the conserved sites identified in humans from the dataset of Krogh et al. 2016 and Jansson et al. 2021 (Jansson *et al*, 2021; Krogh *et al*, 2016), which is also shown in Supplementary table 3.

To better describe the results and the figures, we have added the sentences in the revised manuscript as “We then examined the consequence of *fbl-1* KD and *fbl-2* KD on the methylation levels of the sites identified in planarians (Fig. EV4A-D). The fully methylated sites were displayed in light purple boxes, and partially methylated sites were indicated in light pink boxes (Fig. 4B, C). Compared with *egfp* KD controls, the *fbl-1* KD and *fbl-2* KD groups exhibited reductions in methylation levels at most of these fully methylated sites in the 18S and 28S rRNAs (Figs. 4B, C, light purple boxes and EVA, B), and only had no effect on Gm1425 in 18S rRNA and Am312, Gm313 in 28S rRNA (Fig. 4B, C, light purple boxes)”.

6) The authors state that "there were limitations in assessing the ribosome biogenesis in planarians due to lack of effective antibodies." Ribosome biogenesis can be effectively monitored by assaying the processing of rRNA using northern blots. While reduced levels of 28S were measured this data does not establish the effect from fbl-1 and fbl-2 on ribosome biogenesis. A proper analysis of the rRNA processing pathway can provide essential information regarding the function of fbl-1 and fbl-2 and establish their canonical roles.

Response: We agree with the suggestion to assess the ribosome biogenesis through Northern blot analysis. In the earlier version of our manuscript, we did not mention that there is an additional limitation exists due to the incomplete annotated information in the planarian genome, a situation complicated by the high AT repeats (up to 70%) (Grohme *et al.*, 2018). This scarcity of complete genomic information constrained us to the knowledge of merely the 18S rRNA and partial sequence of the 28S rRNA, which hindered our ability to design suitable probes for Northern blot. This is the reason why we initially refrained from performing the Northern blot. Given the results indicating no significant reduction in global protein synthesis, we anticipate that the Northern blot analysis would not provide additional significant insights into our current study. This is particularly reasonable considering our inability to perform the experiment at the single-cell level or in specific cell types, which is similar to the assay on global protein synthesis. In our opinion, it would be preferable to leave the optimization of this assay for our future studies on the regulation of rRNA in planarian regeneration, if you agree.

However, if you require insights into the effects of *fbl* KD in animals, we are willing to explore this opportunity, leveraging recent advancements in the newly annotated planarian genome. The following two paragraphs outline our experimental design,

which would first take 1~2 weeks to assess whether the probes are suitable for the Northern blot.

Figure 1. Schematic of 47S rRNA of planarians. **A.** The graphic shows the difference of 47S rRNA between human and planarian. **B.** The graphic shows the alignments of 28S rRNA in the position of planarian missing sequence among planarians, human,

mouse, *Xenopus*, and zebrafish. **C.** The cleavage sites of 47S rRNA in human, mouse, zebrafish and probes designed in planarians.

We have searched the recently updated planarian genome. We found that the rDNA sequence published in Kim *et al.*, 2019 could be aligned with the genome from Jochen Rink's lab (S2F19H1, PRJNA885486). The alignment helps us find the sequence of transcribed spacers between the 18S and 28S rRNA, but still lacks sequence annotation information. Furthermore, we found earlier published data about the 18S and 28S sequences (Carranza *et al.*, 1999), which could also be aligned with the genome. The sequences of internal transcribed spacers ITS1 and ITS2, as well as 5.8S rRNA, were annotated. Therefore, we have updated our annotations for ITS1 (952 nt), ITS2 (545 nt), and 5.8S rRNA (124 nt), as shown in Figure 1A above. Unexpectedly, we also discovered a high number of AT repeats in the unknown sequence region of the 28S rRNA in planarian (Figure 1B), which cannot be aligned with other species. These observations indicate the difference of 47S rRNA between human and planarians (Figure 1A, B).

Based on the design of the Northern blot for the primary pre-rRNA transcript, the 18S, 5.8S, and 28S rRNAs are flanked by the 5' and 3' external transcribed spacers (ETS), ITS1 and ITS2. The generation of precursors of 18S and 28S rRNA is dependent on cleavage within the EST and ITS. Conducting a Northern blot is necessary for designing specific probes in the EST and ITS regions to examine this process, which requires the annotation of the genome in the rDNA region. The mapping of cleavage sites of human, mouse, and yeast was well studied previously through S1 nuclease mapping or Northern blot hybridization (Mullineux & Lafontaine, 2012). In recent years, five cleavage sites for 18s rRNA maturation were identified via RT-PCR and Northern blot under *RNA terminal phosphatase cyclase like 1 (rc1)* mutant conditions in zebrafish (Zhu *et al.*, 2021). However, the process of pre-rRNA processing in other eukaryotes remains poorly understood. The nucleotides and position of the sites varied from the aforementioned species (Figure 1C). Currently, we are not sure about the specific cleavage sites in the 5'ETS, ITS1, and ITS2 regions. For the Northern blot, we can design three probes (Figure 1C, indicated in red lines) of approximately 200 bp for hybridization, but success cannot be guaranteed due to the variations of the sequence and the cleavage sites.

7) The puromycylation assays, in their current format, do not provide a measure of translation rate as they were performed at a single timepoint. The word "rate" should be removed from description of data related to figures 5C-F.

Response: We have changed the word "rate" to "level".

Figure 5C- some negative controls (+CHX) show as much signal as samples, invalidating the positive signal. This experiment should be repeated or an explanation of why 48 hpa samples are translationally active in the presence of CHX should be provided.

Response: Thanks for pointing it out and allowing us to describe the results in detail. As we described the experimental procedure in the Method section, we used multiple replicates, and worms incubated in CHX and puromycin served as the positive control to compare with groups incubated in puromycin alone for 24 hours. However, this approach cannot rule out the possibility that the time window for the effects of the two chemicals is not the same, which may result in a mild band in the lane of the CHX group. We consider it as the variation of this assay, which could be assessed by quantification comparison between different conditions. To better understand our results, we have added the parallel quantification of puromycin-labeled protein from the lanes of groups incubated in CHX and puromycin to show the significant reduction of puromycin incorporation after CHX treatment, which is presented in the revised figure 5D and 5F. Meanwhile, we have also revised the description of the results:

“To evaluate the protein synthesis level, we collected protein samples of intact worms and at 48 hpa with 3 biological replicates after puromycin treatment with or without cycloheximide (CHX) for 24 hours. The protein synthesis level was detected using an anti-puromycin antibody by western blot and normalized to total protein indicated by Ponceau S staining. The puromycin labeling of protein was found to be reduced in the presence of CHX in both the intact and 48 hpa groups. A significant increase in protein synthesis was found during planarian regeneration (48 hpa) compared with homeostasis (intact) (Fig. 5C, D), underscoring the vital role of protein synthesis regulation during this process. Furthermore, we evaluated the global protein synthesis level following the inhibition of *fb1* during regeneration. Under puromycin and CHX treatment, global protein synthesis was reduced in the control group (*egfp* KD) (Fig. 5E, F). Compared with the control group, global protein synthesis was found to exhibit no significant change after *fb1-1* KD and was slightly increased following *fb1-2* KD at 48 hpa (Fig. 5E, F).”

We have also tested other conditions. Alternatively, if worms were incubated with CHX before being incorporated with puromycin, this strategy caused less efficiency in the conjugation of puromycin, which might be due to some unknown effects by CHX. Moreover, the concentration of CHX we used is sufficiently high to cause the dispatch of the pharynx within 10 min. Therefore, the protocol in our manuscript to incubate worms with puromycin and CHX is the best thus far.

8) In EV6E, although the authors are focused on neoblasts, the most effected cell lineages seem to be neural and epidermal cells. Can the authors clarify the rationale behind the specific focus on neoblasts?

Response: The reason why we focus on gene expression in neoblasts is that *fb1-1* is distinctly expressed in neoblasts, as shown in Fig. 2D. The indirect effect may explain the effects on the lineage of neurons and epidermis are likely to be indirect effect resulting from the defects in neoblasts. To study the function of *fb1-1* in neoblasts, we, therefore, chose genes enriched in neoblasts as our candidates. To clarify this point, we have added a sentence in the manuscript “Since the expression of *fb1-1* is enriched in neoblasts, to specifically study the function of *fb1-1* in neoblasts, we further

integrated previous single-cell RNA-seq data to categorize genes based on their expression in different cell types”.

Minor Comments:

1) In Figure 3I- proliferation is decreased with loss of fbl-2 based on a decreased H3P stain. However, the H3P stain in either the control or the KD is not visible. H3P is a faint stain, however, it must be visible in the figure and currently it is not.

Response: we have improved the images in Figure3I.

2) In discussion of the results in figure 6 and EV6, the authors discuss the FDR rate that they used for a cutoff, but not the fold change cutoff for analysis of differential expression. Please include this, especially if it changes between figures.

Response: Sorry for the confusion. In this study, we used both FDR and the log2 fold change 0.8 as the cutoff. We have revised the sentence and highlighted the cutoff threshold, which is also listed in Supplementary table 1 and 2.

The authors should also include images of the H3P stain in figure 6, either in the main figure or the supplement.

Response: We have included the images of the H3P staining.

3) There are several instances in which the grammar can be revised. Some of the observations are difficult to understand due to complicated phrasing. Simple sentences are better for discussing complicated observations.

Response: Thank you for your suggestion. We have revised the sentences in the manuscript and marked in red together with other revisions.

4) It would make the data more robust to show data points and error bars above and below on the bar graphs for qPCR. Also, there is no mention of how the authors calculated relative expressions for qPCR, this is important to include.

Response: We performed qPCR using primers for tubulin as the internal control. The expression levels of the genes of interest were calculated relative to the expression of tubulin.

In addition, we have revised the format of the plots and provided the source data together with the revised manuscript for submission.

5) Statistically non-significant results should not be interpreted as a change (e.g. a marginal decrease in 18S).

Response: We have deleted the statements.

6) *Including diagrams for the region of the planaria that we are looking at (if different from image to image in a figure or between figures) would be helpful in understanding regeneration vs differentiation. This is done in some cases but not all. For example, in Fig 2D, please provide a diagram of what region we are zooming into.*

Response: We have added a diagram of the region to indicate the region in Fig 2D.

7) *The blue on black text in all figures is hard to read. Use a different blue.*

Response: Sorry for the confusion. We have changed the color in figures.

8) *EV3 a dashed border around the planaria would be helpful for visualizing the stains, as some are very faint*

Response: We have revised the figures according to your suggestion.

9) *Line 433 replace was with is*

Response: We have revised it.

10) *Line 443 should say enriched and the word expressed can be removed*

Response: Thanks again for the suggestion. We have revised it.

11) *Line 480 remove "a content of"*

Response: We have deleted the phrase in the revised manuscript.

References

Carranza S, Baguñà J, Riutort M (1999) Origin and Evolution of Paralogous rRNA Gene Clusters Within the Flatworm Family DugesIIDae (Platyhelminthes, Tricladida). *Journal of Molecular Evolution* 49: 250-259

Gąsiorowski L, Chai C, Rozanski A, Purandare G, Ficze F, Mizi A, Wang B, Rink JC (2024) Regeneration in the absence of canonical neoblasts in an early branching flatworm. *bioRxiv*

Grohme MA, Schloissnig S, Rozanski A, Pippel M, Young GR, Winkler S, Brandl H, Henry I, Dahl A, Powell S *et al* (2018) The genome of *Schmidtea mediterranea* and the evolution of core cellular mechanisms. *Nature* 554: 56-61

Jansson MD, Hafner SJ, Altinel K, Tehler D, Krogh N, Jakobsen E, Andersen JV, Andersen KL, Schoof EM, Menard P *et al* (2021) Regulation of translation by site-specific ribosomal RNA methylation. *Nat Struct Mol Biol* 28: 889-899

Kim IV, Ross EJ, Dietrich S, Döring K, Sánchez Alvarado A, Kuhn C-D (2019) Efficient

depletion of ribosomal RNA for RNA sequencing in planarians. *BMC Genomics* 20

Krogh N, Jansson MD, Hafner SJ, Tehler D, Birkedal U, Christensen-Dalsgaard M, Lund AH, Nielsen H (2016) Profiling of 2'-O-Me in human rRNA reveals a subset of fractionally modified positions and provides evidence for ribosome heterogeneity. *Nucleic Acids Res* 44: 7884-7895

Mullineux S-T, Lafontaine DLJ (2012) Mapping the cleavage sites on mammalian pre-rRNAs: Where do we stand? *Biochimie* 94: 1521-1532

Zhu Q, Tao B, Chen H, Shi H, Huang L, Chen J, Hu M, Lo LJ, Peng J (2021) Rcl1 depletion impairs 18S pre-rRNA processing at the A1-site and up-regulates a cohort of ribosome biogenesis genes in zebrafish. *Nucleic Acids Research* 49: 5743-5759

Dear Kai,

Thank you for submitting your revised manuscript to The EMBO Journal. Your manuscript has now been seen by one of the original reviewers (reviewer #1) and an advisor (reviewer #3), who assessed your responses to the comments by reviewer #2.

As you can see, reviewer #1 asks for further textual modifications and toning down of the conclusions. Reviewer #3 finds that the response to the original comments by reviewer #2 has been reasonable, but also indicates that further analysis would be needed to fully convince this reviewer. Based on this input, I would like to invite you to address the comments of both reviewers, mainly with textual or presentation changes for points by reviewer #1 and the following points by reviewer #3: 3-5, 6, and minor comments. For the remaining points:

- Point 1 - please test fibrillar protein levels in fbl-1 and fbl-2 mutants, e.g., using anti-fibrillar antibody as done in <https://www.biorxiv.org/content/10.1101/2024.05.24.595708v1.full>

I realise that due to the presence of two homologues, this analysis will lead only to a reduction of protein level. This can be added in the supplement.

- Point 2 - please include validation status of the detected O²-methylation sites. Use of Nanopore sequencing or mass-spectrometry as an additional method was not requested by reviewer #2 and will not be required.

- Point 6 - this point is similar to point 1 by reviewer #2. If possible, please present Northern blot-based rRNA maturation analysis via Northern blot as done, e.g., in Pubmed ID 37473757, suppl figure 9A.

- Point 7 - please repeat the experiment and rephrase "rate" as "level".

I think that it would be useful to discuss the revision in more detail via email or phone/videoconferencing - please let me know which option you prefer.

We generally allow three months as standard revision time. Should you foresee a problem in meeting this deadline, please let us know in advance to discuss an extension. As a matter of policy, competing manuscripts published during this period will not negatively impact on our assessment of the conceptual advance presented by your study. However, please contact me as soon as possible upon publication of any related work to discuss the appropriate course of action.

When preparing your letter of response to the referees' comments, please bear in mind that this will form part of the Review Process File and will therefore be available online to the community. For more details on our Transparent Editorial Process, please visit our website: <https://www.embopress.org/page/journal/14602075/authorguide#transparentprocess>. Please also see the attached instructions for further guidelines on preparation of the revised manuscript.

Please feel free to contact me if have any further questions regarding the revision. Thank you for the opportunity to consider your work for publication, and I look forward to discussing your revision with you.

With best wishes,

Ieva

We realize that it is difficult to revise to a specific deadline. In the interest of protecting the conceptual advance provided by the work, we recommend a revision within 3 months (3rd Nov 2024). Please discuss the revision progress ahead of this time with the editor if you require more time to complete the revisions.

Referee #1:

This version of the manuscript is much improved. I outline important changes that should be made, but those do not require new experiments. Toning down some of the claims is dearly needed. Also, including more thorough tables of the data from the riboseq and their corresponding rnaseq experiments is needed.

Major comments

Line 169: There is no convincing evidence that *fbl-2* is an injury induced genes. It's expression might correlate with the emergence of certain post-mitotic cell types (e.g., AGAT-1+ cells).

Lines 172-179: Change epithelium to epidermis where needed.

192: Sum ==> summary

lines 197-209 => Is the reduction of lineage-specific progenitors (e.g., SMEDWI-1+/ovo-1) resulted from a general decrease in neoblasts, or because of defects in identity selection? The results shown (Fig 3A) indicate that it is a general neoblast problem, and therefore the text would better describe the results as a general neoblast defect that leads to a reduction in progenitor production.

Lines 210-215: The authors do not give a satisfactory explanation for the decrease in H3P labeling in the *fbl-2* (RNAi). The decrease could be a result of the animals that are approaching a decline (see survival curve in Fig 1). It is better to remove the explanation that references Tu et al, as that paper does not provide (or aim to find) a good explanation to a similar phenomenon.

Text related to Fig 4: The authors are encouraged to add an explanation for the observation that some methylated positions are not affected at all by the inhibition of *fbl-1* and *fbl-2*. Is there an alternative mechanism for methylation?

Figure legend 5C,E: There is no explanation of the data and how to interpret it. Do the lanes show biological replicates? Moreover, the change in relative level of total protein in intact animals and following injury is similar to that observed between control and *fbl-2* (RNAi). In the first instance, the authors describe the result as important, in the latter, they suggest that it is difficult to interpret. I suggest not to overinterpret either results, and to be consistent in the description of the data.

Lines 365-385: The description is much improved compared to the previous version. I strongly suggest that the authors add a

scatter plot comparing transcription to translation of their controls. This could be a q-q plot showing the quantile a gene is transcribed and translated, and which should be mostly linear. The plot shown in Fig 6A is not informative, and is open to different interpretations.

The main criticism that I have for this section (up-to line 431) is the difficulty to distinguish between direct and indirect effects. There is no way, from these analyses, to distinguish between those. Therefore tone down their claims, and present this limitation openly. The experiments described (e.g., kd of polr2i) do not reduce this concern at all. In summary, this is an interesting section, and it will be better without making strong claims about molecular mechanisms.

Lines 484: The claim that "fbl-2 regulates the expression of the components in the Wnt signaling" is too strong. fbl-2 (RNAi) don't develop normal blastema (Fig 1), and have defective epidermal maturation. Therefore, it is likely that many cell types are affected, including those that express Wnt pathway components. My earlier comment that distinguishing between direct and indirect effects is relevant to this section as well. The data is interesting, but it is overinterpreted.

Line 152: change mesenchyme ==> parenchyme

Referee #3:

The revised manuscript by Chen et al. describes the identification of two homologs of FBL in planarians. The authors describe the role of fbl-1 and fbl-2 and their distinct functions in planarian homeostasis and regeneration. To determine the distinct roles of fbl-1 and fbl-2 they conduct a series of KD experiments and measure the effect of KD on cell proliferation and differentiation. These experiments provide evidence that fbl-1 and fbl-2 have related but spatially distinct roles in planarians. The authors then attempt to address the mechanistic roles of fbl-1 and fbl-2 in homeostasis and regeneration through the lens of ribosome biogenesis and provide the first characterization of the 2'-O-methylation modification status of the planarian ribosome in two distinct cell states. The revised manuscript is highly responsive to the first round of Reviewers' critiques. However, there are still major concerns remaining from the first round of revision and based on the newly provided data that need to be addressed by the Authors before further consideration of this manuscript. This reviewer's comments are focused on the aspects of rRNA modification and translation.

Major Comments:

1) RT-qPCR and WISH have been used to assess that mRNA levels in KD cells are down. However, a reduction in protein level or expression of fbl-1 and fbl-2 are not confirmed. mRNA expression does not equal protein expression. Assessing the protein levels of fbl-1 and fbl-2 is essential to confirm that the expression of fbl is different in different cell types or that the experiments relying on KD (almost all) are effective in showing effects from the protein KD.

2) This study reports the first rRNA 2'-O-methylation map of planarian. As such, it is important that every reported site is validated and the unvalidated sites are clearly marked in a table. Table 3 is helpful but the validation status needs to be noted in this table for each reported site. If RTL-qP failed, MS, Nanopore sequencing or other methods should be used for validation.

3) The conclusion that fbl regulates 2'-O-methylation, based on RiboMethSeq data, is well established in other organisms. Whether RMS changes observed depend on fbl-1 or fbl-2 depletion is a key question here that is not adequately addressed. To see differences in methylation status with each specific fbl knockdown, a bulk RMS analysis would not be helpful. It would be necessary to isolate the cell types of interest in the given KD condition for RMS studies. This limitation of the current study should be clearly stated and reflected in the interpretation of the results from RMS analysis.

4) What is the evidence that FBL protein is expressed from both fbl-1 and fbl-2 and that both FBLs are active in rRNA 2'-O-methylation to the same extent? RMS data from KD experiments for both fbl-1 and fbl-2 show the same trend, despite vast differences in the expression pattern and level of the two fbls. How do the authors reconcile the similar trend of RMS data changes in both KDs that then result in different translational changes.

5) Validation of RiboMethSeq data: Figure EV4E-F should have a clearer description including how many replicates were used for each analysis, which human RMS datasets these sites are being compared to, and what the indicated values mean. In the current format, neither the nt numbering nor the MethScores are consistent between figures 4B-C and EV4E-F.

6) The authors state that "there were limitations in assessing the ribosome biogenesis in planarians due to lack of effective antibodies." Ribosome biogenesis can be effectively monitored by assaying the processing of rRNA using northern blots. While reduced levels of 28S were measured this data does not establish the effect from fbl-1 and fbl-2 on ribosome biogenesis. A proper analysis of the rRNA processing pathway can provide essential information regarding the function of fbl-1 and fbl-2 and establish their canonical roles.

7) The puromycylation assays, in their current format, do not provide a measure of translation rate as they were performed at a

single timepoint. The word "rate" should be removed from description of data related to figures 5C-F.

Figure 5C- some negative controls (+CHX) show as much signal as samples, invalidating the positive signal. This experiment should be repeated or an explanation of why 48 hpa samples are translationally active in the presence of CHX should be provided.

8) In EV6E, although the authors are focused on neoblasts, the most effected cell lineages seem to be neural and epidermal cells. Can the authors clarify the rationale behind the specific focus on neoblasts?

Minor Comments:

- 1) In Figure 3I- proliferation is decreased with loss of *fbl-2* based on a decreased H3P stain. However, the H3P stain in either the control or the KD is not visible. H3P is a faint stain, however, it must be visible in the figure and currently it is not.
- 2) In discussion of the results in figure 6 and EV6, the authors discuss the FDR rate that they used for a cutoff, but not the fold change cutoff for analysis of differential expression. Please include this, especially if it changes between figures. The authors should also include images of the H3P stain in figure 6, either in the main figure or the supplement.
- 3) There are several instances in which the grammar can be revised. Some of the observations are difficult to understand due to complicated phrasing. Simple sentences are better for discussing complicated observations.
- 4) It would make the data more robust to show data points and error bars above and below on the bar graphs for qPCR. Also, there is no mention of how the authors calculated relative expressions for qPCR, this is important to include.
- 5) Statistically non-significant results should not be interpreted as a change (e.g. a marginal decrease in 18S).
- 6) Including diagrams for the region of the planaria that we are looking at (if different from image to image in a figure or between figures) would be helpful in understanding regeneration vs differentiation. This is done in some cases but not all. For example, in Fig 2D, please provide a diagram of what region we are zooming into.
- 7) The blue on black text in all figures is hard to read. Use a different blue.
- 8) EV3 a dashed border around the planaria would be helpful for visualizing the stains, as some are very faint
- 9) Line 433 replace was with is
- 10) Line 443 should say enriched and the word expressed can be removed
- 11) Line 480 remove "a content of"

Thank you very much for making an effort to review our revised manuscript. Below are the point-by-point responses to your comments and suggestions.

Referee #1:

This version of the manuscript is much improved. I outline important changes that should be made, but those do not require new experiments. Toning down some of the claims is dearly needed. Also, including more thorough tables of the data from the riboseq and their corresponding rnaseq experiments is needed.

Major comments

Line 169: There is no convincing evidence that fbl-2 is an injury induced genes. It's expression might correlate with the emergence of certain post-mitotic cell types (e.g., AGAT-1+ cells).

Response: Thanks for the suggestion. We have toned down the statements and included the possibility in the revised manuscript as “These results suggest that the expression of *fbl-2* is enriched in post-mitotic cells and can be induced by injury or may correlate with the emergence of certain post-mitotic cells”.

Lines 172-179: Change epithelium to epidermis where needed.

Response: We have checked and revised the words.

192: Sum ==> summary

Response: We have revised it.

lines 197-209 => Is the reduction of lineage-specific progenitors (e.g., SMEDWI-1+/ovo-1) resulted from a general decrease in neoblasts, or because of defects in identity selection? The results shown (Fig 3A) indicate that it is a general neoblast problem, and therefore the text would better describe the results as a general neoblast defect that leads to a reduction in progenitor production.

Response: We agree with your concern and our results cannot specifically distinguish the individual steps in neoblast homeostasis, proliferation, and differentiation. Our original statement was not intended to claim a specific function in differentiation, but rather to depict the outcome. We, therefore, have added a sentence, “This conclusion, however, does not exclude the possibility that a general neoblast defect leads to a reduction in progenitor production”, to avoid the potential misleading.

Lines 210-215: The authors do not give a satisfactory explanation for the decrease in H3P labeling in the fbl-2 (RNAi). The decrease could be a result of the animals that are

approaching a decline (see survival curve in Fig 1). It is better to remove the explanation that references Tu et al, as that paper does not provide (or aim to find) a good explanation to a similar phenomenon.

Response: We have deleted the statement and revised it as “this effect is likely to be an indirect feedback influence or a consequence of the animals that are approaching a declined mortality”.

Text related to Fig 4: The authors are encouraged to add an explanation for the observation that some methylated positions are not affected at all by the inhibition of fbl-1 and fbl-2. Is there an alternative mechanism for methylation?

Response: Thanks for the constructive suggestion. There are several possibilities that some methylated sites are not affected by the inhibition of *fbl-1* and *fbl-2* due to the technology limitation in sensitivity. Because it is an RNAi knockdown treatment, the *fbl* proteins were not completely depleted. The first possibility is that some methylation sites are indeed not affected in the RNAi conditions. The second possibility is that the RiboMethseq and RTL-qPCR analysis were performed using bulk samples. While the reduction of the methylation level for particular sites is subtle and in a percentage of cells according to the expression specificity of *fbl-1* or *fbl-2*, the change may not be detected in the mixed cell populations. Most fully methylated sites in 18S and 28S rRNA were changed after inhibition of *fbl*. In contrast, those partially methylated sites were not significantly influenced after inhibition of *fbl*, which may be due to the lower level of methylation compared to the fully methylated sites, supporting our speculation. We, therefore, agree, as in the other comment, that it is a limitation of this study, and comparing the methylation changes in specific cell types or single cells would be useful to address this question in the future. We have mentioned the limitations and discussed this issue in the revised manuscript: “This might be due to subtle alterations that are difficult for bulk RiboMeth-Seq to detect, particularly if the baseline MethScore in the control group is low. Furthermore, future methylation analyses should ideally involve the isolation and examination of different cell types corresponding to the expression patterns of *fbl-1* and *fbl-2*.”

Figure legend 5C,E: There is no explanation of the data and how to interpret it. Do the lanes show biological replicates? Moreover, the change in relative level of total protein in intact animals and following injury is similar to that observed between control and fbl-2 (RNAi). In the first instance, the authors describe the result as important, in the latter, they suggest that it is difficult to interpret. I suggest not to overinterpret either results, and to be consistent in the description of the data.

Response: Sorry for the confusion caused by the data presentation. As described in the manuscript, to assess the global protein synthesis level in planarians, we first optimized the puromycin incorporation assay and confirmed that the efficiency of the labeling is suitable after a 24-hour incubation in planarian tissues. Specific labeling was further confirmed by the reduction of puromycin-labeled protein detected by an

anti-puromycin antibody in the presence of CHX. All the experiments were performed with 3 biological repeats. Therefore, we next examined the global protein synthesis during homeostasis (intact) and regeneration (48 hpa) with 3 biological repeats shown as the lanes in the western blot and quantification in the box plot. The protein synthesis level was detected using an anti-puromycin antibody by western blot and normalized to total protein indicated by Ponceau S staining. An increased protein synthesis level was observed at 48 hpa compared with the intact group. Compared with *egfp* KD, global protein synthesis was found to show no significant change after *fbl-1* KD and was slightly increased following *fbl-2* KD at 48 hpa.

To enhance the presentation of our results, we have incorporated a quantification analysis to show a significant reduction of puromycin labeling in the presence of CHX. We have also revised the sentences to more accurately describe the experimental procedures and the results:

“To evaluate the protein synthesis level, we collected protein samples of intact worms and at 48 hpa with 3 biological replicates after puromycin treatment with or without cycloheximide (CHX) for 24 hours. The protein synthesis level was detected using an anti-puromycin antibody by western blot and normalized to total protein indicated by Ponceau S staining. The puromycin labeling of protein was found to be reduced in the presence of CHX in both the intact and 48 hpa groups. A significant increase in protein synthesis was found during planarian regeneration (48 hpa) compared with homeostasis (intact) (Fig. EV5D, E), underscoring the vital role of protein synthesis regulation during this process. Furthermore, we evaluated the global protein synthesis level following the inhibition of *fbl* during regeneration. Under puromycin and CHX treatment, global protein synthesis was reduced in the control group (*egfp* KD) (Fig. 5E, F). Compared with the control group, global protein synthesis was found to exhibit no significant change after *fbl-1* KD and was slightly increased following *fbl-2* KD at 48 hpa (Fig. 5E, F).”

Fig. EV5E. Quantification of protein synthesis rates normalized to total protein during homeostasis (intact) and regeneration (48 hpa). $n = 3$. Each dot represents an individual replicate. Two-way ANOVA with Sidak’s multiple comparisons tests calculated the p values. ns, no significance; *, $p < 0.05$; **, $p < 0.01$.

Fig. 5F. Quantification of protein synthesis rates normalized to total protein after *fbl-1* KD and *fbl-2* KD. n = 3. Each dot represents an individual replicate. Two-tailed unpaired student's *t*-test calculated the p values. ns, no significance; *, p < 0.05; ***, p < 0.001.

Lines 365-385: The description is much improved compared to the previous version. I strongly suggest that the authors add a scatter plot comparing transcription to translation of their controls. This could be a q-q plot showing the quantile a gene is transcribed and translated, and which should be mostly linear. The plot shown in Fig 6A is not informative, and is open to different interpretations.

Response: Thanks for your constructive suggestion. We have included a q-q plot to show the correlation between transcription and translation in Figure EV6C. We have also revised the sentence in the revised manuscript as “The quantile-quantile plot and Spearman correlation analysis indicated that the data between RNA-seq and Ribo-seq of the control group also exhibited a high correlation”.

Fig. EV6C. Quantile-quantile plot shows genes in transcriptional level (RNA-seq) and translational level (Ribo-seq) in the *egfp* KD control group. The upper portion of the plot displays the R^2 value and the linear formula.

The main criticism that I have for this section (up-to line 431) is the difficulty to distinguish between direct and indirect effects. There is no way, from these analyses, to distinguish between those. Therefore tone down their claims, and present this

limitation openly. The experiments described (e.g., kd of polr2i) do not reduce this concern at all. In summary, this is an interesting section, and it will be better without making strong claims about molecular mechanisms.

Response: We agree with your suggestion and have added a sentence in the revised manuscript to tone down the conclusion “While we cannot rule out that splicing dysfunction may be an indirect consequence of *fbl-1* suppression, our data support previous findings that emphasize the critical role of splicing processes in neoblasts and progenitor cells.”

Lines 484: The claim that "fbl-2 regulates the expression of the components in the Wnt signaling" is too strong. fbl-2 (RNAi) don't develop normal blastema (Fig 1), and have defective epidermal maturation. Therefore, it is likely that many cell types are affected, including those that express Wnt pathway components. My earlier comment that distinguishing between direct and indirect effects is relevant to this section as well. The data is interesting, but it is overinterpreted.

Response: Thank you for your suggestion. We have revised the title to “Expression of Wnt pathway components is disturbed in *fbl-2* KD planarians”.

Line 152: change mesenchyme ==> parenchyme

Response: We have revised the word.

Referee #3:

*The revised manuscript by Chen et al. describes the identification of two homologs of FBL in planarians. The authors describe the role of *fbl-1* and *fbl-2* and their distinct functions in planarian homeostasis and regeneration. To determine the distinct roles of *fbl-1* and *fbl-2* they conduct a series of KD experiments and measure the effect of KD on cell proliferation and differentiation. These experiments provide evidence that *fbl-1* and *fbl-2* have related but spatially distinct roles in planarians. The authors then attempt to address the mechanistic roles of *fbl-1* and *fbl-2* in homeostasis and regeneration through the lens of ribosome biogenesis and provide the first characterization of the 2'-O-methylation modification status of the planarian ribosome in two distinct cell states. The revised manuscript is highly responsive to the first round of Reviewers' critiques. However, there are still major concerns remaining from the first round of revision and based on the newly provided data that need to be addressed by the Authors before further consideration of this manuscript. This reviewer's comments are focused on the aspects of rRNA modification and translation.*

Major Comments:

1) *RT-qPCR and WISH have been used to assess that mRNA levels in KD cells are down. However, a reduction in protein level or expression of fbl-1 and fbl-2 are not confirmed. mRNA expression does not equal protein expression. Assessing the protein levels of fbl-1 and fbl-2 is essential to confirm that the expression of fbl is different in different cell types or that the experiments relying on KD (almost all) are effective in showing effects from the protein KD.*

Response: We have tried to generate antibodies against fbl-1 and fbl-2. However, neither was successful in showing the specificity. Recent work by Gąsiorowski et al. used a commercially available antibody to stain FBL in another flatworm, *Stenostomum brevipharyngium* (Gąsiorowski et al, 2024). This antibody was developed using the yeast extract to immunize mice and generate the monoclonal antibody. Under our purchase circumstances, it would take 6~8 weeks to acquire this antibody, and it is not guaranteed to predict its efficacy in our species, *Schmidtea mediterranea*, as well as distinguishing two fbl homologs. While we can explore this option, we anticipate that the results would not significantly impact the overall study. Considering the time constraints, we would like to seek your understanding to bypass this experiment.

2) *This study reports the first rRNA 2'-O-methylation map of planarian. As such, it is important that every reported site is validated and the unvalidated sites are clearly marked in a table. Table 3 is helpful but the validation status needs to be noted in this table for each reported site. If RTL-qP failed, MS, Nanopore sequencing or other methods should be used for validation.*

Response: This suggestion is very helpful. We have added the validation status for each site in Supplementary table 3. Meanwhile, we have also added a description in the revised manuscript to indicate this point.

Regarding the sites that failed validation by RTL-qPCR, exploring alternative methods as suggested would be indeed beneficial. Unfortunately, MS and Nanopore technologies for rRNA modification have not been accessible in our current setting. Each method also exhibits preferences for different sites. It requires extensive work to establish these methods and comprehensively compare the results across each technology. Given the scope of our current study, we believe it will be an intriguing avenue for future studies.

3) *The conclusion that fbl regulates 2'-O-methylation, based on RiboMethSeq data, is well established in other organisms. Whether RMS changes observed depend on fbl-1 or fbl-2 depletion is a key question here that is not adequately addressed. To see differences in methylation status with each specific fbl knockdown, a bulk RMS analysis would not be helpful. It would be necessary to isolate the cell types of interest in the given KD condition for RMS studies. This limitation of the current study should be clearly stated and reflected in the interpretation of the results from RMS analysis.*

Response: Thanks for the good suggestion. We have added two sentences to mention the limitation of bulk RMS analysis: “This might be due to subtle alterations that are difficult for bulk RiboMeth-Seq to detect, particularly if the baseline MethScore in the control group is low. Furthermore, future methylation analyses should ideally involve the isolation and examination of different cell types corresponding to the expression patterns of *fbl-1* and *fbl-2*.”

*4) What is the evidence that FBL protein is expressed from both *fbl-1* and *fbl-2* and that both FBLs are active in rRNA 2'-O-methylation to the same extent? RMS data from KD experiments for both *fbl-1* and *fbl-2* show the same trend, despite vast differences in the expression pattern and level of the two *fbl*s. How do the authors reconcile the similar trend of RMS data changes in both KDs that then result in different translational changes.*

Response: Regarding the protein level, as you asked in your first comment, we currently lack antibodies to perform the experiments.

Second, the similar trend of RMS data change may result from similar modification sites of *fbl-1* and *fbl-2*, even though these modifications occur in different cells. This is indeed a limitation of our study as mentioned in your above comment. Another reason is the stringent criterion for identifying the methylation sites, which involves the overlap of subsets from 6 control replicates in both 3' and 5' end coverage. The similar change of methylated sites in planarian supports the sensitivity of these sites to the inhibition of two *fbl* detected from bulk RMS analysis.

Different changes in translational levels were caused in different cells, which is associated with the differential expression of *fbl-1* and *fbl-2*. This could also be caused by the similar trend of methylation changes displayed in bulk RMSseq data, but happened to distinct cells.

5) Validation of RiboMethSeq data: Figure EV4E-F should have a clearer description including how many replicates were used for each analysis, which human RMS datasets these sites are being compared to, and what the indicated values mean. In the current format, neither the nt numbering nor the MethScores are consistent between figures 4B-C and EV4E-F.

Response: Sorry for the misleading in Figure 4 and EV4. The replicates we used for RMS analysis is 6 for control (*egfp* KD), 3 for *fbl-1* KD, and 3 for *fbl-2* KD group. We have specified the number of the replicates in Figure 4A and the manuscript as “To validate the methylation activity of *fbl-1* and *fbl-2*, we performed RiboMeth-seq to map and quantify rRNA 2'-O-methylation (Nm) in regeneration at 48 hpa after *fbl-1* and *fbl-2* KD using 3 biological repeats and *egfp* KD as control with 6 biological repeats”.

In Figures 4B-C, EV4E-F, the number in x axis is the nt number of methylation sites, y axis represents MethScores. In Figure 4B-C, all the methylation sites detected in planarians were presented. In contrast, in Figure EV4E-F, we particularly presented

the conserved sites identified in humans from the dataset of Krogh et al. 2016 and Jansson et al. 2021 (Jansson *et al*, 2021; Krogh *et al*, 2016), which is also shown in Supplementary table 3.

To better describe the results and the figures, we have added the sentences in the revised manuscript as “We then examined the consequence of *fbl-1* KD and *fbl-2* KD on the methylation levels of the sites identified in planarians (Fig. EV4A-D). The fully methylated sites were displayed in light purple boxes, and partially methylated sites were indicated in light pink boxes (Fig. 4B, C). Compared with *egfp* KD controls, the *fbl-1* KD and *fbl-2* KD groups exhibited reductions in methylation levels at most of these fully methylated sites in the 18S and 28S rRNAs (Figs. 4B, C, light purple boxes and EVA, B), and only had no effect on Gm1425 in 18S rRNA and Am312, Gm313 in 28S rRNA (Fig. 4B, C, light purple boxes)”.

6) The authors state that "there were limitations in assessing the ribosome biogenesis in planarians due to lack of effective antibodies." Ribosome biogenesis can be effectively monitored by assaying the processing of rRNA using northern blots. While reduced levels of 28S were measured this data does not establish the effect from fbl-1 and fbl-2 on ribosome biogenesis. A proper analysis of the rRNA processing pathway can provide essential information regarding the function of fbl-1 and fbl-2 and establish their canonical roles.

Response: We agree with the suggestion to assess the ribosome biogenesis through Northern blot analysis. In the earlier version of our manuscript, we did not mention that there is an additional limitation exists due to the incomplete annotated information in the planarian genome, a situation complicated by the high AT repeats (up to 70%) (Grohme *et al*, 2018). This scarcity of complete genomic information constrained us to the knowledge of merely the 18S rRNA and partial sequence of the 28S rRNA, which hindered our ability to design suitable probes for Northern blot. This is the reason why we initially refrained from performing the Northern blot.

To strengthen our study, we searched the recently updated planarian genome. We found that the rDNA sequence published in Kim *et al*, 2019 could be aligned with the planarian genome. The alignment helps us find the sequence of transcribed spacers between the 18S and 28S rRNA, but still lacks sequence annotation information. Furthermore, we found an earlier report about the 18S and 28S sequences (Carranza *et al*, 1999), which could also be aligned with the genome. The sequences of internal transcribed spacers ITS1 and ITS2, as well as 5.8S rRNA, were annotated in this earlier study. Therefore, we have updated our annotations for ITS1 (952 nt), ITS2 (545 nt), and 5.8S rRNA (124 nt), as shown in Figure 1A below. Unexpectedly, we also discovered a high number of AT repeats in the unknown sequence region of the 28S rRNA in planarian (Figure 1B), which cannot be aligned with other species. These observations indicate the difference in 47S rRNA between humans and planarians (Figure 1A, B).

Based on the design of the Northern blot for the pre-rRNA transcript, the 18S, 5.8S, and 28S rRNAs are flanked by the 5' and 3' external transcribed spacers (ETS), ITS1 and ITS2. The generation of precursors of 18S and 28S rRNA is dependent on cleavage within the ETS and ITS. Conducting a Northern blot is necessary for designing specific probes in the ETS and ITS regions to examine this process, which requires the annotation of the genome in the rDNA region. The mapping of cleavage sites of human, mouse, and yeast was well studied previously through S1 nuclease mapping or Northern blot hybridization (Mullineux & Lafontaine, 2012). In recent years, five cleavage sites for 18s rRNA maturation were identified via RT-PCR and Northern blot under *RNA terminal phosphate cyclase like 1 (rc1)* mutant conditions in zebrafish (Zhu *et al*, 2021). However, the process of pre-rRNA processing in other eukaryotes remains poorly understood. The nucleotides and position of the sites varied from the aforementioned species (Figure 1C).

Figure 1. Schematic of 47S rRNA of planarians. **A.** The graphic shows the difference of 47S rRNA between human and planarian. **B.** The graphic shows the alignments of 28S rRNA in the position of planarian missing sequence among planarians, human, mouse, Xenopus, and

zebrafish. **C.** The cleavage sites of 47S rRNA in human, mouse, zebrafish and probes designed in planarians.

Although we were not sure about the specific cleavage sites in the 28S rRNA, ITS1, and ITS2 regions, we designed three probes (Figure 5C, indicated in red lines) of approximately 200 bp for hybridization to observe which rRNA processing pathways were imbalanced after *fbl* KD. Moreover, planarian 28S rRNA is processed into two fragments, α and β , through the removal of a short sequence in the hidden break. To distinguish between the mature forms of 18S rRNA and the 28S rRNA fragments, we have also designed specific probes for both (Fig. 5C). Our results showed that, compared to *egfp* KD controls, *fbl* KD led to a comparable level of 18S rRNA but a decreased level of mature 28S rRNA (Figs. 5D, EV5C). Furthermore, the maintenance of mature 28S rRNA was differently regulated by two *fbl*, that the fragment β of 28S rRNA was increased after *fbl-1* KD, while an unidentified band was detected after *fbl-2* KD, suggesting aberrant cleavage of the 28S rRNA (Figs. 5D, EV5C). The two processing pathways were both affected by *fbl* KD, as evidenced by an imbalance between 41S and 26S intermediates after *fbl-1* KD, while increased 41S and aberrant 28S intermediates after *fbl-2* KD (Figs. 5D, EV5C). Collectively, these results underscore the critical role of *fbl-1* and *fbl-2* in the coordination and fidelity of rRNA processing pathways in planarians.

Fig. 5

- C.** Schematic of planarian precursor rRNA intermediates and mature 18S and 28S rRNA with their processing pathways. ETS, external transcribed spacer; ITS, internal transcribed spacer. The number 1 indicates the major processing pathway, and 2 indicates the minor processing pathway. Red lines indicate the sites of the designed probes.
- D.** Northern blot analysis of rRNA processing after *fbl* KD. Intermediate rRNAs were detected using DIG-labeled ITS1 and ITS2 probes. Mature 18S and two fragments (α and β) of 28S rRNAs were detected using DIG-labeled 18S and 28S probes,

respectively. The loading control *gapdh* was detected using a DIG-labeled probe.

Fig. EV5C. Quantification of rRNA intermediates and mature rRNAs. Precursor rRNA 47/45S, intermediates 30S, 26S, 41S, 36/32S, 18S, and 28S α and β rRNAs are normalized to *gapdh*. $n = 3$. One-way ANOVA with Sidak's test calculated adjusted p values. ns, no significance; *, $p < 0.05$; **, $p < 0.01$; ***, $p < 0.001$.

We have included the results in lines 358-379 and Figures 5 and EV5: "The biogenesis of mature rRNA from the 47S pre-rRNA precursor involves two distinct maturation pathways that result in the generation of 18S, 28S, and 5.8S rRNAs. The major pathway proceeds through intermediates 45S pre-rRNA to 30S and 32S species, while the minor pathway involves intermediates 45S, 41S, then 21S and 32S species (Fig. 5C). To investigate the impact of *fbl* KD on pre-rRNA processes, northern blot analysis was conducted using probes specific to ITS1 and ITS2 regions (Fig. 5C). Moreover, planarian 28S rRNA is processed into two fragments, α and β , through the removal of a short sequence in the hidden break. After heat denaturation, fragment α has a length similar to that of 18S rRNA (Kim *et al*, 2019; Natsidis *et al*, 2019; Sun *et al*, 2012). To distinguish between the mature forms of 18S rRNA and the 28S rRNA fragments, we have designed specific probes for both (Fig. 5C). The results showed that, compared to *egfp* KD controls, *fbl* KD led to a comparable level of 18S rRNA but a decreased level of mature 28S rRNA (Figs. 5D, EV5C). Furthermore, the maintenance of mature 28S rRNA was differently regulated by two *fbl*, that the fragment β of 28S rRNA was increased after *fbl-1* KD, while an unidentified band was detected after *fbl-2* KD, suggesting aberrant cleavage of the 28S rRNA (Figs. 5D, EV5C). Both processing pathways were affected by *fbl* KD, supported by increased 41S and 26S intermediates after *fbl-1* KD, whereas increased 41S and aberrant 28S intermediates after *fbl-2* KD (Figs. 5D, EV5C). Collectively, these results underscore the critical role of *fbl-1* and *fbl-2* in the coordination and fidelity of rRNA processing pathways in planarians."

7) *The puromycylation assays, in their current format, do not provide a measure of translation rate as they were performed at a single timepoint. The word "rate" should be removed from description of data related to figures 5C-F.*

Response: We have changed the word “rate” to “level”.

Figure 5C- some negative controls (+CHX) show as much signal as samples, invalidating the positive signal. This experiment should be repeated or an explanation of why 48 hpa samples are translationally active in the presence of CHX should be provided.

Response: Thanks for pointing it out and allowing us to describe the results in detail. As we described the experimental procedure in the Method section, we used multiple replicates, and worms incubated in CHX and puromycin served as the positive control to compare with groups incubated in puromycin alone for 24 hours. However, this approach cannot rule out the possibility that the time window for the effects of the two chemicals is not the same, which may result in a mild band in the lane of the CHX group. We consider it as the variation of this assay, which could be assessed by quantification comparison between different conditions. To better understand our results, we have added the parallel quantification of puromycin-labeled protein from the lanes of groups incubated in CHX and puromycin to show the significant reduction of puromycin incorporation after CHX treatment, which is presented in the revised figure EV5E (original 5D) and 5F. Meanwhile, we have also revised the description of the results:

“To evaluate the protein synthesis level, we collected protein samples of intact worms and at 48 hpa with 3 biological replicates after puromycin treatment with or without cycloheximide (CHX) for 24 hours. The protein synthesis level was detected using an anti-puromycin antibody by western blot and normalized to total protein indicated by Ponceau S staining. The puromycin labeling of protein was found to be reduced in the presence of CHX in both the intact and 48 hpa groups. A significant increase in protein synthesis was found during planarian regeneration (48 hpa) compared with homeostasis (intact) (Fig. EV5D, E), underscoring the vital role of protein synthesis regulation during this process. Furthermore, we evaluated the global protein synthesis level following the inhibition of *tbl* during regeneration. Under puromycin and CHX treatment, global protein synthesis was reduced in the control group (*egfp* KD) (Fig. 5E, F). Compared with the control group, global protein synthesis was found to exhibit no significant change after *tbl-1* KD and was slightly increased following *tbl-2* KD at 48 hpa (Fig. 5E, F).”

We have also tested other conditions. Alternatively, if worms were incubated with CHX before being incorporated with puromycin, this strategy caused less efficiency in the conjugation of puromycin, which might be due to some unknown effects by CHX. Moreover, the concentration of CHX we used is sufficiently high to cause the dispatch of the pharynx within 10 min. Therefore, the protocol in our manuscript to incubate worms with puromycin and CHX is the best thus far.

8) In EV6E, although the authors are focused on neoblasts, the most effected cell lineages seem to be neural and epidermal cells. Can the authors clarify the rationale behind the specific focus on neoblasts?

Response: The reason why we focus on gene expression in neoblasts is that *fbl-1* is distinctly expressed in neoblasts, as shown in Fig. 2D. The indirect effect may explain the effects on the lineage of neurons and epidermis are likely to be indirect effect resulting from the defects in neoblasts. To study the function of *fbl-1* in neoblasts, we, therefore, chose genes enriched in neoblasts as our candidates. To clarify this point, we have added a sentence in the manuscript “Since the expression of *fbl-1* is enriched in neoblasts, to specifically study the function of *fbl-1* in neoblasts, we further integrated previous single-cell RNA-seq data to categorize genes based on their expression in different cell types”.

Minor Comments:

1) In Figure 3I- proliferation is decreased with loss of *fbl-2* based on a decreased H3P stain. However, the H3P stain in either the control or the KD is not visible. H3P is a faint stain, however, it must be visible in the figure and currently it is not.

Response: we have improved the images in Figure 3I.

Fig. 3I. FISH images show no noticeable reduction of stem cells (*piwi-1*⁺) but fewer proliferating cells (H3P⁺) at 14 dpf in *fbl-2* KD animals compared with those in *egfp* KD controls. Scale bar = 100 μ m.

2) In discussion of the results in figure 6 and EV6, the authors discuss the FDR rate that they used for a cutoff, but not the fold change cutoff for analysis of differential expression. Please include this, especially if it changes between figures.

Response: Sorry for the confusion. In this study, we used both FDR and the log2 fold change 0.8 as the cutoff. We have revised the sentence and highlighted the cutoff threshold, which is also listed in Supplementary table 1 and 2.

The authors should also include images of the H3P stain in figure 6, either in the main figure or the supplement.

Response: We have included the images of the H3P staining.

Fig. 6

B. Live images and staining of H3P⁺ cells show defects of *snrpG* KD and *sf3b5* KD animals compared to *egfp* KD controls at 7 dpf. n = 30. Scale bar = 200 μm.

D. Live images and staining of H3P⁺ cells show defects of *polr2i* KD animals compared to *egfp* KD controls at 7 dpa. n = 30. Scale bar = 200 μm.

3) *There are several instances in which the grammar can be revised. Some of the observations are difficult to understand due to complicated phrasing. Simple sentences are better for discussing complicated observations.*

Response: Thank you for your suggestion. We have revised the sentences in the manuscript and marked in red together with other revisions.

4) *It would make the data more robust to show data points and error bars above and below on the bar graphs for qPCR. Also, there is no mention of how the authors calculated relative expressions for qPCR, this is important to include.*

Response: We performed qPCR using primers for tubulin as the internal control. The expression levels of the genes of interest were calculated relative to the expression of tubulin.

In addition, we have revised the format of the plots and provided the source data together with the revised manuscript for submission.

Fig. EV1

- C. Relative mRNA level of *fbl-1* and *fbl-2* in *fbl-1* KD animals measured by quantitative PCR. n = 3 replicates in each of the two independent experiments. Each dot represents an individual replicate. Two-tailed unpaired student's *t*-test calculated the p values. ns, no significance; ****, p < 0.0001.
- D. Relative mRNA level of *fbl-1* and *fbl-2* in *fbl-2* KD animals measured by quantitative PCR. n = 3 replicates in each of the two independent experiments. Each dot represents an individual replicate. Two-tailed unpaired student's *t*-test calculated the p values. ns, no significance; ****, p < 0.0001.

5) *Statistically non-significant results should not be interpreted as a change (e.g. a marginal decrease in 18S).*

Response: We have deleted the statements.

6) *Including diagrams for the region of the planaria that we are looking at (if different from image to image in a figure or between figures) would be helpful in understanding regeneration vs differentiation. This is done in some cases but not all. For example, in Fig 2D, please provide a diagram of what region we are zooming into.*

Response: We have added a diagram of the region to indicate the region in Fig 2D.

Fig. 2D. Dual FISH of *fbl-1* or *fbl-2* transcripts with pan-neoblast marker (*piwi-1*) and epidermal lineage markers (*prog-1*, *egr-5*, *AGAT-1*, *AGAT-3*, *zpuf-6*, *vim-1*, and *vim-3*) in intact worms. The percentages indicate the ratio of *fbl-1*+ or *fbl-2*+ cells within each cell type. White arrows highlight the double-positive cells. n = 3. Data are represented as mean ± SEM. Scale bar = 50 μm.

7) *The blue on black text in all figures is hard to read. Use a different blue.*

Response: Sorry for the confusion. We have changed the color in figures.

8) *EV3 a dashed border around the planaria would be helpful for visualizing the stains, as some are very faint*

Response: We have revised the figures according to your suggestion.

Fig. 2A. The expressed patterns of *fbl-1* and *fbl-2* in planarians by FISH. The order of expression pattern is shown as the z stack in the cartoon illustration. The color lines in the cartoon illustration indicate the displayed focal panels. Scale bar = 200 μ m.

Fig. 3A. FISH images show the stem cells (*piwi-1*⁺) and proliferating cells (H3P⁺) at 14 dpf in *egfp* KD control and *fbl-1* KD animals. Scale bar = 100 μ m.

9) Line 433 replace was with is

Response: We have revised it.

10) Line 443 should say enriched and the word expressed can be removed

Response: Thanks again for the suggestion. We have revised it.

11) Line 480 remove "a content of"

Response: We have deleted the phrase in the revised manuscript.

References

- Carranza S, Baguña J, Riutort M (1999) Origin and Evolution of Paralogous rRNA Gene Clusters Within the Flatworm Family Dugesidae (Platyhelminthes, Tricladida). *Journal of Molecular Evolution* 49: 250-259
- Gąsiorowski L, Chai C, Rozanski A, Purandare G, Ficze F, Mizi A, Wang B, Rink JC (2024) Regeneration in the absence of canonical neoblasts in an early branching flatworm. *bioRxiv*
- Grohme MA, Schloissnig S, Rozanski A, Pippel M, Young GR, Winkler S, Brandl H, Henry I, Dahl A, Powell S *et al* (2018) The genome of *Schmidtea mediterranea* and the evolution of core cellular mechanisms. *Nature* 554: 56-61
- Jansson MD, Hafner SJ, Altinel K, Tehler D, Krogh N, Jakobsen E, Andersen JV, Andersen KL, Schoof EM, Menard P *et al* (2021) Regulation of translation by site-specific ribosomal RNA methylation. *Nat Struct Mol Biol* 28: 889-899
- Kim IV, Ross EJ, Dietrich S, Döring K, Sánchez Alvarado A, Kuhn C-D (2019) Efficient depletion of ribosomal RNA for RNA sequencing in planarians. *BMC Genomics* 20
- Krogh N, Jansson MD, Hafner SJ, Tehler D, Birkedal U, Christensen-Dalsgaard M, Lund AH, Nielsen H (2016) Profiling of 2'-O-Me in human rRNA reveals a subset of fractionally modified positions and provides evidence for ribosome heterogeneity. *Nucleic Acids Res* 44: 7884-7895
- Mullineux S-T, Lafontaine DLJ (2012) Mapping the cleavage sites on mammalian pre-rRNAs: Where do we stand? *Biochimie* 94: 1521-1532
- Natsidis P, Schiffer PH, Salvador-Martínez I, Telford MJ (2019) Computational discovery of hidden breaks in 28S ribosomal RNAs across eukaryotes and consequences for RNA Integrity Numbers. *Scientific Reports* 9
- Sun S, Xie H, Sun Y, Song J, Li Z (2012) Molecular characterization of gap region in 28S rRNA molecules in brine shrimp *Artemia parthenogenetica* and planarian *Dugesia japonica*. *Biochemistry (Moscow)* 77: 411-417
- Zhu Q, Tao B, Chen H, Shi H, Huang L, Chen J, Hu M, Lo LJ, Peng J (2021) Rcl1 depletion impairs 18S pre-rRNA processing at the A1-site and up-regulates a cohort of ribosome biogenesis genes in zebrafish. *Nucleic Acids Research* 49: 5743-5759

Dear Kai,

Thank you for submitting a revised version of your manuscript. I have now gone through your response to the remaining reviewers' comments, and I find it reasonable. Therefore, there now remain only a few editorial points that need to be addressed before I can extend official acceptance of the manuscript:

1. Please check that the funding information is correct and identical both in the manuscript and our online system. Currently, Zhejiang Provincial Key Laboratory Construction Project and the start-up fund from the Westlake Education Foundation are missing in our online system.
2. Please submit a complete author checklist, which you can download from our author guidelines (<https://www.embopress.org/pb-assets/embo-site/EMBO%20Press%20Author%20Checklist-1642513524327.xlsx>). Please insert information in the checklist that is also reflected in the manuscript. The completed author checklist will also be part of the Review Process File.
3. Please remove figures from the manuscript text file.
4. Please assemble main and EV figure legends at the end of the manuscript text file.
5. Please rename "Code and data availability section" into "Data availability section" and move before "Acknowledgments".
6. Please rename "Material and Methods" section into "Methods".
7. CRedit has replaced the traditional author contributions section because it offers a systematic, machine-readable author contributions format that allows for more effective research assessment. Please remove the Authors Contributions from the manuscript and use the free text boxes beneath each contributing author's name in our online submission system to add specific details on the author's contribution. More information is available in our guide to authors.
8. Please rename "Conflict of interest" section into "Disclosure and competing interests statement" (further info: <https://www.embopress.org/page/journal/14602075/authorguide#conflictsofinterest>).
9. Please remove Reagents and Tools Table from the manuscript text and upload it as a separate file choosing the file type "Reagent Table".
10. Please rename Supplementary Table 1-4 into Dataset EV1-EV4 and update the nomenclature in the manuscript accordingly.
11. In our standard image integrity check, we noted a reuse of the image panels between the following figures:
 - Fig.3L & Figure.EV7B
 - Fig.6D & Figure.EV6M
 - Fig.6B & Figure.EV7D
 - Fig.6D & Figure.EV6MIf this is intentional, please indicate in the figure legends that the images are derived from the same experiment/sample.
12. Our data editors have flagged the following issues in figure legends that need correcting:
 - Please note that the figures 2D-F does not contain any error bars, please correct the information in the figure legend.
 - Please provide the exact p values in the legends of figures 1F-I; 3B, D, F, H, J, K, M, N; 4B-I; 5B, F; 6C, E; 7E', E', H; EV1 C, D, F; EV2 B, C', D'; EV3 B, D, E, H, I; EV4 I, K, L, M; EV5 B, C, E; EV6 N; EV7 C, E, F; EV8 E, F.
 - Please indicate the statistical test used for data analysis in the legends of figures EV7 G, H.
 - Please provide information on the number and nature of replicates in the legends of figures 3F, 7E'-E'; EV2 B; EV3 B; EV4 I, K-M; EV6 N; EV8 E, F.
 - Please describe the nature of replicates, e.g., biological or technical, in the legend of figure EV5 C.
 - Please define the error bars in the legends of figures 3B, D, F, H, J, K, M, N; 4B-I; 5F; EV2 B, C', D'; EV3 D, E, H, I; EV5 B, C, E; EV6 N.
 - Please define the scale bar for figure 3K.
 - Please define the white arrow heads in the legend of figures 3C, E, G.
 - Please define the white dotted lines in the legend of figures 3A, I; 6B, D; EV3 A.
 - Please note that the figure 2C does not contain white arrows, however they are defined in the legends. Please correct the figure legend appropriately.
13. Papers published in The EMBO Journal are accompanied online by a 'Synopsis' to enhance discoverability of the manuscript. It consists of A) a short (1-2 sentences) summary of the findings and their significance, B) 3-4 bullet points highlighting key results and C) a synopsis image that is 550x300-600 pixels large (width x height, jpeg or png format). You can either show a model or key data in the synopsis image. Please note that the image size is rather small and that text needs to be readable at the final size. Please send us this information together with the revised manuscript.

With best wishes,

leva

leva Gailite, PhD
Senior Scientific Editor
The EMBO Journal
Meyerhofstrasse 1
D-69117 Heidelberg
Tel: +4962218891309
i.gailite@embojournal.org

We realize that it is difficult to revise to a specific deadline. In the interest of protecting the conceptual advance provided by the work, we recommend a revision within 3 months (20th Jan 2025). Please discuss the revision progress ahead of this time with the editor if you require more time to complete the revisions.

The authors addressed the remaining editorial issues.

Dear Kai,

I hope you are either still enjoying your European sojourn, or have already safely returned from your travels. Thank you for addressing the final formatting points. I am now happy to inform you that your manuscript has been accepted for publication.

Before we forward your manuscript to our publishers, I would like to propose some edits in the manuscript title, abstract and synopsis (please see below and the attached file). I have also written a short blurb that will accompany the title of your manuscript in our online system. Please let me know if any corrections or adjustments are needed.

Title:

Fibrillarin homologs regulate translation in divergent cell lineages during planarian homeostasis and regeneration

Blurb:

The duplicate RNA 2'-O-methyltransferase genes regulate translation efficiency of cell type-specific RNA pools in the regenerating planarian *Schmidtea mediterranea*.

Synopsis:

The role of translation regulation in planarian regeneration remains largely unknown. This study shows that two fibrillarin genes, *fbl-1* and *fbl-2*, regulate translation specificity via rRNA 2'-O methylation and pre-rRNA processing during regeneration in the planarian *Schmidtea mediterranea*.

- *fbl-1* and *fbl-2* are required for planarian homeostasis and regeneration.
- *fbl-1* and *fbl-2* display complementary expression patterns and functions in epidermal cell lineage development.
- *fbl-1* and *fbl-2* are required for 2'-O-methylation and fidelity of pre-rRNA processing.
- *fbl-1* and *fbl-2* regulate translation efficiency in neoblasts and epidermal cells, respectively.

Finally, we would like to promote your manuscript among the Chinese readership. Therefore, we would like to invite you to prepare a short summary of the manuscript in Chinese (1500-2000 Chinese characters), which we will promote on the WeChat platform 'BioArt' with more than 610,000 followers.

If you are interested in this opportunity, we recommend covering the article very close to its online publication date. Thus, ideally we would very much appreciate if you could send us a draft within the next 7 working days. Please let us know whether or not you would be interested in contributing such a short summary in Chinese.

I have included below some general guidelines on how to prepare a summary and a link to recent examples for your reference. Please let me know if you have any questions about this.

If you have any questions, please do not hesitate to contact the Editorial Office. Thank you for this contribution to The EMBO Journal and congratulations on a nice study!

With best wishes,

leva

leva Gailite, PhD
Senior Scientific Editor
The EMBO Journal
Meyerhofstrasse 1
D-69117 Heidelberg
Tel: +4962218891309

General WeChat Summary Guidelines

1. These summary articles are meant to be targeting general audience so please limit the use of specialized technical terms, acronyms and jargon.
2. A summary usually starts with brief background information of the reported work, which is followed by explaining the findings in some detail, and ends with a short review of the conclusions as well as the implications of the work and future directions for the research.
3. The summary should at least contain one graphical item, such as a scheme or a figure from the paper.
4. Please provide ONE SINGLE document containing all text and graphical materials, ideally as a Word.docx or .doc file. Please DO NOT provide the document as a .pdf file.
5. Please DO NOT publicly release the document before the paper is officially published online.

Summary Examples

EMBO J | 罗招庆/欧阳松应揭示谷酰胺脱氨酶MvcA的去泛素化功能

EMBO J | 王松灵院士团队揭示组织内应力调控大型哺乳动物乳恒牙替换的新机制
